# THE IMPLICIT BIAS OF STRUCTURED STATE SPACE MODELS CAN BE POISONED WITH CLEAN LABELS

## ABSTRACT

Neural networks are powered by an implicit bias: a tendency of gradient descent to fit training data in a way that generalizes to unseen data. A recent class of neural network models gaining increasing popularity is structured state space models (SSMs), regarded as an efficient alternative to transformers. Prior work argued that the implicit bias of SSMs leads to generalization in a setting where data is generated by a low dimensional teacher. In this paper, we revisit the latter setting, and formally establish a phenomenon entirely undetected by prior work on the implicit bias of SSMs. Namely, we prove that while implicit bias leads to generalization under many choices of training data, there exist special examples whose inclusion in training completely distorts the implicit bias, to a point where generalization fails. This failure occurs despite the special training examples being labeled by the teacher, *i.e.* having clean labels! We empirically demonstrate the phenomenon, with SSMs trained independently and as part of non-linear neural networks. In the area of adversarial machine learning, disrupting generalization with cleanly labeled training examples is known as clean-label poisoning. Given the proliferation of SSMs, particularly in large language models, we believe significant efforts should be invested in further delineating their susceptibility to clean-label poisoning, and in developing methods for overcoming this susceptibility.

## 1 INTRODUCTION

Overparameterized neural networks can fit their training data in multiple ways, some of which generalize to unseen data, while others do not. Remarkably, when the training data is fit via gradient descent (or a variant thereof), generalization tends to occur. This phenomenon—one of the greatest mysteries in modern machine learning (Zhang et al. (2021); Chatterjee and Zielinski (2022))—is often viewed as stemming from an *implicit bias*: a tendency of gradient descent, when applied to neural network models, to fit training data in a way that complies with common data-generating distributions. The latter view was formalized for several neural network models and data-generating distributions (Soudry et al. (2018); Gunasekar et al. (2018); Razin et al. (2022); Neyshabur (2017)).

A recent class of neural network models gaining increasing popularity is *structured state space models* (*SSMs*). SSMs are often regarded as a computationally efficient alternative to transformers (Wang et al. (2022)), and underlie prominent neural networks such as S4 (Gu et al. (2021)), Mamba (Gu and Dao (2023)), LRU (Orvieto et al. (2023)), Mega (Ma et al. (2023)), S5 (Smith et al. (2023)) and more (Poli et al. (2023); Dao and Gu (2024)). The implicit bias of SSMs, *i.e.*, of gradient descent over SSMs, was formally studied in prior works, *e.g.* Emami et al. (2021); Cohen-Karlik et al. (2022; 2023). Notable among these is Cohen-Karlik et al. (2023), which considered a setting where data is generated by a low dimensional teacher SSM, and gradient flow (gradient descent with infinitesimally small step size) applied to a high dimensional student SSM fits training data comprising infinitely many sequences of a certain length.[1] In this setting, the student SSM can fit the training data in multiple ways, some of which generalize to sequences longer than those seen in training, while others do not. It was shown in Cohen-Karlik et al. (2023) that under mild conditions, an implicit bias leads to generalization.

---

[1] More precisely, the training data is formed from a continuous (Gaussian) distribution of sequences having a certain length, all labeled by the teacher SSM.

In this paper, we revisit the setting of Cohen-Karlik et al. (2023), with one key exception: rather than training data comprising infinitely many sequences, we consider the realistic case where the number of sequences is finite. Surprisingly, our theory and experiments reveal a phenomenon entirely undetected by prior works on the implicit bias of SSMs. Namely, we find that while implicit bias leads the student SSM to generalize under many choices of sequences to include in training, there exist special sequences which if included in training completely distort the implicit bias, resulting in the student SSM failing to generalize. This failure to generalize takes place despite the fact that the special sequences are labeled by the teacher SSM, *i.e.* they have clean labels! In the area of adversarial machine learning, the phenomenon of generalization being disrupted by training instances with clean labels is known as *clean-label poisoning*, and received significant attention in recent years, both empirically (Huang et al. (2020); Shafahi et al. (2018)) and theoretically (Suya et al. (2021); Blum et al. (2021)). To our knowledge, the current paper is the first to formally prove susceptibility of SSMs to clean-label poisoning.

Our theoretical analysis comprises two main results, which may be of independent interest. First, is a dynamical characterization of gradient flow over an SSM, trained individually or as part of a non-linear neural network. The dynamical characterization reveals that *greedy low rank learning* (Sun et al. (2021); Li et al. (2020); Razin et al. (2021; 2022))—a sufficient condition for generalization with a low dimensional teacher SSM—is implicitly induced under many, but not all, choices of training sequences. Our second theoretical contribution builds on our dynamical characterization for a fine-grained analysis of gradient flow over an SSM, employing an advanced tool from dynamical systems theory: a *non-resonance linearization theorem* (Sell (1985)). The analysis proves that there exist situations where: *(i)* training a student SSM on a collection of sequences labeled by a low dimensional teacher SSM exhibits an implicit bias that leads to generalization; and *(ii)* adding to the training set a single sequence, also labeled by the teacher SSM (*i.e.*, that also has a clean label), entirely distorts the implicit bias, to an extent where generalization fails.

We corroborate our theory via experiments, which demonstrate how adding a small amount of cleanly labeled sequences to the training set of an SSM can completely ruin its generalization. In light of the growing prominence of SSMs, particularly in the context of large language models, we believe significant research efforts should be invested in further delineating their susceptibility to clean-label poisoning, and in developing methods for overcoming this susceptibility.

## 2 PRELIMINARIES

### 2.1 NOTATIONS

We use non-boldface lowercase letters for denoting scalars (*e.g.*, $\alpha \in \mathbb{R}$, $d \in \mathbb{N}$), boldface lowercase letters for denoting vectors (*e.g.*, $\mathbf{x} \in \mathbb{R}^d$), and non-boldface uppercase letters for denoting matrices (*e.g.*, $A \in \mathbb{R}^{d \times d}$). For $d \in \mathbb{N}$, we let: $\mathbf{1}_d$ be the all-ones vector of dimension $d$; $\mathbf{0}_d$ be the all-zeros vector of dimension $d$; and $[d]$ be the set $\{1, 2, \ldots, d\}$. For $d \in \mathbb{N}$ and $i \in [d]$, we denote by $\mathbf{e_i}$ the $i$'th standard basis vector (*i.e.*, a vector holding one in entry $i$ and zeros elsewhere) of dimension $d$, where for simplicity the dimension is omitted from the notation and should be inferred from context. Scalar series of finite lengths are identified with vectors.

### 2.2 STRUCTURED STATE SPACE MODELS

A *structured state space model* (*SSM*) *of dimension* $d \in \mathbb{N}$ is parameterized by three matrices: $A \in \mathbb{R}^{d,d}$, a *state transition matrix*, which conforms to a predefined structure (*e.g.* is constrained to be diagonal); $B \in \mathbb{R}^{d,1}$, an *input matrix*; and $C \in \mathbb{R}^{1,d}$, an *output matrix*. Given the values of $A$, $B$ and $C$, the SSM realizes a mapping $\phi_{(A,B,C)}(\cdot)$ that receives as input a length $k$ scalar sequence $\mathbf{x} \in \mathbb{R}^k$, for any $k \in \mathbb{N}$, and produces as output a scalar $y \in \mathbb{R}$ equal to the last element of the series $\mathbf{y} \in \mathbb{R}^k$ defined through the following recursive formula:

$$\mathbf{s}_{k'} = A\mathbf{s}_{k'-1} + Bx_{k'} \ , \ y_{k'} = C\mathbf{s}_{k'} \ , \ k' \in [k], \tag{1}$$

where $(\mathbf{s}_{k'} \in \mathbb{R}^d)_{k' \in [k]}$ is a sequence of *states*, and $\mathbf{s}_0 = \mathbf{0}_d$. It is straightforward to show that the mapping $\phi_{(A,B,C)}(\cdot)$ is fully determined by the sequence $(CA^{k'}B)_{k'=0}^{\infty}$, known as the *impulse response* of the SSM. In particular, for any $k \in \mathbb{N}$ and $\mathbf{x} \in \mathbb{R}^k$:

$$y = \phi_{(A,B,C)}(\mathbf{x}) = (CB, CAB, CA^2B, \ldots, CA^{k-1}B)^{\top}\mathbf{x}. \tag{2}$$

For convenience, we often identify an SSM with the triplet $(A, B, C)$ holding its parameter matrices, and regard the (single column) matrices $B$ and $C^\top$ as vectors. Perhaps the most common form of structure imposed on SSMs is *diagonality* (Gu et al. (2022); Gupta et al. (2022); Orvieto et al. (2023); Ma et al. (2023); Gu and Dao (2023)). Accordingly, unless stated otherwise, we assume that the state transition matrix $A$ of an SSM is diagonal.

Some of our results will account for SSMs that are part of non-linear neural networks—or more specifically, for SSMs whose output undergoes a transformation $\sigma(\cdot, \mathbf{w})$, where: $\sigma : \mathbb{R} \times \mathcal{W} \to \mathbb{R}$ is some differentiable mapping; $\mathcal{W}$ is some Euclidean space, regarded as a parameter space; and $\mathbf{w} \in \mathcal{W}$, regarded as a parameter vector. Given values for $A$, $B$, $C$ and $\mathbf{w}$, such a neural network realizes the mapping $\phi_{(A,B,C),\mathbf{w}}(\cdot) := \sigma(\phi_{(A,B,C)}(\cdot), \mathbf{w})$. This architecture (namely, an SSM followed by a parametric transformation) is ubiquitous among SSM-based neural networks, for example Gu et al. (2021), Gupta et al. (2022), and Gu et al. (2022).

### 2.3 TEACHER-STUDENT SETTING

We consider the *teacher-student* setting of Cohen-Karlik et al. (2023), specified hereafter. Data is labeled by a teacher SSM $(A^*, B^*, C^*)$ of dimension $d^*$, *i.e.* the ground truth label of $\mathbf{x} \in \mathbb{R}^k$, for any $k \in \mathbb{N}$, is $\phi_{(A^*,B^*,C^*)}(\mathbf{x}) \in \mathbb{R}$. For some $\kappa \in \mathbb{N}$, $\bar{t} > 2d^*$, we are given a training set $\mathcal{S}$ comprising $n$ labeled sequences of length $\kappa$, *i.e.* $\mathcal{S} := (\mathbf{x}^{(i)}, y^{(i)})_{i=1}^n$ where $\mathbf{x}^{(i)} \in \mathbb{R}^\kappa$ and $y^{(i)} = \phi_{(A^*,B^*,C^*)}(\mathbf{x}^{(i)})$ for every $i \in [n]$. A *student* SSM $(A, B, C)$ of dimension $d \in \mathbb{N}$, $d > \bar{t}$, is trained, *i.e.* optimized, by minimizing the square loss over $\mathcal{S}$, referred to as the *training loss*:

$$\ell(A, B, C; \mathcal{S}) := \frac{1}{n} \sum_{i=1}^n \left( y^{(i)} - \phi_{(A,B,C)}(\mathbf{x}^{(i)}) \right)^2. \tag{3}$$

Optimization is implemented via *gradient flow*, which is formally equivalent to gradient descent with infinitesimally small step size (learning rate), and was shown to well-approximate gradient descent so long as the step size is moderately small Elkabetz and Cohen (2021):

$$(\dot{A}(t), \dot{B}(t), \dot{C}(t)) = -\nabla \ell(A(t), B(t), C(t); \mathcal{S}) \ , \ \ t \in \mathbb{R}_{>0}, \tag{4}$$

where $(\dot{A}(t), \dot{B}(t), \dot{C}(t)) := \frac{d}{dt}(A(t), B(t), C(t))$, and $(A(\cdot), B(\cdot), C(\cdot))$ is a curve representing the optimization trajectory. Generalization of the student at time $t \in \mathbb{R}_{\geq 0}$ of optimization is measured by the extent to which $\phi_{(A(t),B(t),C(t))}(\cdot)$ approximates $\phi_{(A,B,C)}(\cdot)$, not only over input sequences of length $\kappa$ as used for training, but of other lengths as well. This allows accounting not just for in-distribution generalization as considered in classical machine learning theory (Shalev-Shwartz and Ben-David (2014)), but for out-of-distribution generalization (extrapolation) as prevalent in modern machine learning (Liu et al. (2021)). Formally, in line with Equation (2), generalization is quantified through the first $k$ entries of the student and teacher impulse responses, for different values of $k$.

**Definition 1.** The *generalization error of the student SSM over sequence length $k$* is:

$$\max_{k' \in \{0,1,\dots,k-1\}} \left| BA^{k'}C - B^*(A^*)^{k'}C^* \right|. \tag{5}$$

Clearly, there exist assignments for $(A, B, C)$ with which the training loss $\ell(\cdot)$ is minimized (*i.e.*, equals zero) and the student SSM perfectly generalizes over any sequence length $k$.[2] On the other hand, it was shown in Cohen-Karlik et al. (2023) that, regardless of the size of the training set $\mathcal{S}$ and the input sequences it comprises (namely, $(\mathbf{x}^{(i)})_{i=1}^n$), there exist assignments for $(A, B, C)$ with which the training loss $\ell(\cdot)$ is minimized and yet the student has arbitrarily high generalization error over sequence lengths beyond $\kappa$, *e.g.* $\kappa + 1$ (for completeness, we prove this fact in Section A). The latter two facts together imply that if minimization of the training loss $\ell(\cdot)$ via gradient flow (Equation (4)) produces an assignment for $(A, B, C)$ with which the student SSM generalizes, it must be a result of implicit bias. The main result in Cohen-Karlik et al. (2023) states that if the training set $\mathcal{S}$ is infinite and each entry of each input sequence $\mathbf{x}^{(i)}$ is independently drawn from the standard normal distribution (in other words, if the training loss $\ell(\cdot)$ is the expected value of $(y - \phi_{(A,B,C)}(\mathbf{x}))^2$, where the entries of $\mathbf{x}$ are independently drawn from the standard normal

---

[2]This is the case, for example, if $A$, $B$ and $C$ are respectively attained by padding $A^*$, $B^*$ and $C^*$ with zeros on the right and/or bottom.

distribution and $y = \phi_{(A^*,B^*,C^*)}(\mathbf{x})$), then under mild conditions the implicit bias of gradient flow is such that it convergences to a solution which generalizes over any sequence length $k$.

In this paper, we focus on the realistic case where the training set $\mathcal{S}$ is finite. Surprisingly, our theory and experiments (Sections 3 and 4, respectively) will reveal a phenomenon completely undetected by Cohen-Karlik et al. (2023), and any other work we are aware of on the implicit bias of SSMs.

## 3 THEORETICAL ANALYSIS

In this section we present our theoretical analysis. For streamlining the presentation, we embed definitions and assumptions in the body of the text (rather than placing them in dedicated environments). Readers who wish to view a concentrated list of all assumptions underlying each theoretical result are referred to Section B.

### 3.1 DYNAMICAL CHARACTERIZATION

In this subsection we derive a dynamical characterization of gradient flow over an SSM, trained individually or as part of a non-linear neural network. The dynamical characterization will reveal that *greedy low rank learning* (Arora et al. (2019); Li et al. (2020); Razin et al. (2021; 2022))—a sufficient condition for generalization with a low dimensional teacher SSM—is implicitly induced under many, but not all, choices of training sequences. Section 3.2 will build on the dynamical characterization to prove that the implicit bias of SSMs can be poisoned with clean labels.

Our dynamical characterization applies to a setting more general than that laid out in Section 2.3. Namely, it applies to the same setting, with two exceptions: *(i)* the student SSM is potentially embedded in a non-linear neural network, *i.e.* the mapping $\phi_{(A,B,C)}(\cdot)$ is replaced by $\phi_{(A,B,C),\mathbf{w}}(\cdot)$ as defined in Section 2.2; and *(ii)* the training labels $(y^{(i)})_{i=1}^n$ need not be assigned by a teacher SSM, *i.e.* they may be arbitrary. We denote the resulting training loss—a generalization of $\ell(\cdot)$ defined in Equation (3)—by $\tilde{\ell}(\cdot)$, namely:

$$\tilde{\ell}(A, B, C, \mathbf{w}; \mathcal{S}) := \frac{1}{n} \sum_{i=1}^n \left( y^{(i)} - \phi_{(A,B,C),\mathbf{w}}(\mathbf{x}^{(i)}) \right)^2. \tag{6}$$

Proposition 1 below establishes our dynamical characterization—equations of motion for the (diagonal) entries of $A$ during gradient flow over $\tilde{\ell}(\cdot)$.

**Proposition 1.** *Consider optimization of the generalized loss $\tilde{\ell}(\cdot)$ defined in Equation (6) via gradient flow, namely:*

$$(\dot{A}(t), \dot{B}(t), \dot{C}(t), \dot{\mathbf{w}}(t)) = -\nabla \tilde{\ell}(A(t), B(t), C(t), \mathbf{w}(t); \mathcal{S}) \ , \ t \in \mathbb{R}_{>0}, \tag{7}$$

*where $(\dot{A}(t), \dot{B}(t), \dot{C}(t), \dot{\mathbf{w}}(t)) := \frac{d}{dt}(A(t), B(t), C(t), \mathbf{w}(t))$, and $(A(\cdot), B(\cdot), C(\cdot), \mathbf{w}(\cdot))$ is a curve representing the optimization trajectory. For $j \in [d]$, denote by $a_j(\cdot)$ the $j$'th diagonal entry of $A(\cdot)$, by $b_j(\cdot)$ the $j$'th entry of $B(\cdot)$, and by $c_j(\cdot)$ the $j$'th entry of $C(\cdot)$. Assume that the training sequence length $\kappa$ is greater than or equal to two. For $l \in [\kappa]$ and $i \in [n]$, denote the $l$'th entry of the $i$'th training sequence $\mathbf{x}^{(i)}$ by $x_l^{(i)}$. Then:*

$$\dot{a}_j(t) := \frac{d}{dt} a_j(t) = b_j(t) c_j(t) \sum_{l=0}^{\kappa-2} \gamma^{(l)}(t) \cdot a_j(t)^l \ , \ j \in [d], \ t \in \mathbb{R}_{>0}, \tag{8}$$

*where:*

$$\gamma^{(l)}(t) := \frac{2(l+1)}{n} \sum_{i=1}^n \delta^{(i)}(t) \xi^{(i)}(t) x_{\kappa-l-1}^{(i)}, \tag{9}$$

*with:*

$$\delta^{(i)}(t) := y^{(i)} - \phi_{(A(t),B(t),C(t)),\mathbf{w}(t)}(\mathbf{x}^{(i)}), \tag{10}$$

$$\xi^{(i)}(t) := \frac{\partial}{\partial z} \sigma(z, \mathbf{w}(t))\big|_{z=\phi_{(A(t),B(t),C(t)),\mathbf{w}(t)}(\mathbf{x}^{(i)})}. \tag{11}$$

*Proof sketch (proof in Section C).* The desired result readily follows from differentiation of $\tilde{\ell}(\cdot)$ (Equation (6)) with respect to each diagonal entry of $A$. □

**Interpretation.** Proposition 1 (Equations (8) to (11)) implies that during gradient flow, the motion of $a_j(\cdot)$—the $j$'th diagonal entry of the state transition matrix $A(\cdot)$—is given by a degree $\kappa - 2$ polynomial in $a_j(\cdot)$, where the coefficients of the polynomial are time-varying. In particular, at time $t \in \mathbb{R}_{>0}$ of optimization, the coefficient of the $l$'th power in the polynomial, for $l \in \{0, 1, \ldots, \kappa - 2\}$, is a product of two factors: *(i)* $\gamma^{(l)}(t)$, which depends on the power $l$ but not on the entry $j$; and *(ii)* $b_j(t)c_j(t)$—$j$'th entry of the input matrix $B(\cdot)$ times the $j$'th entry of the output matrix $C(\cdot)$—which does not depend on the power $l$ but does depend on the entry $j$. Consider the case where $A(\cdot)$ emanates from standard near-zero initialization (Glorot and Bengio (2010); He et al. (2015); Parnichkun et al. (2024)), *i.e.* where $a_j(0) \approx 0$ for all $j$. If the factor $\gamma^{(0)}(\cdot)$ is small throughout—as is the case, *e.g.*, if the $\kappa - 1$'th entry of each training sequence $\mathbf{x}^{(i)}$ is small (see Equation (9))—then the constant term in the polynomial determining the motion of $a_j(\cdot)$ is negligible. The dynamics of $(a_j(\cdot))_{j=1}^d$ then exhibit greedy learning, similarly to the dynamics of various quantities in various types of neural networks (Arora et al. (2019); Li et al. (2020); Razin et al. (2021; 2022)). Namely, $(a_j(\cdot))_{j=1}^d$ all progress slowly at first, following near-zero initialization, and then, whenever an entry reaches a critical threshold, it starts moving rapidly—see empirical demonstrations in Figure 2. The greedy learning of $(a_j(\cdot))_{j=1}^d$ implies a greedy low rank learning of the state transition matrix $A$. More specifically, it implies a tendency to fit training data with $A$ having low rank, meaning a tendency to generalize if data is generated by a low dimensional teacher SSM. In stark contrast, if the training sequences $(\mathbf{x}^{(i)})_{i=1}^n$ are such that the factor $\gamma^{(0)}(\cdot)$ is not small, then the polynomials determining the motions of $(a_j(\cdot))_{j=1}^d$ have non-negligible constant terms, and greedy low rank learning will generally *not* take place—see empirical demonstrations in Figure 2 and Section F.1.

## 3.2 Clean-Label Poisoning

Building on the dynamical characterization from Section 3.1, in this subsection we provide a fine-grained analysis of gradient flow over an SSM. The analysis considers a teacher-student setting as in Section 2.3, and proves existence of situations where: *(i)* training a student SSM on a collection of sequences labeled by a low dimensional teacher SSM exhibits an implicit bias that leads to generalization; and *(ii)* adding to the training set a single sequence, also labeled by the teacher SSM (*i.e.*, that also has a clean label), entirely distorts the implicit bias, to an extent where generalization fails. To our knowledge, this constitutes the first formal proof of susceptibility of SSMs to clean label poisoning. Facilitating our analysis is an advanced tool from dynamical systems theory—a *non-resonance linearization theorem*—which may be of independent interest.

Hereinbelow we present our analysis in a basic setting, deferring more elaborate settings (all treated similarly) to Section E. Suppose the teacher SSM has dimension $d^* = 2$ and parameters:

$$A^* = \begin{pmatrix} 1 & 0 \\ 0 & 0 \end{pmatrix} \ , \ \ B^* = \begin{pmatrix} 1 & \sqrt{d-1} \end{pmatrix}^\top \ , \ \ C^* = \begin{pmatrix} 1 & \sqrt{d-1} \end{pmatrix} . \tag{12}$$

Suppose also that the state transition matrix of the student SSM $A(\cdot)$ emanates from standard near-zero initialization (Glorot and Bengio (2010); He et al. (2015); Parnichkun et al. (2024)), and its input and output matrices $B(\cdot)$ and $C(\cdot)$ are fixed at $\mathbf{1}_d$ and $\mathbf{1}_d^\top$, respectively. In this setting, a sufficient condition for the student SSM to achieve low generalization error over all sequence lengths (Definition 1) is that one of the diagonal entries of $A(\cdot)$ be close to one while the rest are close to zero. Theorem 1 below shows that the latter sufficient condition for generalization is satisfied under some choices of training sequences, and yet, despite the condition being mild, adding a single sequence labeled by the teacher SSM (*i.e.*, a single sequence that has a clean label) can entirely fail generalization.

**Theorem 1.** *Assume that the training sequence length and the dimension of the student SSM respectively satisfy $\kappa \in \{7, 9, 11, \ldots\}$ and $d \geq 8$. Let $k \in \mathbb{N}_{\geq \kappa+2}$ and $\epsilon \in \mathbb{R}_{>0}$. Then, for any $n \in \mathbb{N}$, there exist a training set $\mathcal{S} = (\mathbf{x}^{(i)}, y^{(i)})_{i=1}^n$ (where, for every $i \in [n]$, $\|\mathbf{x}^{(i)}\|_\infty = 1$), a labeled sequence $(\mathbf{x}^\dagger, y^\dagger) \in \mathbb{R}^\kappa \times \mathbb{R}$ (where the entry of $\mathbf{x}^\dagger$ with largest absolute value is the second-to-last, holding the value $n^{1/2}$), and an open set $\mathcal{I}$ of initializations for the student SSM,[3] such that, with any initialization in $\mathcal{I}$, the following holds:*

- *gradient flow converges to a point at which the training loss is minimal (i.e., equals zero) and the generalization errors over sequences lengths $1, 2, \ldots, k$ are no greater than $\epsilon$; and*

---

[3] That is, an open subset of the set of diagonal matrices in $\mathbb{R}^{d,d}$.

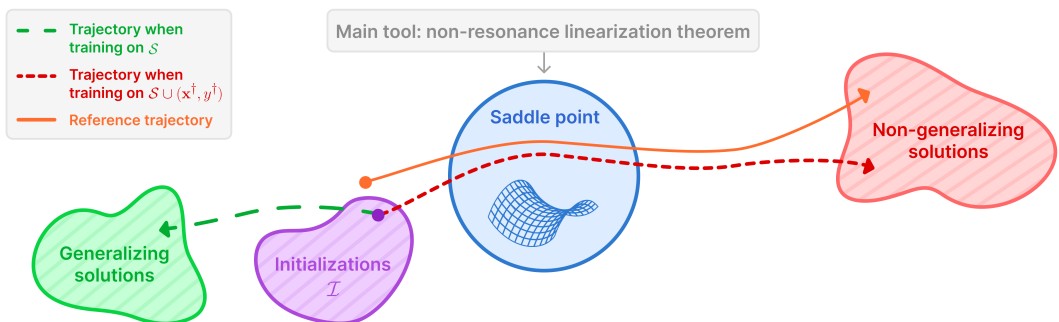

Figure 1: Illustration of the main ideas behind the proof of Theorem 1. See proof sketch for an annotation.

> - *if the labeled sequence $(\mathbf{x}^\dagger, y^\dagger)$ is appended to the training set $\mathcal{S}$, gradient flow converges to a point at which the training loss is minimal and the generalization error over sequence length $k$ is at least $\min\{0.1, 1/(9d) \cdot (1 - (0.6)^{1/(\kappa-1)})\}$.*

*Proof sketch (proof in Section D).* Figure 1 illustrates the main ideas behind the proof. Below is a description of these ideas, along with an annotation of the figure.

The proof shows that the student SSM admits low generalization error (over any sequence length) when its state transition matrix $A$ has a single (diagonal) entry close to one and the remaining entries close to zero. That is, the set labeled "generalizing solutions" in Figure 1 comprises a neighborhood of one-hot assignments for $A$. Using the dynamical characterization from Proposition 1, the proof establishes that with a properly constructed training set $\mathcal{S}$ (namely, a training set $\mathcal{S}$ without sequences in which the last elements are relatively large[4]), if gradient flow emanates from standard near-zero initialization, then it exhibits greedy low rank learning of $A$ (see interpretation following Proposition 1, as well as empirical demonstrations in Figure 2 and Section F.1), and accordingly converges to a generalizing solution. Thus, under choices of $\mathcal{I}$ (set of initializations) close to the origin, when training on $\mathcal{S}$, a gradient flow trajectory emanating from $\mathcal{I}$ converges to the set of generalizing solutions—as illustrated in Figure 1.

To analyze the behavior of gradient flow when training on $\mathcal{S} \cup (\mathbf{x}^\dagger, y^\dagger)$, the proof makes use of the structure of $\mathbf{x}^\dagger$ (namely, the fact that its last elements are relatively large[5]) to show that greedy low rank learning does not take place (see interpretation following Proposition 1, as well as empirical demonstrations in Figure 2 and Section F.1). This allows identifying certain reference trajectories that converge to non-generalizing solutions (one such reference trajectory is illustrated in Figure 1). These reference trajectories emanate from initializations that cannot be included in $\mathcal{I}$, since they lead gradient flow to converge to non-generalizing solutions even when training on $\mathcal{S}$. However, the proof shows that $\mathcal{I}$ can comprise initializations near those of reference trajectories, since under such choice of $\mathcal{I}$, each of its initializations leads gradient flow to: *(i)* converge to a generalizing solution when training on $\mathcal{S}$; and *(ii)* closely track a reference trajectory when training on $\mathcal{S} \cup (\mathbf{x}^\dagger, y^\dagger)$, resulting in convergence to a non-generalizing solution—as illustrated in Figure 1. This concludes the proof.

The main technical challenge faced by the proof lies in item *(ii)* above, namely, in establishing that when training on $\mathcal{S} \cup (\mathbf{x}^\dagger, y^\dagger)$, an initialization near that of a reference trajectory leads gradient flow to closely track the reference trajectory. Since the training loss is non-convex, gradient flow trajectories can diverge from one another exponentially fast. Establishing that a reference trajectory is tracked thus requires sharp bounds on convergence times. The crux of the challenge is to derive such bounds, as trajectories pass near saddle points, and a-priori, may not escape (the vicinities of) these saddle points sufficiently fast. To show that saddle points are escaped swiftly, the proof employs an advanced tool from dynamical systems theory which may be of independent interest: a non-resonance linearization theorem (Sell (1985)). Namely, rather than directly analyzing trajectories in

---

[4]For simplicity, the proof considers $\mathcal{S} = \{p_i \cdot \mathbf{e}_1, p_i\}_{i=1}^n$, where $\sum_{i=1}^n p_i^2 = n$, but it is possible to account for a much wider class of training sets—see Section E for details.

[5]The proof takes $\mathbf{x}^\dagger = n^{\frac{1}{2}} \mathbf{e}_{\kappa-1}$.

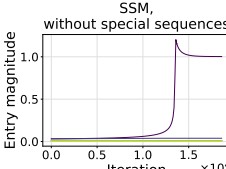 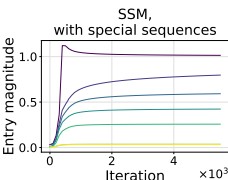 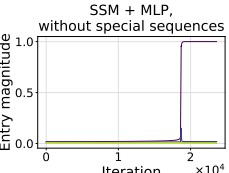 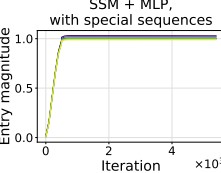

Figure 2: Demonstration of the dynamical characterization derived in Proposition 1—optimization of an SSM, trained individually or as part of a non-linear neural network, implicitly induces greedy learning of the (diagonal) entries of the state transition matrix $A$ under some, but not all, choices of training sequences. First (leftmost) plot shows the magnitudes of the entries of $A$ throughout the iterations of gradient descent, in a case where a student SSM of dimension $d = 10$ is trained individually on a training set labeled by a teacher SSM of dimension $d = 1$, and the training set does not include "special" sequences, *i.e.* sequences in which the last elements are relatively large. Second plot portrays the exact same scenario, except that special sequences are included in the training set. Third and fourth plots adhere to the descriptions of first and second plots, respectively, except that the student SSM is trained along with a successive multi-layer perceptron (non-linear neural network), and the teacher SSM is followed by a (fixed) multi-layer perceptron. Notice that, with and without a multi-layer perceptron, greedy learning takes place when special sequences are excluded, and does not take place when they are included. For further details and experiments (including teachers of higher dimension) see Sections F.1 and G.1.

the vicinity of a saddle point, the proof constructs linear approximations, and uses the non-resonance linearization theorem to show that the linear approximations are sufficiently accurate, which in turn implies that the trajectories escape the saddle point sufficiently fast. The non-resonance linearization theorem requires the spectrum of the Hessian of the training loss to be free of certain algebraic dependencies known as resonances. If these resonances are absent—which the proof shows to be the case—the non-resonance linearization theorem provides guarantees on the accuracy of linear approximations that are far better than guarantees attainable via standard smoothness arguments. □

## 4 EXPERIMENTS

This section presents experiments corroborating our theory. It is organized as follows. Section 4.1 demonstrates the dynamical characterization we derived (in Proposition 1), showcasing that optimization of an SSM implicitly induces greedy low rank learning (a sufficient condition for generalization with a low dimensional teacher SSM) under some, but not all, choices of training sequences. Section 4.2 then demonstrates the clean-label poisoning phenomenon we established (in Theorem 1), by showing that adding a small amount of cleanly labeled sequences to the training set of an SSM can completely ruin its generalization. Code for reproducing all of the experiments will be made publicly available.

### 4.1 DYNAMICAL CHARACTERIZATION

As discussed in Section 3.1, the dynamical characterization in Proposition 1 (Equations (8) to (11)) implies that optimization of an SSM—trained individually or as part of a non-linear neural network—implicitly induces greedy learning of the (diagonal) entries of the state transition matrix $A$ under some, but not all, choices of training sequences. For example, if the penultimate entry of each training sequence is small then greedy learning takes place, and if these entries are not small then greedy learning may not take place. Figure 2 clearly demonstrates this, for a standalone SSM as well as one included in a non-linear neural network. Further experiments are reported in Section F.1.

### 4.2 CLEAN-LABEL POISONING

Theorem 1 proved existence of situations where clean-label poisoning of an SSM takes place, *i.e.* situations where: *(i)* training a student SSM on a collection of sequences labeled by a low dimensional teacher SSM exhibits an implicit bias that leads to generalization; and *(ii)* adding to the training set a sequence also labeled by the teacher SSM (*i.e.*, that also has a clean label) entirely distorts the implicit bias, to an extent where generalization fails. Table 1 demonstrates clean-label poisoning of SSMs in three different settings: the setting of Theorem 1; an SSM setting beyond Theorem 1 (*i.e.*,

an SSM setting that does not satisfy the assumptions of Theorem 1, *e.g.* it includes learning of the input and output matrices $B$ and $C$, respectively); and a setting where an SSM is part of a non-linear neural network.

We further demonstrate clean-label poisoning of SSMs in a real-world (non-synthetic) setting. To that end, we employ an adapted version of a method known as Gradient Matching (Geiping et al., 2020) for generating cleanly labeled examples that poison (*i.e.*, that disrupt generalization for) an SSM-based S4 neural network (Gu et al., 2021) trained on the sequential MNIST dataset (Deng, 2012). In this experiment we do not have access to a teacher (*i.e.*, to a ground truth labeling function), and accordingly, a cleanly labeled poisonous example is generated from a given training example by introducing human-imperceptible noise to the input sequence, while keeping the label intact. The noise introduced for generating a cleanly labeled poisonous example has its last entries relatively large, in line with our theory. After training with cleanly labeled poisonous examples as above, the S4 neural network misclassifies preselected test instances—see Table 2.

We believe the susceptibility of SSMs to clean-label poisoning goes far beyond the demonstrations herein. In light of the growing prominence of SSMs, particularly in the context of large language models, further delineating this susceptibility, and developing methods for overcoming it, are of prime importance.

Table 1: Demonstration of clean-label poisoning of SSMs in three different settings: the setting of Theorem 1; an SSM setting beyond Theorem 1 (*i.e.*, an SSM setting that does not satisfy the assumptions of Theorem 1, *e.g.* it includes learning of the input and output matrices $B$ and $C$, respectively); and a setting where an SSM is part of a non-linear neural network, *i.e.* is followed by a multi-layer perceptron. In each setting, a high dimensional student is trained until convergence (*i.e.*, until the training loss is lower than 0.01), and data is generated (*i.e.*, sequences are labeled) by a low dimensional teacher of the same architecture as the student. Reported are generalization errors (each averaged over 4 random seeds) for two training sets per setting: a training set that does not include "special" sequences, *i.e.* sequences in which the last elements are relatively large; and a training set that does include such sequences. In the first two settings (SSMs trained independently) generalization errors are measured via impulse responses, as defined in Definition 1. In the third setting (SSM trained as part of non-linear neural network) generalization errors are measured using a held-out test set. All reported generalization errors were normalized (scaled) such that a zero mapping corresponds to a value of one. Notice that across all settings, special training sequences significantly deteriorate generalization. For further details and experiments (including teachers of higher dimension) see Sections F.2 and G.2.

| Setting | Without special sequences | With special sequences |
|---|---|---|
| SSM per Theorem 1 | $1.34 \times 10^{-3}$ | $4.1 \times 10^{-2}$ |
| SSM beyond Theorem 1 | $2.13 \times 10^{-1}$ | 24.66 |
| SSM in non-linear neural network | $2.92 \times 10^{-3}$ | $8.93 \times 10^{-2}$ |

## 5 RELATED WORK

SSMs can be viewed as a special case of *linear dynamical systems* (*LDSs*)—a classic object of study in areas such as systems theory (Oppenheim et al. (1996)) and control theory (Sontag (1990)). The problem of learning from data an SSM that admits in-distribution and out-of-distribution generalization is an instance of what is known in the LDS literature as *system identification* (Simpkins (2012)). Determination of whether a high dimensional SSM realizes a mapping that is also realizable by a low dimensional SSM (in our context, these are a student and a teacher, respectively) is considered in the LDS literature under the topic of minimal realization theory (Silverman (1971)). Despite these connections, our work is clearly distinct from classic LDS literature: it studies the implicit bias of gradient descent—a phenomenon brought to light by the recent rise of overparameterized neural networks (Neyshabur (2017)).

Several recent works formally studied the implicit bias of gradient descent in the context of recurrent neural networks (Lim et al. (2021); Emami et al. (2021); Cohen-Karlik et al. (2023))—a broad class of models that includes SSMs. Some of these works, namely Emami et al. (2021); Cohen-Karlik et al. (2023) focus specifically on SSMs, in particular Cohen-Karlik et al. (2023) which we extend (by lifting the unrealistic assumption of infinite training data). However, to our knowledge, none of

Table 2: Demonstration of clean-label poisoning of SSMs in a real-world (non-synthetic) setting. Each row in the table summarizes an experiment using an adapted version of the Gradient Matching method for generating cleanly labeled examples that poison a four layer SSM-based S4 neural network trained on the sequential MNIST dataset. Each cleanly labeled poisonous example is generated from a given training example by introducing human-imperceptible noise to the input sequence, while keeping the label intact. The noise introduced for generating each cleanly labeled poisonous example has its last entries relatively large, in line with our theory. The first column in the table specifies the number of test instances preselected for misclassification. The second column indicates the percentage of cleanly labeled poisonous examples, *i.e.*, of training examples subject to poisoning. The third and fourth columns present the number of test instances correctly classified before and after poisoning, respectively. The fifth column reports the size of the last elements of the noise in poisonous examples, quantified by the (Euclidean) norm of the last 3% of the elements in a noise sequence as a fraction of the norm of the entire sequence, averaged across all poisonous examples. In all cases, the S4 neural network achieved training accuracies exceeding 85%, with or without poisonous examples. For implementation details see Section G.2.4.

| # of test instances | % poison | Without poison | With poison | Last size |
|---|---|---|---|---|
| 1 | 10 | 1 out of 1 | 0 out of 1 | 0.898 |
| 1 | 1 | 1 out of 1 | 0 out of 1 | 0.602 |
| 5 | 5 | 5 out of 5 | 0 out of 5 | 0.981 |
| 5 | 1 | 5 out of 5 | 0 out of 5 | 0.841 |
| 10 | 1 | 9 out of 10 | 1 out of 10 | 0.744 |

the prior works on the implicit bias of gradient descent over SSMs of recurrent neural networks have formally established susceptibility to clean-label poisoning, as we do.

Since its demonstration in Shafahi et al. (2018), clean-label poisoning has received significant empirical attention (Huang et al. (2020); Zhu et al. (2019); Aghakhani et al. (2021); Zhao et al. (2020)). It was also studied theoretically for convex models in Suya et al. (2021); Blum et al. (2021).[6] To the best our knowledge, none of the prior works on clean-label poisoning have formally established the phenomenon for SSMs, whose optimization results in a nonconvex objective.

We note that the vast majority of literature (theoretical and empirical) on clean-label poisoning pertains to classification problems, where the discontinuous nature of labels can be leveraged in favor of poisoning (*e.g.*, training examples close to true decision boundaries can be used to distort the learned classifier's decision boundaries). In contrast, our work pertains to regression problems, where, arguably, the continuous nature of labels renders it more challenging to establish clean-label poisoning.

## 6 LIMITATIONS

While this paper provides meaningful contributions to the understanding of the implicit bias of SSMs and of clean-label poisoning, it is important to acknowledge several of its limitations. First, while Theorem 1—our theoretical result establishing susceptibility of SSMs to clean-label poisoning—is extended in Section E, it is still an existence result that applies to specific settings. For example, although it applies to a set of initializations that has positive volume, this volume may be low. Moreover, although it allows the input and output matrices to be learned, their learning rates must be small compared to that of the state transition matrix. Second, both Theorem 1 and our experiments pertain to near-zero initialization, and while such initialization is generally standard for neural networks (Glorot and Bengio (2010); He et al. (2015)), it does not account for modern SSM initializations designed to alleviate vanishing gradients (Gu et al. (2020; 2022)). Third, due to vanishing gradients—which result in long run times—all of our experiments have relatively low dimension for the teacher SSM. Finally, while some of our theory treats SSMs trained as part of non-linear neural networks, these non-linear neural networks do not account for various architectural features present in modern SSM-based neural networks (*e.g.*, multiple SSM layers as in S4 Gu et al. (2021), and

---

[6]Non-convex models were also studied theoretically, for example in Mahloujifar et al. (2019); Mahloujifar and Mahmoody (2018); Gao et al. (2021), but these works considered a different type of poisoning, namely one where training examples are replaced (rather than added).

selectivity as in Mamba Gu and Dao (2023)). Addressing the above limitations is regarded as an important set of directions for future research.

## 7 CONCLUSION

The proliferation of SSMs, particularly in large language models, renders it crucial to understand their implicit bias. In this paper, we revisited prior beliefs by which the implicit bias of SSMs leads to generalization when data is generated by a low dimensional teacher. We formally proved and empirically demonstrated that, in stark contrast to these beliefs, there exist special examples whose inclusion in training completely distorts the implicit bias, to a point where generalization with a low dimensional teacher fails. This failure occurs despite the special training examples being labeled by the teacher, *i.e.* having clean labels!

Our results suggest significant challenges in both the theory and practice of SSMs. On the theoretical front, our results suggest that generalization in SSMs cannot be explained via the traditional view of implicit complexity minimization (Yun et al. (2020); Soudry et al. (2018); Gunasekar et al. (2017)), or through the nascent view by which generalization is typical (Mingard et al. (2021; 2023); Buzaglo et al. (2024)). Indeed, if generalization in SSMs was due to the implicit bias finding a solution which, among all solutions fitting training data, minimizes some (data-independent) complexity measure, then training with additional cleanly labeled examples would not change the solution found, and thus would not disrupt generalization.[7] Moreover, if generalization in SSMs was due to typicality, *i.e.*, to the majority of solutions fitting training data being ones that generalize, then additional cleanly labeled training examples would only improve generalization, as they enhance the dominance of such majority. We believe fundamentally new approaches may be needed in order to theoretically pinpoint the source of generalization in SSMs.

Moving to the practical side, the fact that generalization in SSMs can be disrupted by cleanly labeled training examples—*i.e.*, that SSMs are susceptible to clean-label poisoning—raises significant concerns regarding safety, robustness and reliability. For example, large language models, which are becoming more and more reliant on SSMs (Glorioso et al. (2024); Pióro et al. (2024); Alonso et al. (2024)), are often fine-tuned via supervised learning on public internet data (Le Scao et al. (2023); Touvron et al. (2023); Lhoest et al. (2021)), and in this process, it may be easy for a malicious actor to add cleanly labeled training examples, *e.g.*, by adding unlabeled training examples prior to label generation. We believe significant research efforts should be invested in further delineating the susceptibility of SSMs to clean-label poisoning, and in developing methods for overcoming this susceptibility.

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

## A   ZERO LOSS, NON EXTRAPOLATING STUDENTS

For completeness, we give a proof of Proposition 3 in Cohen-Karlik et al. (2023), which shows that there exist student SSMs which generalize up to any given horizon $k$, but not up to a longer horizon.

**Lemma 1.** *Assume $d > \kappa$, and let $c > 0$ and $q > k$. Then, for any teacher $(A^*, B^*, C^*)$, there exists a $d$ dimensional student $(A, B, C)$ such that its generalization error over sequences of length $k$ equals zero, and yet its generalization error over sequences of length $q$ is $> c$.*

*Proof.* Let $A = \mathrm{Diag}(\lambda_1, \ldots, \lambda_d)$. We first note that for any vector $\mathbf{r} \in \mathbb{R}^d$, the system of equations

$$(CA^i B)_{0 \leq i \leq d-1} = \mathbf{r}$$

can be rewritten as $V^\top \mathbf{g} = \mathbf{r}$, where

$$V = \begin{pmatrix} 1 & \lambda_1 & \lambda_1^2 & \cdots & \lambda_1^{d-1} \\ 1 & \lambda_2 & \lambda_2^2 & \cdots & \lambda_2^{d-1} \\ \vdots & \vdots & \vdots & \ddots & \vdots \\ 1 & \lambda_d & \lambda_d^2 & \cdots & \lambda_d^{d-1} \end{pmatrix}$$

and $\mathbf{g} = (b_1 c_1, \ldots, b_d c_d)^\top$. $V$ is a Vandermonde matrix, and it is well known that it is invertible as long as $\lambda_1, \ldots, \lambda_d$ are all distinct. Therefore for any such $\mathbf{r}$, and fixed, distinct $\lambda_1, \ldots, \lambda_d$, one can solve the equation with $\mathbf{g} = (V^T)^{-1}\mathbf{r}$. To solve $\mathbf{g} = (V^T)^{-1}\mathbf{r}$, one can simply set $B = \mathbf{1}_d, C = \mathbf{g}^T$. To prove the claim, we choose $\mathbf{r}$ such that its first $k$ entries coincide with $(C^*(A^*)^i B^*)_{0 \leq i \leq k-1}$, and its final $d - k$ entries are $> c$. $\qquad\square$

## B   ASSUMPTIONS

For the convenience of the reader, we provide below a concentrated list of all assumptions underlying each of our theoretical results.

**Assumptions underlying Proposition 1:**

- The structure imposed on the teacher and student SSMs is diagonality, *i.e.* their state transition matrices, $A^*$ and $A$, respectively, are constrained to be diagonal.

**Assumptions underlying Theorem 1:**

- The structure imposed on the teacher and student SSMs is diagonality, *i.e.* their state transition matrices, $A^*$ and $A$, respectively, are constrained to be diagonal.

- The teacher and student SSMs are not part of non-linear neural networks, *i.e.* their outputs do not undergo a transformation $\sigma(\cdot, \mathbf{w})$ as described in Section 2.2.

- The teacher SSM has dimension $d^* = 2$ and parameters:

$$A^* = \begin{pmatrix} 1 & 0 \\ 0 & 0 \end{pmatrix} \ , \ \ B^* = \begin{pmatrix} 1 & \sqrt{d-1} \end{pmatrix}^\top \ , \ \ C^* = \begin{pmatrix} 1 & \sqrt{d-1} \end{pmatrix} \ ,$$

  and the training labels $(y^{(i)})_{i=1}^n$ are assigned by the teacher.

- Throughout gradient flow, the input and output matrices of the student SSM (namely, $B(\cdot)$ and $C(\cdot)$, respectively) are fixed at $\mathbf{1}_d$ and $\mathbf{1}_d^\top$, respectively.

## C   PROOF OF PROPOSITION 1

In this section we prove Proposition 1 by deriving equations of motion for each diagonal entry of the state inequality matrix $A$.

*Proof.* Fix $t \geq 0$. We use the following shorthands for simplicity:

$$\widetilde{\phi}(\mathbf{x}^{(i)}) := \phi_{(A(t),B(t),C(t)),\mathbf{w}(t)}(\mathbf{x}^{(i)}), \quad \phi(\mathbf{x}^{(i)}) := \phi_{(A(t),B(t),C(t))}(\mathbf{x}^{(i)})$$

The objective $\tilde{\ell}$ in time $t$ takes the following form:

$$\tilde{\ell}((A(t),B(t),C(t))) = \frac{1}{n} \sum_{i=1}^{n} (y^{(i)} - \widetilde{\phi}(\mathbf{x}^{(i)}))^2$$

Fix $j \in [d]$. Deriving w.r.t $a_j(t)$ and consecutively applying the chain rule we obtain the following

$$\frac{\partial}{\partial a_j(t)} \tilde{\ell}((A(t),B(t),C(t))) = \frac{1}{n} \sum_{i=1}^{n} \frac{\partial}{\partial \widetilde{\phi}(\mathbf{x}^{(i)})} (y^{(i)} - \widetilde{\phi}(\mathbf{x}^{(i)}))^2 \cdot \frac{\partial}{\partial a_j(t)} \widetilde{\phi}(\mathbf{x}^{(i)}) =$$

$$= \frac{2}{n} \sum_{i=1}^{n} \underbrace{(\widetilde{\phi}(\mathbf{x}^{(i)}) - y^{(i)})}_{=-\delta^{(i)}(t)} \frac{\partial}{\partial a_j(t)} \sigma\big(\phi(\mathbf{x}^{(i)}), \mathbf{w}(t)\big) =$$

$$= -\frac{2}{n} \sum_{i=1}^{n} \delta^{(i)}(t) \underbrace{\frac{\partial}{\partial z} \sigma(z, \mathbf{w}(t))|_{z=\phi(\mathbf{x}^{(i)})}}_{=\xi^{(i)}(t)} \frac{\partial}{\partial a_j(t)} \phi(\mathbf{x}^{(i)}) =$$

$$= -\frac{2}{n} \sum_{i=1}^{n} \delta^{(i)}(t) \xi^{(i)}(t) \frac{\partial}{\partial a_j(t)} \left( \sum_{l=1}^{\kappa} C(t)A(t)^{L-l}B(t)x_l^{(i)} \right) = (*)$$

Recalling that $A$ is diagonal, we have that $C(t)A(t)^{\kappa-l}B(t)x_l^{(i)} = \sum_{j'=1}^{d} c_{j'}(t)a_{j'}(t)^{\kappa-l}b_{j'}(t)x_l^{(i)}$. Hence,

$$(*) = -\frac{2}{n} \sum_{i=1}^{n} \delta^{(i)}(t) \xi^{(i)}(t) \frac{\partial}{\partial a_j(t)} \left( \sum_{l=1}^{\kappa} \sum_{j'=1}^{d} c_{j'}(t)a_{j'}(t)^{\kappa-l}b_{j'}(t)x_l^{(i)} \right) =$$

$$= -\frac{2}{n} \sum_{i=1}^{n} \delta^{(i)}(t) \xi^{(i)}(t) \left( \sum_{l=1}^{\kappa} (\kappa-l)c_j(t)a_j(t)^{\kappa-l-1}b_j(t)x_l^{(i)} \right) = (**)$$

Reversing the order of summation and reordering we receive the following:

$$(**) = -c_j(t)b_j(t) \sum_{l=0}^{\kappa-2} a_j(t)^l \left( \sum_{i=1}^{n} \frac{2(l+1)}{n} \cdot \delta^{(i)}(t)\xi^{(i)}(t)x_{\kappa-l-1}^{(i)} \right)$$

The proof concludes by noting that $\dot{a_j}(t) = -\frac{\partial}{\partial a_j(t)} \tilde{\ell}((A(t),B(t),C(t)))$. $\qquad\square$

## D  PROOF OF THEOREM 1

In this section we prove Theorem 1. The outline of the proof is as follows; Section D.1 details the exact theoretical setting we consider. Section D.2 analyzes gradient flow over $\ell$ on a dataset without "poisoned" samples, and we show that it converges to a generalizing solution . Section D.3 analyzes gradient flow after the addition of "poisoned" samples, showing that generalization is degraded. Section D.4 proves that the different initialization sets considered in Section D.2 and Section D.3 intersect, and that one can construct an open set $\mathcal{I} \subseteq \mathbb{R}^d$ such that both phenomena occur. Section D.5 contains auxiliary theorems and lemmas used throughout the proof.

### D.1  SETTING AND ADDITIONAL NOTATIONS

We will slightly change our notation and use $L$ to denote the sequence length, and $k$ as an index. For any $\mathbf{x} \in \mathbb{R}^d$ and any $r \geq 0$ we use $B_r(\mathbf{x})$ to denote

$$B_r(\mathbf{x}) := \{ \mathbf{z} \in \mathbb{R}^d : \|\mathbf{x} - \mathbf{z}\|_2 < r \} \tag{13}$$

and $\overline{B_r}(\mathbf{x})$ to denote

$$\overline{B_r}(\mathbf{x}) := \{\mathbf{z} \in \mathbb{R}^d : \|\mathbf{x} - \mathbf{z}\|_2 \leq r\} \tag{14}$$

For any $\mathbf{x} \in \mathbb{R}^d$ and any $\mathcal{V} \subseteq \mathbb{R}^d$ we define the Euclidean distance between $\mathbf{x}$ and $\mathcal{V}$ as

$$\text{Dist}(\mathbf{x}, \mathcal{V}) := \min_{\mathbf{z} \in \mathcal{V}} \|\mathbf{x} - \mathbf{z}\|_2 \tag{15}$$

We use $\mathcal{W}_1$ and $\mathcal{W}_2$ to respectively denote

$$\mathcal{W}_1 := span\{\mathbf{1}_d\}, \ \mathcal{W}_2 := span\{\mathbf{e_1} - \mathbf{e_2}, \ldots, \mathbf{e_1} - \mathbf{e_d}\} \tag{16}$$

Note that for any $j \in \{2, \ldots, d\}$ it holds that

$$\mathbf{1}_d^\top (\mathbf{e_1} - \mathbf{e_j}) = 1 - 1 = 0$$

Hence $\mathcal{W}_1$ and $\mathcal{W}_2$ are orthogonal. Additionally, it holds that

$$\mathcal{W}_1 \cap \mathcal{W}_2 = \{\mathbf{0}_d\}, \ \dim \mathcal{W}_1 = 1, \ \dim \mathcal{W}_2 = d - 1$$

hence $\mathcal{W}_1 \cup \mathcal{W}_2 = \mathbb{R}^d$. Finally, for any $\psi \geq 0$ we use $\text{Diff}(\psi)$ to denote

$$\text{Diff}(\psi) := \left\{\mathbf{x} \in \mathbb{R}^d : \ \forall i, j \in [d], |x_i - x_j| \leq \psi\right\} \tag{17}$$

and $\text{Diff}(\psi)^{\mathcal{C}}$ to denote

$$\text{Diff}(\psi)^{\mathcal{C}} := \left\{\mathbf{x} \in \mathbb{R}^d : \ \exists i, j \in [d] \ s.t. \ |x_i - x_j| > \psi\right\} \tag{18}$$

Recall that the teacher SSM (Equation (12)) is given by $(A^*, B^*, C^*)$, where

$$A^* = \begin{pmatrix} 1 & 0 \\ 0 & 0 \end{pmatrix} \ , \ B^* = \begin{pmatrix} 1 & \sqrt{d-1} \end{pmatrix}^\top \ , \ C^* = \begin{pmatrix} 1 & \sqrt{d-1} \end{pmatrix} .$$

We claim that the teacher is equivalent, *i.e.* has the same impulse response, as a d-dimensional SSM with $A^d = \text{Diag}(1, 0, ..., 0), B^d = \mathbf{1}_d, C^d = \mathbf{1}_d^\top$.

**Proposition 2.** *For all $i \geq 0$*

$$C^*(A^*)^i B^* = C^d(A^d)^i B^d$$

*Proof.* It is easy to see that both expressions evaluate to $d$ when $i = 0$, and to 1 when $i \geq 1$. □

We will henceforth abuse notation slightly and redefine the teacher $(A^*, B^*, C^*)$ to equal this $d$ dimensional teacher, *i.e.* we set $A^* := A^d, B^* := B^d, C^* := C^d$.

We denote the generalization error on sequences of length $L$ (Definition 1) by $Gen_L(A)$, *i.e.*

$$Gen_L(A) := \max_{L' \in \{0, 1, \ldots, L-1\}} \left| BA^{L'}C - B^*(A^*)^{L'}C^* \right|.$$

note that $B, C$ are kept implicit in this notation, as they are fixed to the values $B = \mathbf{1}_d, C = \mathbf{1}_d^\top$ throughout our analysis.

We will prove a slightly more general claim than the one appearing in the main text (Theorem 1):

**Theorem 2.** *Assume that the training sequence length and the dimension of the student SSM respectively satisfy $L \in \{7, 9, 11, \ldots\}$ and $d \geq 8$. Let $L' \in \mathbb{N}_{\geq L+2}$ and $\epsilon \in \mathbb{R}_{>0}$. Then for any $n < m \in \mathbb{N}$, there exist training sets $\mathcal{S}_1 \subseteq \mathcal{S}_2$ with $|\mathcal{S}_1| = n, |\mathcal{S}_2| = m$ and*

$$\forall i \in [n], \ \|\mathbf{x}^{(i)}\|_\infty = p_i, \quad \forall i \in [m] \setminus [n], \ \mathbf{x}_{L-1}^{(i)} = q_i, \quad \sum_{i \in [n]} p_i^2 = \sum_{i \in [m] \setminus [n]} q_i^2$$

*for some $p_1, \ldots, p_n, q_{n+1}, \ldots, q_m \in \mathbb{R}$, and an open set $\mathcal{I}$ of initializations for the student SSM,[8] such that, with any initialization in $\mathcal{I}$, the following holds:*

---

[8]That is, an open subset of the set of diagonal matrices in $\mathbb{R}^{d,d}$.

- *under $\mathcal{S}_1$ gradient flow converges to a point, which we denote $\widehat{A_1}$, at which the training loss is minimal (i.e., equals zero) and $Gen_{L'}(\widehat{A_1}) \leq \epsilon$.*

- *Under $\mathcal{S}_2$ gradient flow converges to a point, which we denote $\widehat{A_2}$, at which the training loss is minimal and $Gen_{L'}(\widehat{A_2}) \geq \min\{0.1, \frac{1}{9d} \cdot (1 - (0.6)^{\frac{1}{L-1}})\}$.*

One recovers the original statement by taking $m = n + 1$, $\mathcal{S}_1 = \mathcal{S}$, $\mathcal{S}_2 = \mathcal{S} \cup (\mathbf{x}^\dagger, y^\dagger)$, $p_i = 1$ for all $i \in [n]$ and $q_{n+1} = n^{\frac{1}{2}}$.

Examining the teacher weights $(A^*, B^*, C^*)$, one can note that for any $j \in [L-1]$ and any $z \in \mathbb{R}$ it holds that

$$\phi_{(A^*, B^*, C^*)}(z \cdot \mathbf{e_j}) = \sum_{k=1}^{d} c_k^* (a_k^*)^{L-j} b_k^* z = 1 \cdot 1^{L-j} \cdot 1 \cdot z + 0 \cdot z = z$$

**Definition 2.** The datasets $\mathcal{S}_1, \mathcal{S}_2$ are defined as follows:

$$\mathcal{S}_1 := \{(p_i \mathbf{e_1}, \phi_{(A^*, B^*, C^*)}(p_i \mathbf{e_1}))\}_{i=1}^{n} = \{(p_i \mathbf{e_1}, p_i)\}_{i=1}^{n}$$

$$\mathcal{S}_2 := \mathcal{S}_1 \cup \{(q_i \mathbf{e_{L-1}}, \phi_{(A^*, B^*, C^*)}(q_i \mathbf{e_{L-1}}))\}_{i=1}^{m-n} = \mathcal{S}_1 \cup \{(q_i \mathbf{e_{L-1}}, q_i)\}_{i=1}^{m-n}$$

where $\{p_i\}_{i=1}^{n}$ and $\{q_i\}_{i=1}^{m-n}$ are real numbers such that $P := \sum_{i=1}^{n} p_i^2 = \sum_{i=1}^{m-n} q_i^2 > 0$.

The objective $\ell(\cdot; \mathcal{S}_1)$ takes the following form:

$$\ell(A; \mathcal{S}_1) = \frac{1}{n} \sum_{i=1}^{n} (\phi_{(A^*, B^*, C^*)}(p_i \mathbf{e_1}) - \phi_{(A,B,C)}(p_i \mathbf{e_1}))^2 = \tag{19}$$

$$= \frac{1}{n} \sum_{i=1}^{n} (p_i - \sum_{k=1}^{d} a_k^{L-1} p_i)^2 = \frac{1}{n} \sum_{i=1}^{n} p_i^2 (1 - \sum_{k=1}^{d} a_k^{L-1})^2 = \tag{20}$$

$$= \frac{P}{n} (1 - \sum_{k=1}^{d} a_k^{L-1})^2 \tag{21}$$

For any time $t \geq 0$ and any index $j \in [d]$ the gradient flow update $\dot{a}_j(t; \mathcal{S}_1)$ takes the following form

$$\dot{a}_j(t; \mathcal{S}_1) = -\frac{\partial}{\partial a_j(t; \mathcal{S}_1)} \ell(A(t; \mathcal{S}_1); \mathcal{S}_1) = \tag{22}$$

$$= 2(L-1) \frac{P}{n} (1 - \sum_{k=1}^{d} a_k(t; \mathcal{S}_1)^{L-1}) a_j(t; \mathcal{S}_1)^{L-2} \tag{23}$$

The objective $\ell(\cdot; \mathcal{S}_2)$ takes the following form:

$$\ell(A; \mathcal{S}_2) = \frac{1}{m} \sum_{i=1}^{n} (\phi_{(A^*, B^*, C^*)}(p_i \mathbf{e_1}) - \phi_{(A,B,C)}(p_i \mathbf{e_1}))^2 + \tag{24}$$

$$+ \frac{1}{m} \sum_{i=1}^{m-n} (\phi_{(A^*, B^*, C^*)}(q_i \mathbf{e_{L-1}}) - \phi_{(A,B,C)}(q_i \mathbf{e_{L-1}}))^2 = \tag{25}$$

$$= \frac{1}{m} \sum_{i=1}^{n} (p_i - \sum_{k=1}^{d} a_k^{L-1} p_i)^2 + \frac{1}{m} \sum_{i=1}^{m-n} (q_i - \sum_{k=1}^{d} a_k q_i)^2 = \tag{26}$$

$$= \frac{P}{m} \left( (1 - \sum_{k=1}^{d} a_k^{L-1})^2 + (1 - \sum_{k=1}^{d} a_k)^2 \right) \tag{27}$$

For any time $t \geq 0$ and any index $j \in [d]$ the gradient flow update $\dot{a}_j(t; \mathcal{S}_2)$ takes the following form

$$\dot{a}_j(t; \mathcal{S}_2) = -\frac{\partial}{\partial a_j(t; \mathcal{S}_2)} \ell(A(t; \mathcal{S}_2); \mathcal{S}_2) = \tag{28}$$

$$= \frac{2P}{m} \left( (L-1)(1 - \sum_{k=1}^{d} a_k(t; \mathcal{S}_2)^{L-1}) a_j(t; \mathcal{S}_2)^{L-2} + (1 - \sum_{k=1}^{d} a_k(t; \mathcal{S}_2)) \right) \tag{29}$$

Note that by Lemma 24 the above flows are defined for all $t \geq 0$. We denote by $\mathcal{I}_0$ a set of initial values for the matrix $A$ which we will use throughout the proof:[9]

$$\mathcal{I}_0 := \left\{ \alpha \cdot (\zeta_1, \ldots, \zeta_d)^\top \in \mathbb{R}^d : \alpha \in (0, \frac{1}{2d}), 1 = \zeta_1 > \zeta_2 > \cdots > \zeta_d > 0 \right\} \tag{30}$$

Throughout Section D.2 and Section D.3 we will be concerned with subsets of $\mathcal{I}_0$ for which the respective claims hold.

## D.2 Gradient flow under $\mathcal{S}_1$ generalizes

Throughout this part, we omit the dependence on $\mathcal{S}_1$ for simplicity. We begin by proving that when initializing at some $A(0) \in \mathcal{I}_0$, the parameters of $A$ converge to a point where the training loss equals zero:

**Lemma 2.** *Suppose we initialize at $A(0) \in \mathcal{I}_0$ and evolve $A(t)$ according to the gradient flow dynamics in Equation (19). Then the limit $\lim_{t \to \infty} A(t) =: \widehat{A_1}$ exists and satisfies*

$$\ell(\widehat{A_1}) = 0$$

*Proof.* We first prove that for any $j \in [d]$ and for any time $t \geq 0$ it holds that

$$\alpha \zeta_j \leq a_j(t) \leq 1$$

Recall that

$$\dot{a}_j(t) = 2(L-1)\frac{P}{n}(1 - \sum_{k=1}^{d} a_k(t)^{L-1})a_j(t)^{L-2}$$

Hence, by Equation (30) it must hold that $\dot{a}_j(t) \geq 0$ for any $t \geq 0$ - It holds that $\alpha \zeta_j > 0$ and since $L - 1$ is even we have

$$1 - \sum_{k=1}^{d}(\alpha \zeta_k)^{L-1} \geq 1 - d \cdot (\alpha \zeta_1)^{L-1} \geq 1 - d(\frac{1}{2d})^{L-1} > 0$$

Hence at time $t = 0$ we have $\dot{a}_j(0) > 0$. For any $t > 0$, if the derivative equals zero then either $1 - \sum_{k=1}^{d} a_k(t)^{L-1} = 0$ or $a_j(t) = 0$, implying the derivative must remain equal to zero for $t' > t$. Hence, $\alpha \zeta_j \leq a_j(t)$ for any $t \geq 0$. Additionally, for any time $t \geq 0$ it holds that $1 - \sum_{k=1}^{d} a_k(t)^{L-1} \geq 0$ - at initialization it is positive by the above, and again if at some point it is equals zero then it must remain zero thereafter. Therefore, $a_j(t)$ can never reach 1 - since $L - 1$ is even and since all entries are strictly positive, if it were to reach or cross 1 we would reach a contradiction to the previous argument. Thus, we have showed that the gradient flow trajectory is contained in the following open and bounded set:

$$\mathcal{V} = B_d(\mathbf{0}_d) \setminus \overline{B_{\frac{\alpha \zeta_d}{2}}(\mathbf{0}_d)}$$

Note that the teacher $A^*$ is within $\mathcal{V}$. Next, we claim that within $\mathcal{V}$ the objective $\ell$ satisfies the PL condition (see Definition 13) with PL coefficient $\frac{1}{n} \cdot 2(L-1)^2 P(\frac{\alpha \zeta_d}{2\sqrt{d}})^{2L-4}$ - indeed, for any $A \in \mathcal{V}$ and any $j \in [d]$ it holds that

$$\frac{\partial}{\partial a_j}\ell(A) = 2(L-1)\frac{P}{n}(1 - \sum_{k=1}^{d} a_k^{L-1})a_j^{L-2}$$

For any $A \in \mathcal{V}$ there must exist an index $j^* \in [d]$ for which $|a_{j^*}| \geq \frac{\alpha \zeta_d}{2\sqrt{d}}$ and thus

$$\|\nabla \ell(A)\|_2^2 \geq \left(2(L-1)\frac{P}{n}(1 - \sum_{k=1}^{d} a_k^{L-1})a_{j^*}^{L-2}\right)^2 = \frac{4(L-1)^2 P a_{j^*}^{2L-4}}{n}\ell(A) \geq$$

$$\geq 2 \cdot \frac{2(L-1)^2 P(\frac{\alpha \zeta_d}{2\sqrt{d}})^{2L-4}}{n}\ell(A)$$

---

[9] $A$ is a diagonal matrix , so we treat $\mathcal{I}_0$ as a subset of $\mathbb{R}^d$.

Finally, there exists some constant $M > 0$ such that within $\mathcal{V}$ the objective $\ell$ has $M$-Lipschitz gradients, since $\ell$ is analytic in $\mathbb{R}^d$ and since $\mathcal{V}$ is contained within the compact and bounded $\overline{B}_d(\mathbf{0}_d)$. The above conditions allows us to invoke Lemma 26 which states that the limit $\lim_{t \to \infty} A(t) =: \widehat{A}_1$ exists and satisfies $\ell(A(t)) = 0$ as required. $\qquad\square$

We now introduce a set $\mathcal{I}_1 \subseteq \mathcal{I}_0$, under which we prove the rest of the claims in this section:

**Definition 3.** Let $\eta > 0$. We use $\mathcal{I}_1(\eta_1)$ to denote the following subset of $\mathcal{I}_0$:

$$\mathcal{I}_1(\eta) := \left\{ A \in \mathcal{I}_0 : \forall j \in \{2, \dots, d\}. \ \alpha \leq \left( \frac{1 - (1 - \eta)^{L-1} - \eta \sqrt{\frac{n}{P}}}{d - 1} \right)^{\frac{1}{L-1}} \frac{1}{\zeta_j} (1 - \zeta_j^{L-3})^{\frac{1}{L-3}} \right\}$$

We now prove that if $A(0) \in \mathcal{I}_1$, the first diagonal entry tends to 1, while the rest of the entries must remain close to 0:

**Proposition 3.** *Let $\eta_1 > 0$. Suppose we initialize at $A(0) \in \mathcal{I}_1(\eta_1)$ and evolve $A(t)$ according to the gradient flow dynamics in Equation (19). For any $j \in \{2, \dots, d\}$ and for any time $t \geq 0$ it holds that:*

$$0 \leq a_j(t) \leq \left( \frac{1 - (1 - \eta_1)^{L-1} - \eta_1 \sqrt{\frac{n}{P}}}{d - 1} \right)^{\frac{1}{L-1}}$$

*Additionally, there exists some time $t^* \geq 0$ such that for any time $t \geq t^*$ it holds that:*

$$1 - \eta_1 \leq a_1(t) \leq 1$$

*Proof.* Per the proof of Lemma 2, $\dot{a}_j(t) \geq 0$ for any $j \in [d]$ and $t \geq 0$ and thus the entries $a_j(t)$ are positive and non-decreasing (as functions of $t$). Reordering the dynamics, we have the following for any $j \in \{2, \dots, d\}$ and for any time $\tau \geq 0$:

$$\dot{a}_j(\tau) a_j(\tau)^{-L+2} = \frac{\dot{a}_j(\tau)}{a_j(\tau)^{L-2}} = 2(L-1)\frac{P}{n}\left(1 - \sum_{k=1}^{d} a_k(\tau)^{L-1}\right) = \frac{\dot{a}_1(\tau)}{a_1(\tau)^{L-2}} = \dot{a}_1(\tau) a_1(\tau)^{-L+2}$$

Integrating both sides w.r.t time, we receive the following for any time $t \geq 0$:

$$\frac{a_j(t)^{-L+3}}{-L+3} - \frac{a_j(0)^{-L+3}}{-L+3} = \int_0^t \dot{a}_j(\tau) a_j(\tau)^{-L+2} d\tau =$$

$$= \int_0^t \dot{a}_1(\tau) a_1(\tau)^{-L+2} d\tau = \frac{a_1(t)^{-L+3}}{-L+3} - \frac{a_1(0)^{-L+3}}{-L+3}$$

Organizing the equation and plugging the initial values, we get that

$$a_j(t)^{-L+3} = a_1(t)^{-L+3} + (\alpha \zeta_j)^{-L+3} - \alpha^{-L+3}$$

Both sides are positive by our first argument and since $\alpha \zeta_j < \alpha$, and so taking the $\frac{1}{L-3}$ root yields

$$a_j(t) = \left( \frac{1}{a_1(t)^{-L+3} + (\alpha \zeta_j)^{-L+3} - \alpha^{-L+3}} \right)^{\frac{1}{L-3}} \leq \left( \frac{1}{\frac{1}{(\alpha \zeta_j)^{L-3}} - \frac{1}{\alpha^{L-3}}} \right)^{\frac{1}{L-3}} =$$

$$= \left( \frac{(\alpha \zeta_j)^{L-3}}{1 - \zeta_j^{L-3}} \right)^{\frac{1}{L-3}} = \alpha \zeta_j \left( \frac{1}{1 - \zeta_j^{L-3}} \right)^{\frac{1}{L-3}} = (*)$$

Since $A(0) \in \mathcal{I}_1(\eta_1)$, we obtain that

$$a_j(t) \leq \left( \frac{1 - (1 - \eta_1)^{L-1} - \eta_1 \sqrt{\frac{n}{P}}}{d - 1} \right)^{\frac{1}{L-1}} \frac{1}{\zeta_j} (1 - \zeta_j^{L-3})^{\frac{1}{L-3}} \zeta_j \left( \frac{1}{1 - \zeta_j^{L-3}} \right)^{\frac{1}{L-3}} =$$

$$= \left( \frac{1 - (1 - \eta_1)^{L-1} - \eta_1 \sqrt{\frac{n}{P}}}{d - 1} \right)^{\frac{1}{L-1}}$$

as desired. We Now show that there exists $t^* \geq 0$ such that for any time $t \geq t^*$ it holds that

$$a_1(t) \geq 1 - \eta_1$$

By Lemma 2, there exists time $t^* \geq 0$ such that for any $t \geq t^*$ it holds that

$$\ell(A(t)) = \frac{P}{n}(1 - \sum_{k=1}^{d} a_k(t)^{L-1})^2 \leq \eta_1^2$$

Therefore, for any time $t \geq t^*$ we have

$$|1 - \sum_{k=1}^{d} a_k(t)^{L-1}| \leq \eta_1 \sqrt{\frac{n}{P}} \implies 1 - \eta_1 \sqrt{\frac{n}{P}} \leq \sum_{k=1}^{d} a_k(t)^{L-1} \leq 1 + \eta_1 \sqrt{\frac{n}{P}}$$

Focusing on the left hand side and plugging the bound on the rest of the entries, we receive

$$1 - \eta_1 \sqrt{\frac{n}{P}} \leq a_1(t)^{L-1} + (d-1) \cdot \frac{1 - (1 - \eta_1)^{L-1} - \eta_1 \sqrt{\frac{n}{P}}}{d-1} =$$

$$= a_1(t)^{L-1} + 1 - (1 - \eta_1)^{L-1} - \eta_1 \sqrt{\frac{n}{P}}$$

Rearranging yields

$$(1 - \eta_1)^{L-1} \leq a_1(t)^{L-1} \implies 1 - \eta_1 \leq a_1(t)$$

Additionally, $a_1(t)$ can never cross 1 - since $L - 1$ is even and since all entries are strictly positive, if it were to cross 1 we would reach a contradiction to the argument in Lemma 2 stating that the residual $1 - \sum_{k=1}^{d} a_k(t)^{L-1}$ is always non-negative. With this we complete our proof. $\qquad\square$

An immediate result from Proposition 3 is the following corollary regarding the student's recovery of the teacher:

**Corollary 1.** *Let $\eta_1 > 0$. Suppose we initialize at $A(0) \in \mathcal{I}_1(\eta_1)$ and evolve $A(t)$ according to the gradient flow dynamics in Equation (19). The limit $\lim_{t \to \infty} A(t) =: \widehat{A_1}$ satisfies*

$$\|\widehat{A_1} - A^*\|_2 \leq \sqrt{\eta_1^2 + (d-1)(\frac{1 - (1 - \eta_1)^{L-1} - \eta_1 \sqrt{\frac{n}{P}}}{d-1})^{\frac{2}{L-1}}}$$

*Proof.* By Proposition 3, there exists time $t^* \geq 0$ such that for any time $t \geq t^*$ it holds that

$$\|A(t) - A^*\|_2 = \sqrt{(1 - a_1(t))^2 + \sum_{k=2}^{d}(0 - a_k(t))^2} \leq$$

$$\leq \sqrt{\eta_1^2 + (d-1)(\frac{1 - (1 - \eta_1)^{L-1} - \eta_1 \sqrt{\frac{n}{P}}}{d-1})^{\frac{2}{L-1}}}$$

The argument follows from Lemma 2 and from continuity. $\qquad\square$

**Remark 1.** *Note that the upper bound in corollary 1 satisfies the following*

$$\lim_{\eta_1 \to 0} \sqrt{\eta_1^2 + (d-1)(\frac{1 - (1 - \eta_1)^{L-1} - \eta_1 \sqrt{\frac{n}{P}}}{d-1})^{\frac{2}{L-1}}} =$$

$$= \sqrt{\lim_{\eta_1 \to 0} \eta_1^2 + (d-1)(\frac{1 - (1 - \eta_1)^{L-1} - \eta_1 \sqrt{\frac{n}{P}}}{d-1})^{\frac{2}{L-1}}} = \sqrt{0} = 0$$

*Hence, for any recovery threshold $\delta > 0$ there exists $\eta_{1,\delta} > 0$ such that if $A(0) \in \mathcal{I}_1(\eta_{1,\delta})$ then $\widehat{A_1}$ recovers $A^*$ with an error of no more than $\delta$.*

So far, we have argued that the parameters of A converge to a point which is close $A^*$. We conclude by showing that this leads to low generalization error.

**Proposition 4.** *Let $L' \geq L+2$. For any $\epsilon > 0$ there exists an open set of initializations $\mathcal{I}_1 := \mathcal{I}_1(\delta_\epsilon)$ such that under $\mathcal{S}_1$, A converges to a point such that $Gen_{L'}(A) \leq \epsilon$.*

*Proof.* Under the dataset $\mathcal{S}_1$, we have shown above that for any $\delta > 0$ there exists an open set of initializations $\mathcal{I}_1(\delta)$ such that GF will converge to a solution a whose parameters satisfy $\|A - A^*\|^2 \leq \delta$. It follows from the continuity of the length $L'$ impulse response that there is an open set of initializations from which we converge to a point $Gen_{L'}(A) \leq \epsilon$. $\qquad\square$

We abuse notation slightly and denote $\mathcal{I}_1(\epsilon) := \mathcal{I}_1(\eta_{1,\delta_1})$ where $\delta_1$ is the maximal $\delta$ that guarantees $Gen_{L'}(\widehat{A_1}) \leq \epsilon$.

### D.3 Gradient flow over $\mathcal{S}_2$ converges but doesn't generalize

In this section we show that one can find a set of initialization $\mathcal{I}_2$ such that gradient flow under $\mathcal{S}_2$ converges to a point with high generalization error. The proof shows that gradient flow trajectories initialized in $\mathcal{I}_2$ evolve similarly to reference trajectories which provably stays away from any permutation of $A^*$.[10] Since the training loss is non-convex, gradient flow trajectories can diverge from one another exponentially fast. Establishing that a reference trajectory is tracked thus requires sharp bounds on convergence times. The proof in this section is rather involved and is thus split into several parts;

- D.3.1 defines the reference trajectories and shows their poor ability of generalization.

- D.3.2 characterizes the critical points of the objective $\ell$, focusing on a specific saddle point of interest (which we denote $\mathbf{s}$).

- D.3.3 presents relevant background on dynamical systems, introducing a linearization result needed for the rest of the proof.

- In D.3.4 we start analyzing the trajectories itself, showing that it must pass near $\mathbf{s}$.

- D.3.5 shows that the trajectories must escape sufficiently fast from $\mathbf{s}$ using the tools presented in D.3.3.

- D.3.6 proves that after escaping from $\mathbf{s}$ the trajectories converge to global minima.

- D.3.7 shows that the overall divergence between trajectories emanating from $\mathcal{I}_2$ and their corresponding reference trajectories can be bounded from above, implying the former trajectories have poor generalization.

Throughout this part, we omit the dependence on $\mathcal{S}_2$ for simplicity.

### D.3.1 Reference trajectories

We begin our proof by proving the following useful lemma which states that gradient flows maintains the order of the entries of $A$:

**Lemma 3.** *Suppose we initialize at $A(0) \in \mathbb{R}^d$ and evolve $A(t)$ according to Equation* (28). *Let $\pi : [d] \to [d]$ be a permutation such that for any $j \in [d-1]$:*

$$a_{\pi(j)}(0) \geq a_{\pi(j+1)}(0)$$

*Then for any $j \in [d-1]$ and any $t \geq 0$ it holds that*

$$a_{\pi(j)}(t) \geq a_{\pi(j+1)}(t)$$

*Proof.* Recall the dynamics from Equation (28):

$$\dot{a_j}(t) = \frac{2P}{m}\left((L-1)(1 - \sum_{k=1}^{d} a_k(t)^{L-1})a_j(t)^{L-2} + (1 - \sum_{k=1}^{d} a_k(t))\right)$$

---

[10]Any permutation of $A^*$ yields a system with the same impulse response.

Fix $j \in [d-1]$. By the linearity of the derivative, we obtain the following equality by plugging the above dynamics

$$\frac{d}{dt}\left(a_{\pi(j)}(t) - a_{\pi(j+1)}(t)\right) = \dot{a}_{\pi(j)}(t) - \dot{a}_{\pi(j+1)}(t) =$$

$$= \frac{2(L-1)P}{m}(1 - \sum_{k=1}^{d} a_k(t)^{L-1})(a_{\pi(j)}(t)^{L-2} - a_{\pi(j+1)}(t)^{L-2})$$

Assume on the contrary there exists some time $t_1 \geq 0$ for which $a_{\pi(j)}(t_1) < a_{\pi(j+1)}(t_1)$. By the assumption, $t_1 > 0$. By continuity, there must exist some time $t_2 \in [0, t_1)$ for which $a_{\pi(j)}(t_2) = a_{\pi(j+1)}(t_2)$. This would imply that for any $t \geq t_2$, the derivative $\frac{d}{dt}\left(a_{\pi(j)}(t) - a_{\pi(j+1)}(t)\right)$ is equal zero, which in turn would imply that

$$a_{\pi(j)}(t) - a_{\pi(j+1)}(t) = a_{\pi(j)}(t_2) - a_{\pi(j+1)}(t_2) = 0$$

in contradiction to the assumption on $t_1$. $\qquad\square$

In what follows, we define the notion of *reference initialization*:

**Definition 4.** Let $A \in \mathcal{I}_0$ be some initialization of the parameters. The corresponding *reference initialization* $A^{ref}$ is defined as

$$\forall j \in [d].\ a_j^{ref} = \begin{cases} a_1, & j = 1, 2 \\ a_j, & \text{otherwise} \end{cases}$$

We use $A^{ref}(t)$ to denote the gradient flow trajectories emanating from the reference initializations.

We now prove that any point with zero training loss which is sufficiently close to a reference trajecory has poor generalization.

**Lemma 4.** *Let $L' \geq L + 2$. There exists some $\delta_2 > 0$ such that any point $A = (a_1, a_2, ...a_d) \in \mathbb{R}^d$ which satisfies:*

- $\ell(A) = 0$

- $a_1 \geq a_2 \geq ... \geq a_d$

- $\|A - A^{eq}\| \leq \delta_2$ *for some point $A^{eq} = (a_1^{eq}, \ldots, a_d^{eq}) \in \mathbb{R}^d$ such that $a_1^{eq} = a_2^{eq}$.*

*must satisfy $Gen_{L'}(A) \geq \min\{0.1, \frac{1}{9d} \cdot (1 - (0.6)^{\frac{1}{L-1}})\}$.*

*Proof.* Let $L^* \in \{L+1, \ldots, L'\}$ such that $L^*$ is even. We now show that

$$\sum_{k=1}^{d} a_k^{L^*-1} \leq 1 - c$$

for some constant $c > 0$ which is independent of $L'$. This in turn implies that

$$Gen_{L'}(A) \geq (1 - CA^{L^*-1}B) \geq c$$

which gives us the desired lower bound. To do this, we write

$$\sum_k^d a_k^{L^*-1} = \sum_k^d a_k^{L-1}a_k^{L^*-L}$$

First note that $|a_k| \leq 1$ for all $k \in [d]$ - this follows from the fact that $L-1$ is even and from the fact that $\ell(A) = 0$ and hence $\sum_k a_k^{L-1} = 1$. Therefore, for all $k \in [d]$ we have

$$|a_k^{L^*-1}| = |a_k^{L-1}a_k^{L^*-L}| = |a_k^{L-1}| \cdot |a_k^{L^*-L}| \leq a_k^{L-1}$$

Assume first that $a_1 = a_2 = a$. Then clearly $a_1^{L-1} + a_2^{L-1} = 2a^{L-1} \leq 1$ and hence

$$a_1^{L-1}, a_2^{L-1} \leq \frac{1}{2} \implies a_1, a_2 \leq (\frac{1}{2})^{\frac{1}{L-1}}$$

Now by continuity it follows that for sufficiently small $\delta_2 > 0$, we have that if $\|A - A^{eq}\| \leq \delta_2$ then

$$a_1, a_2 \leq (0.6)^{\frac{1}{L-1}}$$

Let $J := \{r : a_r \leq 0\}$. For such indices we have $a_r^{L^*-1} \leq 0$. Suppose that

$$\sum_{k \in J} a_k^{L-1} \geq 0.1$$

Then we have that

$$\sum_{k=1}^{d} a_k^{L^*-1} \leq \sum_{k \notin J} a_k^{L^*-1} = \sum_{k \notin J} a_k^{L-1} a_k^{L^*-L} \leq \sum_{k \notin J} a_k^{L-1} = 1 - \sum_{k \notin J} a_k^{L-1} \leq 0.9$$

so we can take $c = 0.1$. Otherwise we have that

$$\sum_{k \notin J} a_k^{L-1} \geq 0.9$$

so there exists some $k^* \notin J$ such that $a_{k^*}^{L-1} \geq \frac{1}{9d}$. On the other hand, we have

$$a_{k^*} \leq a_1 \leq |a_1| \leq (0.6)^{\frac{1}{L-1}}$$

Therefore, since $k^* \notin J$ we have $0 \leq a_{k^*}^{L^*-L} \leq a_{k^*}$ and so

$$a_{k^*}^{L-1} - a_{k^*}^{L^*-1} = a_{k^*}^{L-1}(1 - a_{k^*}^{L^*-L}) \geq \frac{1}{9d}(1 - a_{k^*}) \geq \frac{1}{9d}(1 - (0.6)^{\frac{1}{L-1}})$$

This yields the following:

$$1 - \sum_{k=1}^{d} a_k^{L^*-1} = \sum_{k=1}^{d} a_k^{L-1} - \sum_{k=1}^{d} a_k^{L^*-1} = \sum_{k=1}^{d}(a_k^{L-1} - a_k^{L^*-1}) \geq \frac{1}{9d}(1 - ((0.6)^{\frac{1}{L-1}})$$

which gives us $c = \frac{1}{9d}(1 - (0.6)^{\frac{1}{L-1}})$. In either case we can find a constant $c > 0$ proving the argument. □

Lemma 4 motivates us to find an open subset of initializations under which the respective gradient flow trajectories remain close to their reference trajectory counterparts, as this would allow us to lower bound generalization error.

### D.3.2 CHARACTERIZATION OF CRITICAL POINTS

In this section we characterize the critical points of the objective $\ell$.

**Lemma 5.** *Let $A \in \mathbb{R}^d$ be a point such that*

$$\nabla \ell(A) = 0$$

*Then either $A$ is a global minimum, i.e. $\ell(A) = 0$, or exists $s \in \mathbb{R}$ such that $A = s \cdot \mathbf{1}_d$.*

*Proof.* By Equation (28), for any $j \in [d]$ it holds that

$$\frac{\partial}{\partial a_j} \ell(A) = \frac{2P}{m}\left((L-1)(\sum_{k=1}^{d} a_k^{L-1} - 1)a_j^{L-2} + (\sum_{k=1}^{d} a_k - 1)\right) = 0$$

If $\sum_{k=1}^{d} a_k^{L-1} - 1 = 0$, then the above simplifies to

$$\frac{2P}{m}(\sum_{k=1}^{d} a_k - 1) = 0$$

which implies by our assumption on $P$ being positive that $\sum_{k=1}^{d} a_k - 1 = 0$. This in turn yields that

$$\ell(A) = \frac{P}{m}\left((1 - \sum_{k=1}^{d} a_k^{L-1})^2 + (1 - \sum_{k=1}^{d} a_k)^2\right) = 0$$

*i.e.* $A$ is a global minimum. Suppose $\sum_{k=1}^{d} a_k^{L-1} - 1 \neq 0$. Then we obtain by rearranging that

$$a_j^{L-2} = \frac{(1 - \sum_{k=1}^{d} a_k)}{(L-1)(\sum_{k=1}^{d} a_k^{L-1} - 1)}$$

$L - 2$ is odd, and so taking the L-2 root on both sides we obtain that

$$a_j = \left(\frac{(1 - \sum_{k=1}^{d} a_k)}{(L-1)(\sum_{k=1}^{d} a_k^{L-1} - 1)}\right)^{\frac{1}{L-2}} := s$$

completing our proof. $\qquad\square$

Lemma 5 establishes that critical points of $\ell$ which are not global minima must reside within $\mathcal{W}_1 = span\{\mathbf{1}_d\}$ (Equation (16)). These saddle points pose an obstacle to the convergence of gradient flow to a global minimum. The following lemma outlines the type of points gradient flow could ever encounter assuming we initialize at $\mathcal{I}_0$ or at a reference initialization:

**Lemma 6.** *Suppose we initialize at $A(0) \in \mathcal{I}_0$ and at $A^{ref}(0)$, and evolve $A(t)$ and $A^{ref}(t)$ according to Equation (28). Then for any time $t \geq 0$ it holds that*

$$\ell(A(t)), \ell(A^{ref}(t)) \leq \frac{2P}{m}$$

*Proof.* Per Equation (30) the entries at initialization are arranged in descending order. Since $L - 1$ is even we have that the initializations satisfy the inequalities

$$1 - \sum_{k=1}^{d} a_k(0)^{L-1} = 1 - \sum_{k=1}^{d}(\alpha\zeta_k)^{L-1} \geq 1 - 2(\alpha\zeta_1)^{L-1} - \sum_{k=3}^{d}(\alpha\zeta_k)^{L-1} = 1 - \sum_{k=1}^{d} a_k^{ref}(0)^{L-1}$$

and

$$1 - \sum_{k=1}^{d} a_k(0) = 1 - \sum_{k=1}^{d}(\alpha\zeta_k) \geq 1 - 2(\alpha\zeta_1) - \sum_{k=3}^{d}(\alpha\zeta_k) = 1 - \sum_{k=1}^{d} a_k^{ref}(0)$$

By Equation (30) it holds that $\alpha\zeta_1 < \frac{1}{2d}$, thus we have that

$$1 - 2(\alpha\zeta_1)^{L-1} - \sum_{k=3}^{d}(\alpha\zeta_k)^{L-1} \geq 1 - d \cdot (\alpha\zeta_1)^{L-1} \geq 1 - d(\frac{1}{2d})^{L-1} > 0$$

and

$$1 - 2(\alpha\zeta_1) - \sum_{k=3}^{d}(\alpha\zeta_k) \geq 1 - d \cdot (\alpha\zeta_1) \geq 1 - d(\frac{1}{2d}) > 0$$

On the other hand, by Equation (30) it also holds that $\alpha\zeta_d > 0$, thus we have that

$$1 - \sum_{k=1}^{d}(\alpha\zeta_k)^{L-1} \leq 1 - d \cdot (\alpha\zeta_d)^{L-1} < 1$$

and

$$1 - \sum_{k=1}^{d}(\alpha\zeta_k) \leq 1 - d \cdot (\alpha\zeta_d) < 1$$

Therefore, both the original initialization and the reference initalization satisfy

$$1 - \sum_{k=1}^{d} a_k(0)^{L-1}, 1 - \sum_{k=1}^{d} a_k^{ref}(0)^{L-1} \in (0,1)$$

and

$$1 - \sum_{k=1}^{d} a_k(0), 1 - \sum_{k=1}^{d} a_k^{ref}(0) \in (0,1)$$

Thus, the objective at both initializations is no more than $\frac{2P}{m}$ since both satisfy

$$\ell(A) = \frac{P}{m}\left( (1 - \sum_{k=1}^{d} a_k^{L-1})^2 + (1 - \sum_{k=1}^{d} a_k)^2 \right) \leq \frac{P}{m}\left(1^2 + 1^2\right) \leq \frac{2P}{m}$$

The proof is completed by the argument in Lemma 23 which states that under gradient flow the objective is non-increasing. □

The following lemma shows that only a specific region of $\mathcal{W}_1$ potentially contains critical points with loss lower than that of the initialization points we consider, implying by Lemma 23 that only a specific region of $\mathcal{W}_1$ is relevant:

**Lemma 7.** *Let $A \in \mathbb{R}^d$ be a point for which there exists $a \in \mathbb{R}$ such that*

$$A = a \cdot \mathbf{1}_d$$

*If $a \notin [\frac{1}{d}, \frac{3}{d}]$ then either $\nabla \ell(A) \neq 0$ or $\ell(A) > \frac{2P}{m}$*

*Proof.* We begin by proving that for any $a \in \mathbb{R}$, if $a \notin (0, \frac{3}{d}]$ then $a \cdot \mathbf{1}_d$ must incur a loss greater than $\frac{2P}{m}$. If $a > \frac{3}{d}$ then it holds that $d \cdot a > 3$, hence we obtain that

$$\ell(a \cdot \mathbf{1}_d) = \frac{P}{m}\left( (1 - d \cdot a^{L-1})^2 + (1 - d \cdot a)^2 \right) \geq \frac{P(1 - d \cdot a)^2}{m} > \frac{2P}{m}$$

The same argument applies when $a < -\frac{1}{d}$, since in that case $d \cdot a < -1 \implies (1 - d \cdot a)^2 > 2$. Next, we show that if $a \in [-\frac{1}{d}, \frac{1}{d})$ then $\nabla \ell(a \cdot \mathbf{1}_d) \neq 0$. Suppose $a \in [-\frac{1}{d}, 0]$. $L-1$ is even and $d \geq 8$ hence

$$d \cdot a^{L-1} - 1 \in [-1, 0) \implies (L-1)(d \cdot a^{L-1} - 1)a^{L-2} \in (0, \frac{L-1}{d^{L-2}}) \subseteq (0, \frac{L-1}{8^{L-2}})$$

The function $f(L) := \frac{L-1}{8^{L-2}}$ is decreasing for $L \geq 3$ and acheives the value $0.25$ when $L = 3$, hence since $L \geq 3$ we get $f(L) \leq 0.25$. Thus we have for any $j \in [d]$ that the gradient's $j$th entry statisfies

$$\nabla \ell(a \cdot \mathbf{1}_d) = \frac{2P}{m}\left( (L-1)(d \cdot a^{L-1})a^{L-2} + (d \cdot a - 1) \right) \leq \frac{2P}{m}\left(0.25 + (d \cdot a - 1)\right) \leq$$

$$\leq \frac{2P}{m}\left(0.25 - 1\right) < 0$$

Suppose $a \in (0, \frac{1}{d})$. In this case, we have that

$$a^{L-1} < \frac{1}{d} \implies d \cdot a^{L-1} < 1 \implies (L-1)(d \cdot a^{L-1} - 1)a^{L-2} < 0$$

Hence, since $d \cdot a - 1 < 0$ we have for any $j \in [d]$ that the gradient's $j$th entry satisfies

$$\nabla \ell(a \cdot \mathbf{1}_d)_j = \frac{2P}{m}\left( (L-1)(d \cdot a^{L-1} - 1)a^{L-2} + (d \cdot a - 1) \right) < 0$$

Therefore, any critical point which is not a global minimum and has value in $(0, \frac{2P}{m})$ cannot reside outside of $[\frac{1}{d}, \frac{3}{d}]$. □

Having disqualified most of $\mathcal{W}_1$, we now identify the unique critical point on the non-disqualified region of $\mathcal{W}_1$ and show that it is not a global minimum:

**Lemma 8.** *There exists a unique $s \in [\frac{1}{d}, \frac{3}{d}]$ for which $\nabla \ell(s \cdot \mathbf{1}_d) = 0$. Additionally, $s$ satisfies*

$$\mathbf{s} := s \cdot \mathbf{1}_d = \operatorname*{argmin}_{A \in \mathcal{W}_1} \ell(A)$$

*and*

$$\ell(\mathbf{s}) \geq \frac{P}{4(m)} > 0$$

*Proof.* We focus on the following function:

$$f(a) = \frac{P}{m}\left((1 - d \cdot a^{L-1})^2 + (1 - d \cdot a)^2\right)$$

Note that $f(a) = \ell(a \cdot \mathbf{1}_d)$. It holds that

$$f^{'}(a) := \frac{2P \cdot d}{m}\left((L-1)(d \cdot a^{L-1} - 1)a^{L-2} + (d \cdot a - 1)\right)$$

Note that $f^{'}(a) = \nabla \ell(a \cdot \mathbf{1}_d)_j$ for any $j \in [d]$, and so $f^{'}(a) = 0$ if and only if $\nabla \ell(a \cdot \mathbf{1}_d) = \mathbf{0}_d$. We proceed to show that within $[-\frac{1}{d}, \frac{3}{d}]$, $f'(a)$ has a root and is monotonic. It holds that

$$f^{'}(0) = \frac{2P \cdot d}{m}\left((L-1)(d \cdot (0)^{L-1} - 1)(0)^{L-2} + (d \cdot 0 - 1)\right) = -\frac{2P \cdot d}{m} < 0$$

Next, since $d \geq 8$ it holds that

$$(L-1)(1 - d \cdot (\frac{3}{d})^{L-1})(\frac{3}{d})^{L-2} \leq (L-1)(\frac{3}{d})^{L-2} \leq \frac{L-1}{2^{L-2}}(\frac{3}{4})^{L-2} =: h(L)$$

$h(L)$ is a decreasing function for $L \geq 3$ and achieves the value $0.75$ when $L = 3$, hence since $L \geq 3$ we get $h(L) \leq 1$. Therefore,

$$f^{'}(\frac{3}{d}) = \frac{2P \cdot d}{m}\left((L-1)(d \cdot (\frac{3}{d})^{L-1} - 1)(\frac{3}{d})^{L-2} + (d \cdot \frac{3}{d} - 1)\right) =$$

$$= \frac{2P \cdot d}{m}\left(2 - (L-1)(1 - d \cdot (\frac{3}{d})^{L-1})(\frac{3}{d})^{L-2}\right) \geq \frac{2P \cdot d}{m}\left(2 - 1\right) > 0$$

Hence by continuity, $f^{'}(a)$ has a root within $[-\frac{1}{d}, \frac{3}{d}]$. Note that by Lemma 5, $f^{'}(a)$ doesn't have a root within $[-\frac{1}{d}, \frac{1}{d})$, implying the root is actually achieved in $[\frac{1}{d}, \frac{3}{d}]$. Next, it holds that

$$f^{''}(a) = \frac{2P \cdot d}{m}\left((L-1)(2L-3)d \cdot a^{2L-4} - (L-1)(L-2)a^{L-3} + d\right) \geq$$

$$\geq \frac{2P \cdot d}{m}\left(d - (L-1)(L-2)a^{L-3}\right)$$

Because $d \geq 8$ and $L - 3$ is even, we have for any $a \in [-\frac{1}{d}, \frac{3}{d}]$

$$(L-1)(L-2)a^{L-3} \leq (L-1)(L-2)(\frac{3}{d})^{L-3} \leq \frac{(L-1)(L-2)}{2^{L-3}}(\frac{3}{4})^{L-3} =: g(L)$$

$g(L)$ is a decreasing function for $L \geq 4$ and achieves the value $2.25$ when $L = 4$, hence since $L \geq 4$ we get $g(L) \leq 2.25$. Therefore,

$$f^{''}(a) \geq \frac{2P \cdot d}{m}\left(d - 2.25\right) > 0$$

implying $f^{'}$ is monotonically increasing in $[-\frac{1}{d}, \frac{3}{d}]$. Hence, there exists a unique $s \in [\frac{1}{d}, \frac{3}{d}]$ such that $f^{'}(s) = 0$, which implies that $\nabla \ell(s \cdot \mathbf{1}_d) = 0$. Note that we showed that $s$ is a minimizer of

$f$ over $[-\frac{1}{d}, \frac{3}{d}]$, as $f$'s derivative is zero at $s$ and the second derivative is positive along the interval. Finally, let $a \in \mathbb{R} \setminus [-\frac{1}{d}, \frac{3}{d}]$. By Lemma 5, it holds that

$$f(a) = \ell(a \cdot \mathbf{1}_d) \geq \frac{2P}{m}$$

On the other hand, it also holds that

$$f(s) < f(\frac{1}{d}) = \frac{P}{m}\left((1 - d \cdot (\frac{1}{d})^{L-1})^2 + (1 - d \cdot \frac{1}{d})^2\right) \leq \frac{P}{m}$$

Thus, $s$ is a minimizer of $f$ over $\mathbb{R}$, meaning that $\mathbf{s} := s \cdot \mathbf{1}_d$ is a minimizer of $\ell$ over $\mathcal{W}_1$ as required. On the other hand, since $d \geq 8$ and $L \geq 4$ it holds that

$$1 - d \cdot s^{L-1} \geq 1 - d \cdot (\frac{3}{d})^{L-1} = 1 - 3 \cdot (\frac{3}{d})^{L-2} \geq 1 - 3 \cdot (\frac{3}{8})^2 \geq \frac{1}{2}$$

Therefore,

$$\ell(\mathbf{s}) = \frac{P}{m}\left((1 - d \cdot s^{L-1})^2 + (1 - d \cdot s)^2\right) \geq \frac{P}{m}\left(1 - d \cdot s^{L-1}\right)^2 \geq \frac{P}{4(m)} > 0$$

completing the proof. $\qquad\square$

In the last two lemmas of this section, we explicitly compute an eigendecomposition of $\ell$'s hessian in $\mathbf{s}$ and bound its eigenvalues:

**Lemma 9.** *Consider $\mathbf{s}$ defined in Lemma 8. An eigendecomposition of the symmetric hessian matrix $\nabla^2 \ell(\mathbf{s})$ is the following:*

- *The eigenvector $\mathbf{1}_d$ with the eigenvalue*

$$\lambda_+ := \frac{2P}{m}\left((L-1)((2L-3)d \cdot s^{L-1} - (L-2))s^{L-3} + d\right)$$

- *For $j \in \{2, \ldots, d\}$ the eigenvector $\mathbf{e_1} - \mathbf{e_j}$ with the eigenvalue*

$$\lambda_- := \frac{2P}{m}\left((L-1)(L-2)(d \cdot s^{L-1} - 1)s^{L-3}\right)$$

*Proof.* We begin by computing the hessian matrix $\nabla^2 \ell(A)$ for a general $A \in \mathbb{R}^d$, which is symmetric since $\ell(A)$ is analytic. by Equation (28), for any $j \in [d]$ it holds that

$$\frac{\partial}{\partial a_j}\ell(A) = \frac{2P}{m}\left((L-1)(\sum_{k=1}^d a_k^{L-1} - 1)a_j^{L-2} + (\sum_{k=1}^d a_k - 1)\right)$$

Therefore, for any $j \in [d]$ we have that

$$\left(\nabla^2 \ell(A)\right)_{jj} = \frac{2P}{m}\left((L-1)((2L-3)a_j^{2L-4} + (L-2)\sum_{k=1, k \neq j}^d a_k^{L-1}a_j^{L-3} - (L-2)a_j^{L-3}) + 1\right)$$

Additionally, for any $j, i \in [d]$ such that $j \neq i$ we have that

$$\left(\nabla^2 \ell(A)\right)_{ij} = \frac{2P}{m}\left((L-1)^2 a_i^{L-2} a_j^{L-2} + 1\right)$$

Now we specialize to $A = \mathbf{s}$. For $j \in [d]$, we obtain

$$\left(\nabla^2 \ell(\mathbf{s})\right)_{jj} = \frac{2P}{m}\left((L-1)((2L-3)s^{2L-4} + (L-2)(d-1)s^{2L-4} - (L-2)s^{L-3}) + 1\right) =$$

$$= \frac{2P}{m}\left((L-1)((2L-3 + L \cdot d - 2d - L + 2)s^{2L-4} - (L-2)s^{L-3}) + 1\right) =$$

$$= \frac{2P}{m}\left((L-1)((L-1+L \cdot d - 2d)s^{2L-4} - (L-2)s^{L-3}) + 1\right) =: \omega_1$$

For $j, i \in [d]$ such that $j \neq i$ we obtain that

$$\left(\nabla^2 \ell(\mathbf{s})\right)_{ij} = \frac{2P}{m}\left((L-1)^2 s^{2L-4} + 1\right) =: \omega_2$$

Observe that

$$\nabla^2 \ell(\mathbf{s}) = (\omega_1 - \omega_2)I_d + \omega_2 \mathbf{1}_{d \times d}$$

Hence, by Lemma 27 we obtain that an eigendecomposition for $\nabla^2 \ell(\mathbf{s})$ is the following:

- The eigenvector $\mathbf{1}_d$ with the eigenvalue $\lambda_+ := \omega_1 + (d-1)\omega_2$.

- For $j \in \{2, \ldots, d\}$ the eigenvector $\mathbf{e_1} - \mathbf{e_j}$ with the eigenvalue $\lambda_- := \omega_1 - \omega_2$.

$\lambda_+$ takes the following form:

$$\lambda_+ = \frac{2P}{m}\left((L-1)\left((L-1+L\cdot d-2d)s^{2L-4} - (L-2)s^{L-3}\right)+1+\right.$$

$$\left. + (d-1)\left((L-1)^2 s^{2L-4} + 1\right)\right) =$$

$$= \frac{2P}{m}\left((L-1)\left((L-1+L\cdot d-2d+Ld-d-L+1)s^{2L-4} - (L-2)s^{L-3}\right)+d\right) =$$

$$= \frac{2P}{m}\left((L-1)\left((2L\cdot d-3d)s^{2L-4} - (L-2)s^{L-3}\right)+d\right) =$$

$$= \frac{2P}{m}\left((L-1)\left((2L-3)d\cdot s^{L-1} - (L-2)\right)s^{L-3} + d\right)$$

$\lambda_-$ takes the following form:

$$\lambda_- = \frac{2P}{m}\left((L-1)\left((L-1+L\cdot d-2d)s^{2L-4} - (L-2)s^{L-3}\right)+1-\right.$$

$$\left. - \left((L-1)^2 s^{2L-4} + 1\right)\right) =$$

$$= \frac{2P}{m}\left((L-1)\left((L-1+L\cdot d-2d-L+1)s^{2L-4} - (L-2)s^{L-3}\right)\right) =$$

$$= \frac{2P}{m}\left((L-1)\left((L\cdot d-2d)s^{2L-4} - (L-2)s^{L-3}\right)\right) =$$

$$= \frac{2P}{m}\left((L-1)\left((L-2)d\cdot s^{L-1} - (L-2)\right)s^{L-3}\right) =$$

$$= \frac{2P}{m}\left((L-1)(L-2)\left(d\cdot s^{L-1} - 1\right)s^{L-3}\right)$$

$\square$

We now turn to bounding $\lambda_+$ and $\lambda_-$:

**Lemma 10.** *The eigenvalue $\lambda_+$ from Lemma 9 statisfies*

$$\lambda_+ \geq \frac{2P(d-1)}{m} > 0$$

*The eigenvalue $\lambda_-$ from Lemma 9 statisfies*

$$\lambda_- \in \left(-\frac{2P}{m}, 0\right)$$

*Proof.* Since $s \in [\frac{1}{d}, \frac{3}{d}]$ and since $d \geq 8$ we obtain

$$(L-1)\big((2L-3)d \cdot s^{L-1} - (L-2)\big)s^{L-3} \geq -(L-1)(L-2)s^{L-3} \geq$$

$$-(L-1)(L-2)(\frac{3}{d})^{L-3} \geq -\frac{(L-1)(L-2)}{2^{L-3}}(\frac{3}{4})^{L-3} =: f(L)$$

$f(L)$ is increasing for $L \geq 7$ and achieves a value that is $> -0.6$ for $L = 7$. Hence since $L \geq 7$ we get $f(L) \geq -0.6 > -1$. Therefore,

$$\lambda_+ = \frac{2P}{m}\left((L-1)\big((2L-3)d \cdot s^{L-1} - (L-2)\big)s^{L-3} + d\right) \geq \frac{2P}{m}\left(d-1\right) > 0$$

Next, since $s \in [\frac{1}{d}, \frac{3}{d}]$ and since $d \geq 8$ we obtain

$$(L-1)(L-2)s^{L-3} \leq (L-1)(L-2)(\frac{3}{d})^{L-3} \leq \frac{(L-1)(L-2)}{2^{L-3}}(\frac{3}{4})^{L-3} =: g(L)$$

$g(L)$ is decreasing for $L \geq 7$ and achieves a value that is $< 0.6$ for $L = 7$. Hence since $L \geq 7$ we get $g(L) \leq 0.6 < 1$. Additionally, note that

$$-1 \leq d \cdot s^{L-1} - 1 \leq 3 \cdot (\frac{3}{d})^{L-2} - 1 \leq 3(\frac{3}{8})^{L-2} < 0$$

Therefore, we obtain that

$$(L-1)(L-2)s^{L-3}(d \cdot s^{L-1} - 1) \in (-1, 0)$$

and so

$$\lambda_- = \frac{2P}{m}\left((L-1)(L-2)\big(d \cdot s^{L-1} - 1\big)s^{L-3}\right) \in (-\frac{2P}{m}, 0)$$

which completes our proof. $\qquad\square$

In the first half of D.3.2 we characterized the critical point $\mathbf{s}$ and established that it is the only critical point that is relevant in our case, since it is not a global minimum and since we cannot exclude the possibility that gradient flow would converge to it. In what follows, we give a closed form solution to the dynamics obtained under the linear approximation around $\mathbf{s}$ to our true dynamics. We will show that under these linearized dynamics, any gradient flow trajectory not initialized in $\mathcal{W}_1$ will escape $\mathbf{s}$ at an exponential rate.

**Lemma 11.** *The linear approximation around $\mathbf{s}$ of the gradient flow dynamics (see Equation (28)) is defined by*

$$\dot{A}^{lin}(t) := -\nabla\ell(\mathbf{s}) - \nabla^2\ell(\mathbf{s})(A^{lin}(t) - \mathbf{s}) = -\nabla^2\ell(\mathbf{s})(A^{lin}(t) - \mathbf{s})$$

*The solution to the above linear differential equations system is given by*

$$A^{lin}(t) = Q\exp(-t \cdot \mathrm{Diag}(\lambda_+, \lambda_-, \ldots, \lambda_-))Q^\top(A^{lin}(0) - \mathbf{s}) + \mathbf{s}$$

*where $\lambda_+$ and $\lambda_-$ are the eigenvalues $\nabla^2\ell(\mathbf{s})$ found in Lemma 9, and $Q$ is an orthogonal matrix whose first column is $\frac{1}{\sqrt{d}}\mathbf{1}_d$ and the rest of its columns are an orthonormal basis of $\mathcal{W}_2$ (defined in Equation (16)).*

*Proof.* First note that the first order Taylor's expansion around $\mathbf{s}$ of $-\nabla\ell(A)$ is given by

$$-\nabla\ell(\mathbf{s}) - \nabla^2\ell(\mathbf{s})(A(t) - \mathbf{s})$$

and since $\mathbf{s}$ is a critical point of $\ell$ (*i.e.*, $\nabla\ell(\mathbf{s}) = \mathbf{0}_d$), we obtain the following linear approximation

$$\dot{A}^{lin}(t) = -\nabla^2\ell(\mathbf{s})(A(t) - \mathbf{s})$$

Per Lemma 9, an eigendecomposition of $\nabla^2\ell(\mathbf{s})$ is given by the eigenvector $\mathbf{1}_d$ with the eigenvalue $\lambda_+$, and the eigenvectors $\{\mathbf{e_1} - \mathbf{e_2}, \ldots, \mathbf{e_1} - \mathbf{e_d}\}$ with the eigenvalue $\lambda_-$. Therefore, we may write $\nabla^2\ell(\mathbf{s})$ as the orthogonal eigendecomposition

$$\nabla^2\ell(\mathbf{s}) = Q\,\mathrm{Diag}(\lambda_+, \lambda_-, \ldots, \lambda_-)Q^\top$$

where the first column of $Q$ is $\frac{1}{\sqrt{d}}\mathbf{1}_d$, and the rest of its columns are an orthonormal basis of $span\{\mathbf{e_1} - \mathbf{e_2}, \ldots, \mathbf{e_1} - \mathbf{e_d}\} = \mathcal{W}_2$. The proof is completed by invoking Lemma 28 which yields the following solution to the linear system:

$$A^{lin}(t) = Q\exp(-t \cdot \mathrm{Diag}(\lambda_+, \lambda_-, \ldots, \lambda_-))Q^\top(A^{lin}(0) - \mathbf{s}) + \mathbf{s}$$

$\square$

The following corollary computes the solution of the linear approximation as a function of the initialization's projections onto $\mathcal{W}_1$ and $\mathcal{W}_2$ (Equation (16)):

**Corollary 2.** *Denote the projection of $A^{lin}(0)$ to $\mathcal{W}_1$ by $\beta_1\mathbf{1}_d$ where $\beta_1 \in \mathbb{R}$, and the projection of $A^{lin}(0)$ to $\mathcal{W}_2$ by $\beta_2 \cdot \mathbf{v} \in \mathcal{W}_2$ where $\mathbf{v} \in \mathcal{W}_2$ is a unit vector $\beta_2 \in \mathbb{R}$. Then for any $t \geq 0$ it holds that*

$$A^{lin}(t) = \left(\exp(-t \cdot \lambda_+)(\beta_1 - s) + s\right)\mathbf{1}_d + (\exp(-t \cdot \lambda_-) \cdot \beta_2)\mathbf{v}$$

*Proof.* Plugging the projections of $A^{lin}(0)$ to $\mathcal{W}_1$ and $\mathcal{W}_2$, we can write the following:

$$A^{lin}(0) - \mathbf{s} = (\beta_1 - s)\mathbf{1}_d + \beta_2\mathbf{v}$$

Hence per Lemma 28 at time $t \geq 0$ the solution $A^{lin}(t)$ takes the following form:

$$A^{lin}(t) = Q\exp(-t \cdot \mathrm{Diag}(\lambda_+, \lambda_-, \ldots, \lambda_-))Q^\top\left((\beta_1 - s)\mathbf{1}_d + \beta_2\mathbf{v}\right) + \mathbf{s}$$

$Q$ is a projection matrix to the respective eigenspaces of $\nabla^2\ell(\mathbf{s})$, hence since $\mathbf{1}_d \in \mathcal{W}_1$ and $\mathbf{v} \in \mathcal{W}_2$ we obtain

$$A^{lin}(t) = \left(\exp(-t \cdot \lambda_+)(\beta_1 - s)\right)\mathbf{1}_d + (\exp(-t \cdot \lambda_-) \cdot \beta_2)\mathbf{v} + \mathbf{s} =$$

$$= \left(\exp(-t \cdot \lambda_+)(\beta_1 - s) + s\right)\mathbf{1}_d + (\exp(-t \cdot \lambda_-) \cdot \beta_2)\mathbf{v}$$

as required. $\square$

**Remark 2.** *Note that if $\beta_2 \neq 0$ (i.e. the initialization $A^{lin}(0)$ was not in $\mathcal{W}_1$), then the solution to the system diverges from $\mathbf{s}$. Since $\lambda_- < 0 < \lambda_+$ we obtain*

$$\lim_{t\to\infty}\left(\exp(-t \cdot \lambda_+)(\beta_1 - s) + s\right)\mathbf{1}_d = s \cdot \mathbf{1}_d = \mathbf{s}$$

$$\lim_{t\to\infty}\|(\exp(-t \cdot \lambda_-) \cdot \beta_2)\mathbf{v} - \mathbf{s}\| \to \infty$$

*On the other hand, if $\beta_2 = 0$ then the solution to the system converges to $\mathbf{s}$.*

### D.3.3 LINEARIZATION OF DYNAMICAL SYSTEMS

In D.3.2 we characterized the critical point $\mathbf{s} \in 1_d$ and established that it is the only non global minimum that we could converge to given our initialization. We would now like to show that in fact gradient flow will escape $\mathbf{s}$ and converge rapidly towards a global minimum. Corollary 2 gives some indication why this may be the case - it shows that the local linearization of the dynanics near $\mathbf{s}$ will tend to repel any trajectory which is not on the line $\mathcal{W}_1$. Intuitively one expects that once we are sufficiently close to $\mathbf{s}$, the linearized dynamics provide a sufficiently good approximation to ensure that the same conclusion will hold for the nonlinear system as well. Unfortunately, existing results from the optimization literature (e.g. Jin et al. (2017)) give escape times which do not suffice for our purposes[11]. To obtain the required bounds on the escape time we will require some results from dynamical systems theory. Informally, the idea is that if a non linear dynamical system satisfies

---

[11]recall that our strategy is to bound the divergence between our trajectory and the reference one, and this divergence depends on the convergence time achieved by gradient flow.

certain conditions on the spectrum of its linearization (these are sometimes called "non-resonance conditions"), then it is locally smoothly equivalent to its linearization. This will allow us to bound the escape time of gradient flow in terms of the closed form dynamics obtained for the linearization in Corollary 2.

We begin by defining the notions of *smooth conjugation* and *smooth linearization* of dynamical systems:

**Definition 5.** Let $f, g : \mathbb{R}^d \to \mathbb{R}^d$ be two $C^M$ vector fields with a common fixed point $\mathbf{x_0} \in \mathbb{R}^d$, *i.e.*, $f(\mathbf{x_0}) = g(\mathbf{x_0}) = \mathbf{0}_d$. For any $K \in [M]$ we say that $f$ and $g$ are $C^K$-*conjugate* near $\mathbf{x_0}$ when there exist neighborhoods $\mathcal{V}_1, \mathcal{U}_1 \subseteq \mathbb{R}^d$ such that $\mathbf{x_0} \in \mathcal{V}_1, \mathcal{U}_1$ and there exist a $C^K$-diffeomorphism $H : \mathcal{V}_1 \to \mathcal{U}_1$ satisfying the following:

- $H(\mathbf{x_0}) = \mathbf{x_0}$

- Whenever $\mathbf{x}(t) \in \mathcal{V}_1$ is a solution of $\dot{\mathbf{x}}(t) = f(\mathbf{x}(t))$ for $t$ in some interval $\mathcal{I} \subseteq \mathbb{R}$ then $\mathbf{y}(t) = H(\mathbf{x}(t))$ is a solution of $\dot{\mathbf{y}}(t) = g(\mathbf{y}(t))$ for $t \in \mathcal{I}$.

- Whenever $\mathbf{y}(t) \in \mathcal{U}_1$ is a solution of $\dot{\mathbf{y}}(t) = g(\mathbf{y}(t))$ for $t$ in some interval $\mathcal{I} \subseteq \mathbb{R}$ then $\mathbf{x}(t) = H^{-1}(\mathbf{y}(t))$ is a solution of $\dot{\mathbf{x}}(t) = f(\mathbf{x}(t))$ for $t \in \mathcal{I}$.

The mapping $H$ is referred to as the $C^K$-*conjugation* between $\dot{\mathbf{x}}(t) = f(\mathbf{x}(t))$ and $\dot{\mathbf{y}}(t) = g(\mathbf{y}(t))$. Consider the first order Taylor's expansion of $f$ around $\mathbf{x_0}$ given by

$$\dot{\mathbf{x}}(t) = f(\mathbf{x}(t)) = A(\mathbf{x}(t) - \mathbf{x_0}) + F(\mathbf{x}(t) - \mathbf{x_0})$$

where $A = Df(\mathbf{x_0})$, $F(\mathbf{0_d}) = \mathbf{0}_d$ and $DF(\mathbf{0}_d) = \mathbf{0}_{d \times d}$. The associated linear equation is given by

$$\dot{\mathbf{y}}(t) = A(\mathbf{y}(t) - \mathbf{x_0})$$

We say that $f$ admits a $C^K$-*linearization* near $\mathbf{x_0}$ when it is $C^K$-conjugate near $\mathbf{x_0}$ with its linear approximation.

We now introduce the *Strict Hyperbolicity* property and the *Non-resonance condition* (also known as the Sternberg condition). These are sufficient conditions for a dynamical system to admit a smooth linearization and we will later show our system satisfies them:

**Definition 6.** Let $A \in \mathbb{R}^{d \times d}$ be a matrix with eigenvalues $\lambda_1, \ldots, \lambda_d \in \mathbb{R}$ repeated with multiplicities. We say that $A$ is *strictly hyperbolic* when:

- For all $j \in [d]$ it holds that $\lambda_j \neq 0$.

- There exist $j_+, j_- \in [d]$ such that $\lambda_{j_+} > 0$ and $\lambda_{j_-} < 0$.

**Definition 7.** Let $A \in \mathbb{R}^{d \times d}$ be a matrix with eigenvalues $\lambda_1, \ldots, \lambda_d \in \mathbb{R}$ repeated with multiplicities. For any $\mathbf{m} \in \mathbb{N}_{\geq 0}^d$ non-negative integers vector and any $\lambda \in \mathbb{R}$ we denote $\gamma(\lambda, \mathbf{m})$ as the following quantity:

$$\gamma(\lambda, \mathbf{m}) := \lambda - \sum_{k=1}^{d} m_k \cdot \lambda_k$$

For any $N \in \mathbb{N}$ such that $N \geq 2$ we say that $A$ satisfies the *non-resonance condition* of order $N$ when for all $j \in [d]$ and all $\mathbf{m} \in \mathbb{N}_{\geq 0}^d$ such that $\sum_{k=1}^{d} m_k \in \{2, \ldots, N\}$ it holds that $\gamma(\lambda_j, \mathbf{m}) \neq 0$.

Finally, we present the property of *matrix Q-smoothness*:

**Definition 8.** Let $A \in \mathbb{R}^{d \times d}$ be a matrix with eigenvalues $\lambda_1, \ldots, \lambda_d \in \mathbb{R}$ repeated with multiplicities. Suppose $A$ is strictly hyperbolic. Denote the following quantities:

$$\rho_+ := \frac{\max\{|\lambda_j| : j \in [d], \lambda_j > 0\}}{\min\{|\lambda_j| : j \in [d], \lambda_j > 0\}}, \quad \rho_- := \frac{\max\{|\lambda_j| : j \in [d], \lambda_j < 0\}}{\min\{|\lambda_j| : j \in [d], \lambda_j < 0\}}$$

Let $Q \in \mathbb{N}_{>0}$. We define the *Q-smoothness* of $A$ to be the largest integer $K \in \mathbb{N}_{\geq 0}$ for which there exist $M, N \in \mathbb{N}_{>0}$ satisfying the following:

- $Q = M + N$

- $M - K\rho_+ \geq 0$

- $N - K\rho_- \geq 0$

We are now ready to present Theorem 1 of Sell (1985), which states conditions under which there exists a smooth linearization of a dynamical system[12]:

**Theorem 3** (Theorem 1 of Sell (1985) (adapted)). *Let $Q \in \mathbb{N}$ such that $Q \geq 2$. Let $f : \mathbb{R}^d \to \mathbb{R}^d$ be an analytic (i.e. $C^\infty$) vector field with a fixed point $\mathbf{x_0}$. Consider the first order Taylor's expansion of $f$ around $\mathbf{x_0}$ given by*

$$\dot{\mathbf{x}}(t) = f(\mathbf{x}(t)) = A(\mathbf{x}(t) - \mathbf{x_0}) + F(\mathbf{x}(t) - \mathbf{x_0})$$

*where $A = Df(\mathbf{x_0})$, $F(\mathbf{0_d}) = \mathbf{0}_d$ and $DF(\mathbf{0}_d) = \mathbf{0}_{d \times d}$. If $A$ is strictly hyperbolic (Definition 6) and satisfies the non-resonance condition of order $Q$ (Definition 7) then $f$ admits a $C^K$ linearization near $\mathbf{x_0}$ where $K$ is the $Q$-smoothness of $A$ (Definition 8).*

*Proof.* See proof of Theorem 1 in Sell (1985). $\qquad\square$

Having introduced these general results on local linearization, we now show that the dynamical system induced by gradient flow admits a smooth linearization near $\mathbf{s}$. We begin by showing that $-\nabla^2 \ell(\mathbf{s})$ is strictly hyperbolic and satisfies the non-resonance condition:

**Proposition 5.** *Consider $\mathbf{s}$ defined in Lemma 8. The hessian matrix $-\nabla^2 \ell(\mathbf{s})$ is strictly hyperbolic and satisfies the non-resonance condition of order $d - 2$.*

*Proof.* Per Lemma 10, it holds that

$$\lambda_+ \geq \frac{2P(d-1)}{m} > 0$$

and

$$-\frac{2P}{m} < \lambda_- < 0$$

Hence by Definition 6, $\nabla^2 \ell(\mathbf{s})$ is strictly hyperbolic. Additionally, for any $m \in \{0, \ldots, d-2\}$ we have that

$$\lambda_+ + m \cdot \lambda_- \geq \frac{2P(d-1)}{m} - \frac{2P \cdot m}{m} > 0$$

Let $\mathbf{m} \in \mathbb{N}_{\geq 0}^d$ such that $\sum_{k=1}^d m_k \in \{2, \ldots, d-2\}$. Per definition Definition 7 we have that

$$\gamma(\lambda_+, \mathbf{m}) = (1 - m_1)\lambda_+ - \lambda_- \sum_{k=2}^d m_k$$

If $m_1 \in \{0, 1\}$ then since $\lambda_- < 0 < \lambda_+$ and $\sum_{k=2}^d m_k \in \{1, \ldots, d-2\}$ we obtain

$$\gamma(\lambda_+, \mathbf{m}) \geq -\lambda_- \sum_{k=2}^d m_k > 0$$

Otherwise, since $\sum_{k=2}^d m_k \in \{0, \ldots, d-4\}$ we obtain by the above that

$$\gamma(\lambda_+, \mathbf{m}) \leq -\lambda_+ - \lambda_- \sum_{k=2}^d m_k \leq -\frac{2P(d-1)}{m} + \frac{2P \sum_{k=2}^d m_k}{m} < 0$$

Hence $\gamma(\lambda_+, \mathbf{m}) \neq 0$. Next, per Definition 7 we have that

$$\gamma(\lambda_-, \mathbf{m}) = (1 - \sum_{k=2}^d m_k)\lambda_- - m_1 \cdot \lambda_+$$

---

[12]We present slightly adapted results that are specialized to our setting.

If $m_1 = 0$ then since $1 - \sum_{k=2}^{d} m_k \in \{-1, \ldots, -d+3\}$ we obtain

$$\gamma(\lambda_-, \mathbf{m}) = (1 - \sum_{k=2}^{d} m_k)\lambda_- > 0$$

If $m_1 = d - 2$ then $1 - \sum_{k=2}^{d} m_k = 1$ and so

$$\gamma(\lambda_-, \mathbf{m}) = \lambda_- - (d-2)\lambda_+ < 0$$

Otherwise, since $\sum_{k=2}^{d} m_k - 1 \in \{0, \ldots, d-3\}$ we obtain by the above that

$$\gamma(\lambda_-, \mathbf{m}) \le -\lambda_+ - (\sum_{k=2}^{d} m_k - 1)\lambda_- \le -\frac{2P(d-1)}{m} + \frac{2P(\sum_{k=2}^{d} m_k - 1)}{m} < 0$$

Hence $\gamma(\lambda_-, \mathbf{m}) \ne 0$. Therefore by Definition 7, $-\nabla^2\ell(\mathbf{s})$ satisfies the non-resonance condition of order $d - 2$. $\qquad\square$

Next, we turn to lower bound the $Q$-smoothness of $-\nabla^2\ell(\mathbf{s})$:

**Proposition 6.** *For any $Q \in \mathbb{N}_{>0}$, the $Q$-smoothness of $-\nabla^2\ell(\mathbf{s})$ is at least $\lfloor \frac{Q}{2} \rfloor$.*

*Proof.* Per Lemma 10, we have the following:

$$\rho_+ = \frac{\max\{\lambda_+\}}{\min\{\lambda_+\}} = 1, \ \rho_- = \frac{\max\{\lambda_-\}}{\min\{\lambda_-\}} = 1$$

Therefore per Definition 8 and since $\nabla^2\ell(\mathbf{s})$ is strictly hyperbolic, the $Q$-smoothness of $-\nabla^2\ell(\mathbf{s})$ is the largest $K \in \mathbb{N}_{\ge 0}$ for which there exist $M, N \in \mathbb{N}_{>0}$ such that

- $Q = M + N$

- $M - K \ge 0$

- $N - K \ge 0$

One can easily verify this implies that the $Q$-smoothness of $-\nabla^2\ell(\mathbf{s})$ is at least $\lfloor \frac{Q}{2} \rfloor$. $\qquad\square$

Finally, we are ready to prove the following proposition which shows that our dynamical system induced by gradient flow admits a linearization which is at least $C^3$:

**Proposition 7.** *The dynamical system induced by gradient flow (see Equation (28)) admits a linearization near $\mathbf{s}$ that is at least $C^3$.*

*Proof.* First note that the vector field $-\nabla\ell(A)$ which gradient flow follows is analytic. Next, per Propositions 5 and 6 it holds that $-\nabla^2\ell(\mathbf{s})$ is strictly hyperbolic, satisfies the non-resonance condition of order at least $d - 2$, and has $Q$-smoothness of at least $\lfloor \frac{Q}{2} \rfloor$ for any $Q \in \mathbb{N}_{>0}$. Hence, by Theorem 3 the vector field $-\nabla\ell(A)$ admits a $C^{\lfloor \frac{d-2}{2} \rfloor}$-linearization near $\mathbf{s}$. The proof concludes by noting that $d \ge 8$ hence $\lfloor \frac{d-2}{2} \rfloor \ge 3$. $\qquad\square$

We denote the above linearization by $H : \mathcal{V}_1 \to \mathcal{U}_1$, where the neighborhoods $\mathcal{V}_1, \mathcal{U}_1 \subseteq \mathbb{R}^d$ are such that $\mathbf{s} \in \mathcal{V}_1, \mathcal{U}_1$. To set the stage for the rest of the proof, we prove the following proposition that considers a restriction of $H$ to a smaller domain that satisfies a few additional conditions which we will require later:

**Proposition 8.** *There exists $r_1 > 0$ which satisfies the following:*

1. $r_1 \le \frac{1}{2d}$

2. *For any $A \in \overline{B_{r_1}}(\mathbf{s})$ it holds that*

$$|\lambda_{min}(\nabla^2\ell(A))| \le 2|\lambda_-|$$

3. $H|_{\overline{B_{r_1}}(\mathbf{s})}$ *is Lipschitz and there exist* $r_2 \in (0, r_1)$ *and* $r_3 > 0$ *such that* $H^{-1}|_{\overline{B_{r_3}}(\mathbf{s})}$ *is Lipschitz and it holds that*

$$H[\overline{B_{r_2}}(\mathbf{s})] \subseteq \overline{B_{r_3}}(\mathbf{s}) \subseteq H[\overline{B_{r_1}}(\mathbf{s})]$$

*Proof.* We show there exist three non empty intervals of the form $(0, b_i]$ for $i \in [3]$, such that if $r_1 \in (0, b_i]$ then it satisfies the corresponding requirement above. This would imply that the minimal upper limit $r_1 := \min\{b_1, b_2, b_3\}$ satisfies all requirements. The first condition is trivial, with $b_1 = \frac{1}{2d}$. Next, since $\nabla\ell(A)$ is analytic it holds that $\nabla^2\ell(A)$ is symmetric for any $A \in \mathbb{R}^d$. Since $\lambda_- < 0$, by the continuity of the eigenvalues of $\nabla^2\ell(A)$ around $\mathbf{s}$ there exists $b_2 > 0$ such that the second requirement is satisfied for all $(0, b_2]$. Lastly, since $H : \mathcal{V}_1 \to \mathcal{U}_1$ is $C^3$ and since $\mathcal{V}_1, \mathcal{U}_1$ are neighborhoods of $\mathbf{s}$, we can invoke Lemma 29 which states that there exists $b_3 > 0$ such that for any $b \in (0, b_3]$ the third and fourth requirements are satisfied. $\qquad\square$

### D.3.4 Movement towards the saddle s

Having established key properties of the loss landscape, we are ready to begin the dynamical analysis of the gradient flow trajectories over time. We first give a simple bound on the magnitude of the entries of $A(t)$:

**Lemma 12.** *Suppose we initialize at* $A(0) \in \mathcal{I}_0$ *and at* $A^{ref}(0)$, *and evolve* $A(t)$ *and* $A^{ref}(t)$ *according to Equation* (28). *For any* $t \geq 0$ *and any* $j \in [d]$ *it holds that*

$$A_j(t), A_j^{ref}(t) \in [-3, 3]$$

*Proof.* First, we have shown in Lemma 6 that the initialization $\mathcal{I}_0$ guarantees all points encountered by gradient flow have loss no larger than $\frac{2P}{m}$. Assume on the contrary that there exist $t \geq 0$ and $j \in [d]$ for which

$$A_j(t) \notin [-3, 3]$$

Since $L - 1$ is even, we obtain that $a_j(t)^{L-1} \geq 3^{L-1} > 3$ and that any $k \in [d], k \neq j$ satisfies $a_k(t)^{L-1} \geq 0$. Hence, we obtain that

$$\ell(A(t)) = \frac{2P}{m}\left((1 - \sum_{k=1}^d a_k(t)^{L-1})^2 + (1 - \sum_{k=1}^d a_k(t))^2\right) \geq \frac{2P}{m}(1 - \sum_{k=1}^d a_k(t)^{L-1})^2 \geq$$

$$\geq \frac{2P}{m}(3 - 1)^2 > \frac{2P}{m}$$

in contradiction to Lemma 23. The proof is identical when we consider the reference trajectory. $\quad\square$

The above yields the following useful corollary:

**Corollary 3.** *There exists* $N > 0$ *such if we initialize at* $A(0) \in \mathcal{I}_0$ *and at* $A^{ref}(0)$, *and evolve* $A(t)$ *and* $A^{ref}(t)$ *according to Equation* (28) *then for any* $t \geq 0$ *the functions* $-\nabla\ell(A(t))$ *and* $-\nabla\ell(A^{ref}(t))$ *are N-Lipschitz.*

*Proof.* As shown in Lemma 12, all points encountered by gradient flow are contained in the compact set $[-3, 3]^d$. The claim thus follows from the fact that $\ell(A)$ is analytic. $\qquad\square$

We continue to prove the following lemma which analyzes the trajectories when initializing in an interval of points on the line $\mathcal{W}_1$ (Equation (16)):

**Lemma 13.** *Let* $a_1, a_2 \in [-\frac{1}{d}, \frac{4}{d}] \setminus \{s\}$ *such that* $a_1 \neq a_2$. *Suppose we initialize at* $A_1(0) = a_1 \cdot \mathbf{1}_d$ *and* $A_2(0) = a_2 \cdot \mathbf{1}_d$, *and evolve* $A_1(t)$ *and* $A_2(t)$ *according to Equation* (28). *It holds that:*

- *There exist functions* $a_1, a_2 : \mathbb{R}_{\geq 0} \to \mathbb{R}$ *such that*

$$A_1(t) = a_1(t) \cdot \mathbf{1}_d, \quad A_2(t) = a_2(t) \cdot \mathbf{1}_d$$

- *For any* $t \geq 0$ *it holds that* $a_2(t) < a_1(t) \iff a_2 < a_1$

- *For any $r > 0$ there exists $t_1 \geq 0$ such that for any $t \geq t_1$ it holds that $A_1(t), A_2(t) \in \overline{B_r}(\mathbf{s})$*

*Proof.* When initializing at $A(0) = a \cdot \mathbf{1}_d$ and evolving $A(t)$ according to the gradient flow dynamics, all entries evolve according to the same dynamics and therefore must stay equal throughout the optimization. Concretely, all entries obey the following dynamics:

$$\dot{a}(t) = \frac{2P}{m}\left((L-1)(1 - d \cdot a(t)^{L-1})a(t)^{L-2} + (1 - d \cdot a(t))\right)$$

Hence, the first claim holds. Rewriting the above in terms of Lemma 8, we have

$$\dot{a}(t) = -\frac{1}{d}f'(a(t))$$

In Lemmas 7 and 8, we showed that the above expression is positive for $a(t) \in [-\frac{1}{d}, s)$ and equals zero at $s$. We now show that the above expression is negative for $a(t) \in (s, \frac{4}{d}]$ - indeed, since $d \geq 8$ we have that

$$-\frac{1}{d}f'(\frac{4}{d}) = \frac{2P}{m}\left((L-1)(1 - d \cdot (\frac{4}{d})^{L-1})(\frac{4}{d})^{L-2} + (1 - d \cdot (\frac{4}{d}))\right) \leq$$

$$\leq \frac{2P}{m}\left((L-1)(\frac{4}{d})^{L-2} - 3\right) \leq \frac{2P}{m}\left(\underbrace{\frac{L-1}{1.5^{L-2}}(\frac{4}{5\frac{1}{3}})^{L-2} - 3}_{=:h(L)}\right) \leq \frac{2P}{m}\left(0.75 - 3\right) < 0$$

where the second to last inequality stems from the fact that $h(L)$ is decreasing for $L \geq 4$, that $h(4) = 0.75$, and that $L \geq 7$. Next, we also have for any $a \in (s, \frac{4}{d}]$ that

$$-\frac{1}{d}f''(a) = \frac{2P}{m}\left(-(L-1)(2L-3)d \cdot a^{2L-4} + (L-1)(L-2)a^{L-3} - d\right) \leq$$

$$\leq \frac{2P}{m}\left(-d + (L-1)(L-2)a^{L-3}\right) \leq \frac{2P}{m}\left(-d + (L-1)(L-2)(\frac{4}{d})^{L-3}\right) \leq$$

$$\leq \frac{2P}{m}\left(-d + \underbrace{\frac{(L-1)(L-2)}{1.5^{L-2}}(\frac{4}{5\frac{1}{3}})^{L-3}}_{=:g(L)}\right) \leq \frac{2P}{m}(-d + \frac{15}{8}) < 0$$

where the second to last inequality stems from the fact that $g(L)$ is decreasing for $L \geq 7$, that $h(4) = \frac{15}{8}$, and that $L \geq 7$. Therefore, since $-\frac{1}{d}f'(s) = 0$ and from monotonicity we obtain that $-\frac{1}{d}f'(a(t))$ is negative for $a(t) \in (s, \frac{4}{d}]$. We continue by noting that per Lemma 31, trajectories of the same system of ODEs with different initalizations must never meet, hence by continuity it must hold that $a_2(t) < a_1(t) \iff a_2 < a_1$ for all $t \geq 0$. Finally, since $\mathbf{s}$ is a critical point in the interval and since $a_1(t), a_2(t)$ evolve monotonically (increase if initialized $< s$ and decrease otherwise), we get by Lemma 31 that $a_1(t)$ and $a_2(t)$ cannot reach $\mathbf{s}$ in any finite time. However, since $\mathbf{s}$ is the unique critical point in the interval, $a_1(t)$ and $a_2(t)$ must converge to $\mathbf{s}$ as $t \to \infty$. Hence, there exists some time $t_1 \geq 0$ such that for any $t \geq t_1$ the entries of both $A_1(t)$ and $A_2(t)$ are within $\overline{B_r}(\mathbf{s})$. $\qquad \square$

We use $A^Z(t)$ and $A^-(t)$ to denote the trajectories generated by initializing at $\mathbf{0}_d$ and $-\frac{1}{d} \cdot \mathbf{1}_d$ respectively and evolving according to Equation (28). Additionally, for any $r > 0$ we use $t_1(r) \geq 0$ to denote the minimal time which satisfies

$$A^Z(t_1(r)) \in \overline{B_r}(\mathbf{s})$$

Note this means that for $r \in (0, s)$ we have $A^Z(t_1(r)) = \frac{r}{\sqrt{d}} \cdot \mathbf{1}_d + \mathbf{s}$.

We denote the following projections of the trajectory $A(t)$ and the reference trajectory $A^{ref}(t)$, which will be used in the rest of the proof:

**Definition 9.** Suppose we initialize at $A(0) \in \mathcal{I}_0$ and at $A^{ref}(0)$, and evolve $A(t)$ and $A^{ref}(t)$ according to Equation (28). For any $t \geq 0$, we denote the projections of $A(t)$ and $A^{ref}(t)$ to the subspace $\mathcal{W}_1$ using

$$\beta_1(t) \cdot \mathbf{1}_d, \ \beta_1^{ref}(t) \cdot \mathbf{1}_d$$

where $\beta_1(t), \beta_1^{ref}(t) \in \mathbb{R}$. Additionally, we denote the projections of $A(t)$ and $A^{ref}(t)$ to the subspace $\mathcal{W}_2$ using

$$\beta_2(t) \cdot \mathbf{v}(t), \ \beta_2^{ref}(t) \cdot \mathbf{v}^{ref}(t)$$

where $\beta_2(t), \beta_2^{ref}(t) \in \mathbb{R}$ and $\mathbf{v}(t), \mathbf{v}^{ref}(t) \in \mathcal{W}_2$ are unit vectors. Per Equation (16), $\mathcal{W}_1$ and $\mathcal{W}_2$ are orthogonal and span $\mathbb{R}^d$, hence we may write

$$A(t) = \beta_1(t) \cdot \mathbf{1}_d + \beta_2(t) \cdot \mathbf{v}(t)$$
$$A^{ref}(t) = \beta_1^{ref}(t) \cdot \mathbf{1}_d + \beta_2^{ref}(t) \cdot \mathbf{v}^{ref}(t)$$

**Remark 3.** *Per Equation* (16)*, $\mathcal{W}_1$ and $\mathcal{W}_2$ are orthogonal therefore since $(\beta_1(t) \cdot \mathbf{1}_d)^\top (\beta_2(t) \cdot \mathbf{v}(t)) = 0$ and $(\beta_1^{ref}(t) \cdot \mathbf{1}_d)^\top (\beta_2(t)^{ref} \cdot \mathbf{v}^{ref}(t)) = 0$, we obtain*

$$Dist(A(t), \mathcal{W}_1) = \|\beta_2(t) \cdot \mathbf{v}(t)\| = |\beta_2(t)|$$
$$Dist(A^{ref}(t), \mathcal{W}_1) = \|\beta_2^{ref}(t) \cdot \mathbf{v}^{ref}(t)\| = |\beta_2^{ref}(t)|$$

*and*

$$Dist(A(t), \mathcal{W}_2) = \|\beta_1(t) \cdot \mathbf{1}_d\| = \sqrt{d}|\beta_1(t)|$$
$$Dist(A^{ref}(t), \mathcal{W}_2) = \|\beta_1^{ref}(t) \cdot \mathbf{1}_d\| = \sqrt{d}|\beta_1^{ref}(t)|$$

Before proving the main claim of this section, we introduce another condition on the initialization which we denote $\mathcal{I}_3$:

**Definition 10.** Let $r > 0$. We use $\mathcal{I}_3(r)$ to denote the following subset of $\mathcal{I}_0$:

$$\mathcal{I}_3(r) := \left\{ A \in \mathcal{I}_0 : \alpha \leq \frac{\min\{r, \|A^Z(t_1(r)) - A^-(t_1(r))\|_2\}}{6d} \exp(-N \cdot t_1(r)), \ \zeta_d \leq \frac{1}{2} \right\}$$

for $N$ of Corollary 3 and for $A^Z(t)$, $A^-(t)$ and $t_1(r)$ of Lemma 13.

We are now ready to prove the main claim of this section, which states that under the above on the initalization, both the original and reference trajectories must enter a sufficiently small sphere around $\mathbf{s}$ and furthermore they arrive at points that are sufficiently faraway from $\mathcal{W}_1$:

**Proposition 9.** *Let $r \in (0, s)$. Suppose we initialize at $A(0) \in \mathcal{I}_3(\frac{r}{4})$ and at $A^{ref}(0)$, and evolve $A(t)$ and $A^{ref}(t)$ according to Equation* (28)*. There exist constants $D_+(r), D_-(r) > 0$ such that:*

- $A\left(t_1(\frac{r}{4})\right), A^{ref}\left(t_1(\frac{r}{4})\right) \in \overline{B_{\frac{r}{2}}}(\mathbf{s})$

- $|\beta_2\left(t_1(\frac{r}{4})\right)|, |\beta_2^{ref}\left(t_1(\frac{r}{4})\right)| \in [\alpha \cdot D_-(r), \alpha \cdot D_+(r)]$

*Proof.* Consider the trajectories $A^Z(t)$ and $A^-(t)$ introduced in Lemma 13. Per Lemma 13, for any time $t \geq 0$ and any index $j \in [d]$ we have

$$a_j^-(t) < a_j^Z(t) < s$$

We begin by showing that for $A(0) \in \mathcal{I}_3(\frac{r}{4})$, the distance between $A^Z\left(t_1(\frac{r}{4})\right)$ and $A\left(t_1(\frac{r}{4})\right)$ is at most $\frac{r}{24}$. First note that per Lemma 13, $A^Z(t)$ never leaves $\overline{B_s}(\mathbf{s}) \subseteq [-3, 3]^d$. Thus per Corollary 3, both $A^Z(t)$ and $A(t)$ are always contained in a compact domain where the vector field $-\nabla \ell(A)$ is $N$-Lipschitz. Therefore, we can invoke Lemma 30 which results in the following:

$$\|A^Z\left(t_1(\frac{r}{4})\right) - A\left(t_1(\frac{r}{4})\right)\|_2 \leq \|A^Z(0) - A(0)\|_2 \cdot \exp(N \cdot t_1(\frac{r}{4})) =$$
$$= \|A(0)\|_2 \cdot \exp(N \cdot t_1(\frac{r}{4})) \leq \alpha \cdot d \cdot \exp(N \cdot t_1(\frac{r}{4}))$$

Per Definition 10, $\alpha$ satisfies

$$\alpha \leq \frac{\min\{\frac{r}{4}, \|A^Z\big(t_1(\frac{r}{4})\big) - A^-\big(t_1(\frac{r}{4})\big)\|_2\}}{6d} \exp(-N \cdot t_1(\frac{r}{4}))$$

Hence, we obtain that

$$\|A^Z\big(t_1(\frac{r}{4})\big) - A\big(t_1(\frac{r}{4})\big)\|_2 \leq$$

$$\leq \frac{\min\{\frac{r}{4}, \|A^Z\big(t_1(\frac{r}{4})\big) - A^-\big(t_1(\frac{r}{4})\big)\|_2\}}{6d} \exp(-N \cdot t_1(\frac{r}{4})) \cdot d \cdot \exp(N \cdot t_1(\frac{r}{4})) =$$

$$= \frac{\min\{\frac{r}{4}, \|A^Z\big(t_1(\frac{r}{4})\big) - A^-\big(t_1(\frac{r}{4})\big)\|_2\}}{6} \leq \frac{r}{24}$$

Therefore, using the triangle inequality we obtain

$$\|A\big(t_1(\frac{r}{4})\big) - \mathbf{s}\|_2 \leq \|A^Z\big(t_1(\frac{r}{4})\big) - \mathbf{s}\|_2 + \|A\big(t_1(\frac{r}{4})\big) - A^Z\big(t_1(\frac{r}{4})\big)\|_2 \leq \frac{r}{4} + \frac{r}{24} \leq \frac{r}{2}$$

Hence, $A(t_1) \in \overline{B_{\frac{r}{2}}}(\mathbf{s})$. Next, by Remark 3 we obtain that

$$|\beta_2\big(t_1(\frac{r}{4})\big)| = \mathrm{Dist}\bigg(A\big(t_1(\frac{r}{4})\big), \mathcal{W}_1\bigg)$$

By Lemma 13 we have $A^Z\big(t_1(\frac{r}{4})\big) \in \mathcal{W}_1$, hence

$$|\beta_2\big(t_1(\frac{r}{4})\big)| \leq \|A^Z\big(t_1(\frac{r}{4})\big) - A\big(t_1(\frac{r}{4})\big)\|_2 \leq \alpha \cdot d \cdot \exp(N \cdot t_1(\frac{r}{4}))$$

Thus, denoting $D_+(r) := d \cdot \exp(N \cdot t_1(\frac{r}{4}))$ we get the first part of the second claim. We now show that $\beta_1\big(t_1(\frac{r}{4})\big) \in (-\frac{1}{d}, s)$; Per Lemma 13, we get by definition of $t_1$ that

$$\|A^Z\big(t_1(\frac{r}{4})\big) - \mathbf{s}\|_2 = \frac{r}{4}$$

and so since $\|A^Z\big(t_1(\frac{r}{4})\big) - A\big(t_1(\frac{r}{4})\big)\|_2 \leq \frac{r}{24}$ and $r < s$ it must hold that $\beta_1\big(t_1(\frac{r}{4})\big) < s$. Since $\|A^Z\big(t_1(\frac{r}{4})\big) - A\big(t_1(\frac{r}{4})\big)\|_2 \leq \frac{\|A^Z\big(t_1(\frac{r}{4})\big) - A^-\big(t_1(\frac{r}{4})\big)\|_2}{6}$, it must hold that

$$\beta_1\big(t_1(\frac{r}{4})\big) > a^-\big(t_1(\frac{r}{4})\big)$$

where $A^-\big(t_1(\frac{r}{4})\big) = a^-\big(t_1(\frac{r}{4})\big) \cdot \mathbf{1}_d$. Note that by Lemma 13 we obtain

$$\beta_1\big(t_1(\frac{r}{4})\big) > a^-\big(t_1(\frac{r}{4})\big) > -\frac{1}{d}$$

as $a^-(t)$ is monotonically increasing. Therefore by Lemma 13 and by continuity, there must exist some point $a \in (-\frac{1}{d}, \beta_1\big(t_1(\frac{r}{4})\big))$ such that if we initialize $A^a(0) = a \cdot \mathbf{1}_d$ and evolve $A^a(t)$ according to the gradient flow dynamics, then it holds that

$$A^a\big(t_1(\frac{r}{4})\big) = \beta_1\big(t_1(\frac{r}{4})\big) \cdot \mathbf{1}_d$$

Per Lemma 13, $A^a(t)$ never leaves $[-3, 3]^d$ where the vector field $-\nabla\ell(A)$ is $N$-Lipschitz. Thus, invoking Lemma 30 we obtain

$$|\beta_2\big(t_1(\frac{r}{4})\big)| = \|\beta_2\big(t_1(\frac{r}{4})\big) \cdot \mathbf{v}\big(t_1(\frac{r}{4})\big)\|_2 =$$

$$= \|\beta_2\big(t_1(\frac{r}{4})\big) \cdot \mathbf{v}\big(t_1(\frac{r}{4})\big) + \beta_1\big(t_1(\frac{r}{4})\big) \cdot \mathbf{1}_d - \beta_2\big(t_1(\frac{r}{4})\big) \cdot \mathbf{v}\big(t_1(\frac{r}{4})\big)\|_2 =$$

$$= \|A\big(t_1(\frac{r}{4})\big) - A^a\big(t_1(\frac{r}{4})\big)\| \geq \|A(0) - A^a(0)\| \cdot \exp(-N \cdot t_1(\frac{r}{4}))$$

As $A^a(0) \in \mathcal{W}_1$, we can lower bound the right hand side by the distance between $A(0)$ and $\mathcal{W}_1$ and obtain

$$|\beta_2\big(t_1(\frac{r}{4})\big)| \geq \mathrm{Dist}(A(0), \mathcal{W}_1) \cdot \exp(-N \cdot t_1(\frac{r}{4}))$$

Next, observe that $\zeta_d \le \frac{1}{2}$ since $A(0) \in \mathcal{I}_3(\frac{r}{4})$, hence

$$\zeta_1 - \frac{\sum_{k=1}^d \zeta_k}{d} \ge 1 - \frac{d-1}{d} - \frac{1}{2d} = \frac{1}{2d}$$

Therefore,

$$\text{Dist}(A(0), \mathcal{W}_1) = \sqrt{\sum_{k=1}^d (\alpha \cdot \zeta_k - \frac{\sum_{k=1}^d \alpha \cdot \zeta_k}{d})^2} =$$

$$= \alpha \sqrt{\sum_{k=1}^d (\zeta_k - \frac{\sum_{k=1}^d \zeta_k}{d})^2} \ge \alpha \cdot |\zeta_1 - \frac{\sum_{k=1}^d \zeta_k}{d}| \ge \alpha \cdot \frac{1}{2d}$$

Hence, we meet the second part of the second claim with $D_-(r)$ defined as

$$D_-(r) := \frac{1}{2d} \cdot \exp(-N \cdot t_1(\frac{r}{4})) > 0$$

Note that the proof for the reference case is identical. $\qquad\square$

We note the following remark which deals with the value of points in a sufficiently small sphere around $\mathbf{s}$:

**Remark 4.** *Let $\mu > 0$. The objective $\ell$ is continuous and $\ell(\mathbf{s}) > 0$ there exists $\overline{r}(\mu) > 0$ such that any $A \in \overline{B_{\overline{r}(\mu)}}(\mathbf{s})$ satisfies*

$$\ell(A) \le (1 + \frac{\mu}{4}) \cdot \ell(\mathbf{s})$$

We conclude this section by proving the following corollary, which states that when $A(0) \in \mathcal{I}_3$, a set of additional properties are satisfied:

**Corollary 4.** *Let $\mu > 0$. Consider $\widetilde{r}(\mu) := \min\{r_2, \overline{r}(\mu)\}$ for the respective $r_2$ and $\overline{r}(\mu)$ of Proposition 8 and Remark 4. Suppose we initialize at $A(0) \in \mathcal{I}_3(\frac{\widetilde{r}}{4})$ and at $A^{ref}(0)$, and evolve $A(t)$ and $A^{ref}(t)$ according to Equation (28). There exist constants $D_+(\mu), D_-(\mu) > 0$ such that:*

- $A\big(t_1(\frac{\widetilde{r}(\mu)}{4})\big), A^{ref}\big(t_1(\frac{\widetilde{r}(\mu)}{4})\big) \in \overline{B_{\frac{r_2}{2}}}(\mathbf{s})$

- $|\beta_2\big(t_1(\frac{\widetilde{r}(\mu)}{4})\big)|, |\beta_2^{ref}\big(t_1(\frac{\widetilde{r}(\mu)}{4})\big)| \in [\alpha \cdot D_-(\mu), \alpha \cdot D_+(\mu)]$

- $\ell\bigg(A\big(t_1(\frac{\widetilde{r}(\mu)}{4})\big)\bigg), \ell\bigg(A^{ref}\big(t_1(\frac{\widetilde{r}(\mu)}{4})\big)\bigg) \le (1 + \frac{\mu}{4})\ell(\mathbf{s})$

*Proof.* We consider the constants $D_+(\mu) := D_+(\widetilde{r}(\mu))$ and $D_-(\mu) := D_-(\widetilde{r}(\mu))$ from Proposition 9. Per Proposition 9 and since $A(0) \in \mathcal{I}_3(\frac{\widetilde{r}(\mu)}{4})$, we have that:

- $A\big(t_1(\frac{\widetilde{r}(\mu)}{4})\big), A^{ref}\big(t_1(\frac{\widetilde{r}(\mu)}{4})\big) \in \overline{B_{\frac{\widetilde{r}(\mu)}{2}}}(\mathbf{s})$

- $|\beta_2\big(t_1(\frac{\widetilde{r}(\mu)}{4})\big)|, |\beta_2^{ref}\big(t_1(\frac{\widetilde{r}(\mu)}{4})\big)| \in [\alpha \cdot D_-(\mu), \alpha \cdot D_+(\mu)]$

As $\widetilde{r}(\mu) \le r_2, \overline{r}(\mu)$, we immediately obtain that

$$A\big(t_1(\frac{\widetilde{r}(\mu)}{4})\big), A^{ref}\big(t_1(\frac{\widetilde{r}(\mu)}{4})\big) \in \overline{B_{\frac{r_2}{2}}}(\mathbf{s})$$

and

$$A\big(t_1(\frac{\widetilde{r}(\mu)}{4})\big), A^{ref}\big(t_1(\frac{\widetilde{r}(\mu)}{4})\big) \in \overline{B_{\frac{\overline{r}(\mu)}{2}}}(\mathbf{s})$$

Finally, recall Remark 4 which combined with the latter argument results in

$$\ell\bigg(A\big(t_1(\frac{\widetilde{r}(\mu)}{4})\big)\bigg), \ell\bigg(A^{ref}\big(t_1(\frac{\widetilde{r}(\mu)}{4})\big)\bigg) \le (1 + \frac{\mu}{4})\ell(\mathbf{s})$$

as required. $\qquad\square$

### D.3.5 ESCAPE FROM THE SADDLE s

In the previous section, we showed that the gradient flow trajectories must reach a sufficiently small sphere around $\mathbf{s}$. Our goal in this section is showing that not only do both trajectories escape it, but they also do it fast enough[13]. To begin this section, we prove the following three lemmas regarding the diffeomorphism $H$ from Proposition 7. The following lemma proves that $W_1$ is mapped into itself under $H$:

**Lemma 14.** *Let $A \in \overline{B_{r_2}}(\mathbf{s}) \setminus \{\mathbf{s}\}$ and denote $\widetilde{A} := H(A)$. If $A \in W_1$ then $\widetilde{A} \in W_1$.*

*Proof.* Since $A \in W_1$ there exists $a \in \overline{B_{r_2}}(\mathbf{s})$ such that $A = a \cdot \mathbf{1}_d$. Per Proposition 8 and Lemma 8, it holds that $r_2 < r_1 \le \frac{1}{2d}$ and $s \in [\frac{1}{d}, \frac{3}{d}]$. Thus we obtain that $a \in [0, \frac{4}{d}]$. Assume on the contrary that $\widetilde{A} \notin W_1$. On the one hand, if we initialize at $A(0) = a \cdot \mathbf{1}_d$ and evolve $A(t)$ according to Equation (28), then per Lemma 13 $A(t) \in \overline{B_{r_2}}(\mathbf{s})$ for all $t \ge 0$, and furthermore $\lim_{t \to \infty} A(t) = \mathbf{s}$. By continuity $H(A(t))$ converges to s as well. On the other hand, if we initialize at $\widetilde{A}(0) = \widetilde{A}$ and evolve $\widetilde{A}(t)$ according to the linear approximation around $\mathbf{s}$ of the gradient flow dynamics (see Lemma 11), then per Remark 2 the solution $\widetilde{A}(t)$ diverges away from $\mathbf{s}$ (since the projection of $\widetilde{A}$ to $W_2$ is not zero). This contradicts our assumption that H is a conjugation (see Definition 5). $\qquad\square$

The following lemma proves the existence of two points in $W_1$ that are mapped by $H$ to "opposite sides" of $\mathbf{s}$:

**Lemma 15.** *There exists $a_1, a_2 \in [s - \frac{r_2}{2\sqrt{d}}, s + \frac{r_2}{2\sqrt{d}}] \setminus \{s\}$ such that there exist $\widetilde{a}_1 \in [s - \frac{r_3}{\sqrt{d}}, s)$ and $\widetilde{a}_2 \in (s, s + \frac{r_3}{\sqrt{d}}]$ for which either*

$$H(a_1 \cdot \mathbf{1}_d) = \widetilde{a}_1 \cdot \mathbf{1}_d, \ H(a_2 \cdot \mathbf{1}_d) = \widetilde{a}_2 \cdot \mathbf{1}_d$$

*or*

$$H(a_1 \cdot \mathbf{1}_d) = \widetilde{a}_2 \cdot \mathbf{1}_d, \ H(a_2 \cdot \mathbf{1}_d) = \widetilde{a}_1 \cdot \mathbf{1}_d$$

*Proof.* Consider $a_1 = s - \frac{r_2}{4\sqrt{d}}$ and $a_2 = s + \frac{r_2}{4\sqrt{d}}$. Both $a_1 \cdot \mathbf{1}_d$ and $a_2 \cdot \mathbf{1}_d$ are within $W_1 \cap \overline{B_{r_2}}(\mathbf{s})$ and so by Proposition 8 and Lemma 14 it holds that $H(a_1 \cdot \mathbf{1}_d), H(a_2 \cdot \mathbf{1}_d) \in W_1 \cap \overline{B_{r_3}}(\mathbf{s})$. Thus we can denote $H(a_1 \cdot \mathbf{1}_d) = \widetilde{a}_1 \cdot \mathbf{1}_d$ and $(a_2 \cdot \mathbf{1}_d) = \widetilde{a}_2 \cdot \mathbf{1}_d$ for some $\widetilde{a}_1, \widetilde{a}_2 \in [s - \frac{r_3}{\sqrt{d}}, s + \frac{r_3}{\sqrt{d}}]$. $\widetilde{a}_1, \widetilde{a}_2$ are distinct and different than $s$ since $a_1, a_2 \ne s$ are distinct and since $H$ is a homeomorphism with $H(\mathbf{s}) = \mathbf{s}$. Assume WLOG that $\widetilde{a}_1 < \widetilde{a}_2$ (otherwise we flip the indices). Assume on the contrary that $\widetilde{a}_1, \widetilde{a}_2 > s$ (the case where $\widetilde{a}_1, \widetilde{a}_2 < s$ is symmetric). Per Remark 2, if we initialize at $\widetilde{A}(0) = \widetilde{a}_2 \cdot \mathbf{1}_d$ and evolve $\widetilde{A}(t)$ according to the linear approximation around $\mathbf{s}$ of the gradient flow dynamics, then our trajectory (which converges to $\mathbf{s}$) must reach $\widetilde{a}_1 \cdot \mathbf{1}_d$ after some finite time $t_2$, *i.e.* we obtain $\widetilde{A}(t_2) = \widetilde{a}_1 \cdot \mathbf{1}_d$. Thus we obtain that

$$H^{-1}(\widetilde{A}(t_2)) = H^{-1}(\widetilde{a}_1 \cdot \mathbf{1}_d) = H^{-1}(H(a_1 \cdot \mathbf{1}_d)) = a_1 \cdot \mathbf{1}_d$$

Hence, per Proposition 7 if we initialize at $A(0) = a_2 \cdot \mathbf{1}_d$ and evolve $A(t)$ according to the gradient flow dynamics, we would get that

$$A(t_2) = H^{-1}(\widetilde{A}(t_2)) = a_1 \cdot \mathbf{1}_d$$

The proof concludes by noting that the above is a contradiction to Lemma 13. $\qquad\square$

The following lemma proves that $W_1$ is mapped into itself under $H^{-1}$:

**Lemma 16.** *Let $\widetilde{A} \in \overline{B_{r_3}}(\mathbf{s}) \setminus \{\mathbf{s}\}$ and denote $A := H^{-1}(\widetilde{A})$. If $\widetilde{A} \in W_1$ then $A \in W_1$.*

*Proof.* Since $\widetilde{A} \in W_1 \cap \overline{B_{r_3}}(\mathbf{s})$ there exists $\widetilde{a} \in [s - r_3, s + r_3] \setminus \{s\}$ such that $\widetilde{A} = \widetilde{a} \cdot \mathbf{1}_d$. Assume WLOG that $\widetilde{a} \in [s - r_3, s)$ (the opposite case is symmetric). Per Lemma 15, there exists $a' \in [s - \frac{r_2}{2\sqrt{d}}, s + \frac{r_2}{2\sqrt{d}}] \setminus \{s\}$ such that there exists $\widetilde{a'} \in [s - \frac{r_3}{\sqrt{d}}, s)$ for which

$$H(a' \cdot \mathbf{1}_d) = \widetilde{a'} \cdot \mathbf{1}_d$$

---

[13] recall that the divergence between the two trajectories depends on the convergence time achieved by gradient flow.

Assume that $\widetilde{a} \leq \widetilde{a'}$. By Lemma 11, if we initialize at $\widetilde{A}(0) = \widetilde{A}$ and evolve $\widetilde{A}(t)$ according to the linear approximation around $\mathbf{s}$ of the gradient flow dynamics, then our trajectory (which converges to $\mathbf{s}$) must reach $\widetilde{a'} \cdot \mathbf{1}_d$ after some finite time $t_2$, *i.e.* we obtain $\widetilde{A}(t_2) = \widetilde{a'} \cdot \mathbf{1}_d$. Thus we obtain that

$$H^{-1}(\widetilde{A}(t_2)) = H^{-1}(\widetilde{a'} \cdot \mathbf{1}_d) = H^{-1}(H(a' \cdot \mathbf{1}_d)) = a' \cdot \mathbf{1}_d$$

Hence, per Proposition 7 if we initialize at $A(0) = A = H^{-1}(\widetilde{A})$ and evolve $A(t)$ according to the gradient flow dynamics, we would get that

$$A(t_2) = H^{-1}(\widetilde{A}(t_2)) = a' \cdot \mathbf{1}_d$$

Invoking Lemma 3 we conclude that $A \in \mathcal{W}_1$. Assume that $\widetilde{a} > \widetilde{a'}$. By Lemma 11, if we initialize at $\widetilde{A'}(0) = \widetilde{A'}$ and evolve $\widetilde{A'}(t)$ according to the linear approximation around $\mathbf{s}$ of the gradient flow dynamics, then our trajectory (which converges to $\mathbf{s}$) must reach $\widetilde{a} \cdot \mathbf{1}_d$ after some finite time $t_2$, *i.e.* we obtain $\widetilde{A'}(t_2) = \widetilde{a} \cdot \mathbf{1}_d = \widetilde{A}$. On the one hand, note that $H^{-1}(\widetilde{A}) = A$. On the other hand, if we initialize at $A'(0) = a' \cdot \mathbf{1}_d$ and evolve $A'(t)$ according to the gradient flow dynamics (defined in Equation (24)), we get by Proposition 7 that $A'(t_2) = H^{-1}(\widetilde{A'}(t_2))$. Thus, $A'(t_2) = A$. The proof concludes by invoking Lemma 13 which states that $A'(t) \in \mathcal{W}_1$ for any $t \geq 0$. $\qquad\square$

The following two lemmas give bounds on the original dynamics in terms of the linearized ones:

**Lemma 17.** *Let $A \in \overline{B_{r_2}}(\mathbf{s}) \setminus \{\mathbf{s}\}$. It holds that*

$$Dist(A, \mathcal{W}_1) \leq \widetilde{G} \cdot Dist(H(A), \mathcal{W}_1)$$

*where $\widetilde{G}$ is the Lipschitz coefficient of $H^{-1}|_{\overline{B_{r_3}}(\mathbf{s})}$.*

*Proof.* First note that per Proposition 8, $H^{-1}|_{\overline{B_{r_3}}(\mathbf{s})}$ is indeed Lipschitz. Let $\widetilde{G} > 0$ be its Lipschitz coefficient. Next, by definition of the Dist measure (Equation (15)) we have that

$$\mathrm{Dist}(H(A), \mathcal{W}_1) = \min_{\widetilde{A} \in \mathcal{W}_1} \|H(A) - \widetilde{A}\|_2$$

Since $A \in \overline{B_{r_2}}(\mathbf{s})$, it holds by Proposition 8 that $H(A) \in \overline{B_{r_3}}(\mathbf{s})$. Thus, since $\overline{B_{r_3}}(\mathbf{s})$ is a ball it must hold that

$$\min_{\widetilde{A} \in \mathcal{W}_1} \|H(A) - \widetilde{A}\|_2 = \min_{\widetilde{A} \in \mathcal{W}_1 \cap \overline{B_{r_3}}(\mathbf{s})} \|H(A) - \widetilde{A}\|_2$$

As $H$ is onto $\overline{B_{r_3}}(\mathbf{s})$, by Proposition 8 there exists $A' \in \overline{B_{r_1}}(\mathbf{s})$ such that

$$H(A') \in \operatorname*{argmin}_{\widetilde{A} \in \mathcal{W}_1 \cap \overline{B_{r_3}}(\mathbf{s})} \|H(A) - \widetilde{A}\|_2$$

Hence by the Lipschitz property of of $H^{-1}$ we obtain

$$\min_{\widetilde{A} \in \mathcal{W}_1 \cap \overline{B_{r_3}}(\mathbf{s})} \|H(A) - \widetilde{A}\|_2 = \|H(A) - H(A')\|_2 \geq$$

$$\geq \frac{1}{\widetilde{G}} \|H^{-1}(H(A)) - H^{-1}(H(A'))\|_2 = \frac{1}{\widetilde{G}} \|A - A'\|_2$$

Since $A' = H^{-1}(\widetilde{A})$ for some $\widetilde{A} \in \mathcal{W}_1 \cap \overline{B_{r_3}}(\mathbf{s})$, it holds by Lemma 16 that $A' \in \mathcal{W}_1$. Thus,

$$\frac{1}{\widetilde{G}} \|A - A'\|_2 \geq \frac{1}{\widetilde{G}} \min_{A'' \in \mathcal{W}_1} \|A - A''\|_2 = \frac{1}{\widetilde{G}} \cdot \mathrm{Dist}(A, \mathcal{W}_1)$$

Combining the above inequalities and multiplying by $\widetilde{G}$ we obtain overall that

$$\widetilde{G} \cdot \mathrm{Dist}(H(A), \mathcal{W}_1) \geq \|A - A'\|_2 \geq \mathrm{Dist}(A, \mathcal{W}_1)$$

as required. $\qquad\square$

**Lemma 18.** *Let* $A \in \overline{B_{r_1}}(\mathbf{s})$. *It holds that*

$$\text{Dist}(H(A), \mathcal{W}_1) \leq G \cdot \text{Dist}(A, \mathcal{W}_1)$$

*where* $G$ *is the Lipschitz coefficient of* $H|_{\overline{B_{r_1}}(\mathbf{s})}$.

*Proof.* First note that per Proposition 8, $H|_{\overline{B_{r_1}}(\mathbf{s})}$ is indeed Lipschitz. Let $G > 0$ be its Lipschitz coefficient . Next, by definition of the Dist measure we have that

$$\text{Dist}(A, \mathcal{W}_1) = \min_{A' \in \mathcal{W}_1} \|A - A'\|_2$$

Since $A \in \overline{B_{r_1}}(\mathbf{s})$ and $\overline{B_{r_1}}(\mathbf{s})$ is a ball, it must hold that

$$\min_{A' \in \mathcal{W}_1} \|A - A'\|_2 = \min_{A' \in \mathcal{W}_1 \cap \overline{B_{r_1}}(\mathbf{s})} \|A - A'\|_2$$

By the Lipschitz property of $H$ we obtain

$$\min_{A' \in \mathcal{W}_1 \cap \overline{B_{r_1}}(\mathbf{s})} \|A - A'\|_2 \geq \min_{A' \in \mathcal{W}_1 \cap \overline{B_{r_1}}(\mathbf{s})} \frac{1}{G} \|H(A) - H(A')\|_2 \geq \text{Dist}(H(A), \mathcal{W}_1)$$

where the last inequality follows from Lemma 14. Multiplying by $G$ gives the result. $\qquad\square$

Before proving the main claims of this section, we introduce another condition on the initialization which we denote $\mathcal{I}_4$:

**Definition 11.** Let $\mu > 0$. We denote $G' := \max\{1, G, \widetilde{G}\}$ for $\widetilde{G}$ and $G$ from Lemmas 17 and 18. We use $\mathcal{I}_4(\mu)$ to denote the following subset of $\mathcal{I}_0$:

$$\mathcal{I}_4(\mu) := \left\{ A \in \mathcal{I}_0 : \alpha \leq \frac{r_3}{4 \max\{2, \exp(-2\lambda_-)\} \cdot G'^2 \sqrt{d} D_+(\mu)} \right\}$$

For $r_3$, $\lambda_-$ and $D_+$ from Proposition 8, Lemma 9, , and Corollary 4 respectively.

In the next two propositions, we bound the time it takes to escape the sphere around $\mathbf{s}$ under the lin-earized dynamics, and prove an additional claim that will be utilized later to show that the trajectory never returns to a certain sphere around $\mathbf{s}$.

We introduce notation which will be used in both propositions; Let $\mu > 0$. Suppose we initialize at $A(0) \in \mathcal{I}_3(\frac{\widetilde{r}(\mu)}{4}) \cap \mathcal{I}_4(\mu)$ (for $\widetilde{r}$ of Corollary 4) and at $A^{ref}(0)$, and evolve $A(t)$ and $A^{ref}(t)$ according to Equation (28). Suppose we initialize $\widetilde{A}(0) = H\left( A\big(t_1(\frac{\widetilde{r}(\mu)}{4})\big) \right)$ and at $\widetilde{A^{ref}}(0) = H\left( A^{ref}\big(t_1(\frac{\widetilde{r}(\mu)}{4})\big) \right)$ (for $t_1$ of Lemma 13), and evolve $\widetilde{A}(t)$ and $\widetilde{A^{ref}}(t)$ according to the linearized dynamics (see Lemma 11). For any time $t \geq 0$, denote the representations of $\widetilde{A}(t)$ and $\widetilde{A^{ref}}(t)$ with the orthogonal subspaces $\mathcal{W}_1$ and $\mathcal{W}_2$ to be

$$\widetilde{A}(t) = \widetilde{\beta_1}(t) \cdot \mathbf{1}_d + \widetilde{\beta_2}(t) \cdot \widetilde{\mathbf{v}}(t)$$

and

$$\widetilde{A^{ref}}(t) = \widetilde{\beta_1^{ref}}(t) \cdot \mathbf{1}_d + \widetilde{\beta_2^{ref}}(t) \cdot \widetilde{\mathbf{v}^{ref}}(t)$$

where $\widetilde{\mathbf{v}}(t), \widetilde{\mathbf{v}^{ref}}(t) \in \mathcal{W}_2$ are unit vectors.

The following proposition give quantitative bounds on the rate of exponential escape from $\mathbf{s}$ of trajectories under the lineaarized dynamics:

**Proposition 10.** *There exist times* $t_2(\mu), t_2^{ref}(\mu) \geq 2$ *for which it holds*

- $|\widetilde{\beta_2}\big(t_2(\mu)\big)|, |\widetilde{\beta_2^{ref}}\big(t_2(\mu)\big)| = \frac{r_3}{2\sqrt{d}}$

- *For $G' := \max\{1, G, \widetilde{G}\}$ it holds that*

$$t_2(\mu), t_2^{ref}(\mu) \in \left[-\frac{1}{\lambda_-}\ln\left(\frac{r_3}{4G'^2\sqrt{d}\alpha D_+(\mu)}\right), -\frac{1}{\lambda_-}\ln\left(\frac{G'^2 r_3}{2\sqrt{d}\alpha D_-(\mu)}\right)\right]$$

- *For any $t \in [0, t_2(\mu)]$ it holds that $\widetilde{A}(t) \in \overline{B_{r_3}}(\mathbf{s})$*

- *For any $t \in [0, t_2^{ref}(\mu)]$ it holds that $\widetilde{A^{ref}}(t) \in \overline{B_{r_3}}(\mathbf{s})$*

*Proof.* We prove the argument for $\widetilde{A}$ (the proof is identical for $\widetilde{A^{ref}}$). Recall Corollary 4 which states that

$$\text{Dist}\left(A\left(t_1\left(\frac{\widetilde{r}(\mu)}{4}\right)\right), \mathcal{W}_1\right) = |\beta_2\left(t_1\left(\frac{\widetilde{r}(\mu)}{4}\right)\right)| \in [\alpha \cdot D_-(\mu), \alpha \cdot D_+(\mu)]$$

Thus, applying Lemmas 17 and 18 we obtain

$$\frac{\alpha \cdot D_-(\mu)}{\widetilde{G}} \leq \text{Dist}(\widetilde{A}(0), \mathcal{W}_1) = |\widetilde{\beta_2}(0)| \leq G \cdot \alpha \cdot D_+(\mu)$$

Per Lemma 11, for any $t \geq 0$ the solution at time $t$ to the linear dynamics initialized at $\widetilde{A}(0)$ is given by

$$\left(\exp(-t \cdot \lambda_+)(\widetilde{\beta_1}(0) - s) + s\right)\mathbf{1}_d + (\exp(-t \cdot \lambda_-) \cdot \widetilde{\beta_2}(0))\widetilde{\mathbf{v}}(0)$$

As noted in Remark 2, the coefficient $|\widetilde{\beta_1}(t) - s|$ tends to zero as $t$ grows, while the coefficient $|\widetilde{\beta_2}(t)|$ tends to $\infty$ as $t$ grows. We first bound the time $t_2(\mu)$ for which $|\widetilde{\beta_2}\left(t_2(\mu)\right)| = \frac{r_3}{2\sqrt{d}}$. Since $A(0) \in \mathcal{I}_4(\mu)$, it holds that

$$\max\{2, \exp(-2\lambda_-)\} \leq \frac{r_3}{4G'^2\sqrt{d}\alpha D_+(\mu)}$$

Therefore since $\lambda_- < 0$ (by Lemma 10) we obtain the following positive time $t_-$:

$$t_- := -\frac{1}{\lambda_-}\ln\left(\frac{r_3}{4G'^2\sqrt{d}\alpha D_+(\mu)}\right) \geq -\frac{1}{\lambda_-}\ln(\exp(-2\lambda_-)) = 2$$

Thus, at time $t_-$ the solution to the linear dynamics satisfies the following:

$$\text{Dist}(\widetilde{A}(t_-), \mathcal{W}_1) = |\widetilde{\beta_2}(t_-)| = |\widetilde{\beta_2}(0)| \cdot \exp(-t_- \cdot \lambda_-) \leq G \cdot D_+(\mu) \cdot \alpha \cdot \exp(-t_- \cdot \lambda_-) =$$

$$= G \cdot D_+(\mu) \cdot \alpha \cdot \exp\left(\frac{\lambda_-}{\lambda_-}\ln\left(\frac{r_3}{4G'^2\sqrt{d}\alpha D_+(\mu)}\right)\right) = \frac{G \cdot D_+(\mu) \cdot \alpha \cdot r_3}{4G'^2\sqrt{d}\alpha D_+(\mu)} \leq \frac{r_3}{4G'\sqrt{d}} \leq \frac{r_3}{4\sqrt{d}}$$

where the last two inequalities stem from the fact that $G' \geq G, 1$. Hence, $t_-$ is a lower bound on $t_2(\mu)$. On the other hand, note that

$$\frac{G'^2 \cdot r_3}{2\sqrt{d}\alpha D_-(\mu)} \geq \frac{r_3}{4G'^2\sqrt{d}\alpha D_+(\mu)}$$

and so since $\lambda_- < 0$ we obtain the following positive time $t_+$:

$$t_+ := -\frac{1}{\lambda_-}\ln\left(\frac{G'^2 \cdot r_3}{2\sqrt{d}\alpha D_-(\mu)}\right) \geq -\frac{1}{\lambda_-}\ln\left(\frac{r_3}{4G'^2\sqrt{d}\alpha D_+(\mu)}\right) = t_-$$

Thus, at time $t_+$ the solution to the linear dynamics statisfies the following:

$$|\widetilde{\beta_2}(t_+)| = |\widetilde{\beta_2}(0)| \cdot \exp(-t_+ \cdot \lambda_-) \geq \frac{\alpha \cdot D_-(\mu)}{\widetilde{G}} \cdot \exp(-t_+ \cdot \lambda_-) =$$

$$= \frac{\alpha \cdot D_-(\mu)}{\widetilde{G}} \cdot \exp\left(\frac{\lambda_-}{\lambda_-}\ln\left(\frac{G'^2 \cdot r_3}{2\sqrt{d}\alpha D_-(\mu)}\right)\right) =$$

$$= \frac{\alpha \cdot D_-(\mu)}{\widetilde{G}} \cdot \frac{G'^2 \cdot r_3}{2\sqrt{d}\alpha D_-(\mu)} \geq \frac{G' \cdot r_3}{2\sqrt{d}} \geq \frac{r_3}{2\sqrt{d}}$$

where the last two inequalities stem from the fact that $G' \geq \widetilde{G}, 1$. Hence, $t_+$ is an upper bound on $t_2(\mu)$. Next, we show that $|\widetilde{\beta_1}(t_-) - s| \leq \frac{r_3}{2\sqrt{d}}$. This will allow us to claim by monotonicity that $\widetilde{A}(t_2(\mu)) \in \overline{B_{r_3}}(\mathbf{s})$, since then we'll have the following:

$$\|\widetilde{A}(t_2(\mu)) - \mathbf{s}\|_2 = \|\widetilde{\beta_1}(t_2(\mu)) \cdot \mathbf{1}_d + \widetilde{\beta_2}(t_2(\mu)) \cdot \mathbf{v}(t_2(\mu)) - \mathbf{s}\|_2 =$$
$$= \|(\widetilde{\beta_1}(t_2(\mu)) - s) \cdot \mathbf{1}_d\|_2 + \|\widetilde{\beta_2}(t_2(\mu)) \cdot \mathbf{v}(t_2(\mu))\|_2 \leq$$
$$\leq \frac{r_3}{2\sqrt{d}} \cdot \sqrt{d} + \frac{r_3}{2\sqrt{d}} \cdot 1 \leq r_3$$

Since $\widetilde{A}(0) \in \overline{B_{r_3}}(\mathbf{s})$ it holds that $|\widetilde{\beta_1}(0) - s| \leq \frac{r_3}{\sqrt{d}}$. Hence, since $\frac{\lambda_+}{\lambda_-} < -1$ (by Lemma 10) it holds that

$$|\widetilde{\beta_1}(t_-) - s| = |\exp(-\lambda_+ \cdot t_-)(\widetilde{\beta_1}(0) - s) + s - s| \leq$$
$$\leq |\widetilde{\beta_1}(0) - s| \cdot \exp(-\lambda_+ \cdot t_-) \leq$$
$$\leq \frac{r_3}{\sqrt{d}} \cdot \left(\frac{r_3}{4G'^2\sqrt{d}\alpha D_+(\mu)}\right)^{\frac{\lambda_+}{\lambda_-}} \leq \frac{r_3}{2\sqrt{d}}$$

where the last inequality stems from the fact that $2 \leq \frac{r_3}{4G'^2\sqrt{d}\alpha D_+(\mu)}$. Finally, since under the linear dynamics $|\widetilde{\beta_2}(t)|$ monotonically grows and $|\widetilde{\beta_1}(t)|$ monotonically tends to $s$, it must hold that for any $t \in [0, t_2(\mu)]$ we have $\widetilde{A}(t) \in \overline{B_{r_3}}(\mathbf{s})$. $\qquad\square$

The next proposition shows that $\widetilde{\beta_2}(t)$ must be larger than some constant throughout a time interval of length 1 before $t_2(\mu)$:

**Proposition 11.** *For any $\tau \in [0, 1]$ it holds that*

$$|\widetilde{\beta_2}(t_2(\mu) - \tau)|, |\widetilde{\beta_2^{ref}}(t_2(\mu) - \tau)| = \frac{r_3}{2\sqrt{d}} \cdot \exp(\lambda_- \cdot \tau)$$

*Proof.* We prove the argument for $\widetilde{A}$ (the proof is identical for $\widetilde{A^{ref}}$). In Proposition 10 we've established that $t_2(\mu) \geq 2$. Thus, per Lemma 11, for any $\tau \in [0, 1]$ the solution at the positive time $t_2(\mu) - \tau \geq 1$ to the linear dynamics initialized at $\widetilde{A}(0)$ is given by

$$\left(\exp(-(t_2(\mu) - \tau) \cdot \lambda_+)(\widetilde{\beta_1}(0) - s) + s\right)\mathbf{1}_d + (\exp(-(t_2(\mu) - \tau) \cdot \lambda_-) \cdot \widetilde{\beta_2}(0))\widetilde{\mathbf{v}}(0)$$

Hence, since $|\widetilde{\beta_2}(t_2(\mu))| = \frac{r_3}{2\sqrt{d}}$ we obtain that

$$|\widetilde{\beta_2}(t_2(\mu) - \tau)| = |\exp(-(t_2(\mu) - \tau) \cdot \lambda_-) \cdot \widetilde{\beta_2}(0)| =$$
$$= |\exp(-t_2(\mu) \cdot \lambda_-) \cdot \widetilde{\beta_2}(0)| \cdot \exp(\lambda_- \cdot \tau) =$$
$$= |\widetilde{\beta_2}(t_2(\mu))| \cdot \exp(\lambda_- \cdot \tau) = \frac{r_3}{2\sqrt{d}} \cdot \exp(\lambda_- \cdot \tau)$$

$\qquad\square$

The above established that there exists a time which is $\mathcal{O}(\ln(\frac{1}{\alpha}))$ where at least one of the linearized trajectories is at a constant distant from $\mathcal{W}_1$. We complete this section by proving the following corollary, which states that the corresponding non linear dynamics trajectory must also be at a constant distance from $\mathcal{W}_1$ during a time interval of length 1. This will eventually allow us to claim that the trajectory must remain trapped within a set where the objective $\ell$ satistfies satisfies the PL condition (see Definition 13), which in turn ensures a rapid convergence to a global minimum (discussed in the next section):

**Corollary 5.** *Let $\mu > 0$. Suppose we initialize at $A(0) \in \mathcal{I}_3(\frac{\widetilde{r}(\mu)}{4}) \cap \mathcal{I}_4(\mu)$ (for $\widetilde{r}$ of Corollary 4) and at $A^{ref}(0)$, and evolve $A(t)$ and $A^{ref}(t)$ according to Equation (28). For any $\tau \in [0,1]$ it holds that*

$$|\beta_2\big(t_1(\frac{\widetilde{r}(\mu)}{4}) + t_2(\mu) - \tau\big)| = Dist\bigg(A\big(t_1(\frac{\widetilde{r}(\mu)}{4}) + t_2(\mu) - \tau\big), \mathcal{W}_1\bigg) \geq \frac{r_3 \cdot \exp(\lambda_-)}{2G \cdot \sqrt{d}}$$

*and*

$$|\beta_2^{ref}\big(t_1(\frac{\widetilde{r}(\mu)}{4}) + t_2^{ref}(\mu) - \tau\big)| = Dist\bigg(A^{ref}\big(t_1(\frac{\widetilde{r}(\mu)}{4}) + t_2^{ref}(\mu) - \tau\big), \mathcal{W}_1\bigg) \geq \frac{r_3 \cdot \exp(\lambda_-)}{2G \cdot \sqrt{d}}$$

*Proof.* We prove the argument for $A$ (the proof is identical for $A^{ref}$). In Proposition 11 we have shown that for any $\tau \in [0,1]$ it holds that $\widetilde{A}\big(t_2(\mu) - \tau\big) \in \overline{B_{r_3}}(\mathbf{s})$, and so the mapping $H$ is a conjugation to the original dynamics. Therefore since we've initialized $\widetilde{A}(0)$ at $H\bigg(A\big(t_1(\frac{\widetilde{r}(\mu)}{4})\big)\bigg)$, it must hold that for any $\tau \in [0,1]$

$$H^{-1}\bigg(\widetilde{A}\big(t_2(\mu) - \tau\big)\bigg) = A\big(t_1(\frac{\widetilde{r}(\mu)}{4}) + t_2(\mu) - \tau\big)$$

Since $H^{-1}[\overline{B_{r_3}}] \subseteq \overline{B_{r_1}}$ we obtain that $A\big(t_1(\frac{\widetilde{r}(\mu)}{4}) + t_2(\mu) - \tau\big) \in \overline{B_{r_1}}$, and thus by Lemma 18 we obtain

$$\mathrm{Dist}(A\big(t_1(\frac{\widetilde{r}(\mu)}{4}) + t_2(\mu) - \tau\big), \mathcal{W}_1) \geq \frac{\mathrm{Dist}(\widetilde{A}\big(t_2(\mu) - \tau\big), \mathcal{W}_1)}{G}$$

Note that by orthogonoality $|\widetilde{\beta_2}\big(t_2(\mu) - \tau\big)| = dist(\widetilde{A}\big(t_2(\mu) - \tau\big), \mathcal{W}_1)$, hence plugging Proposition 11 we receive

$$\mathrm{Dist}(A\big(t_1(\frac{\widetilde{r}(\mu)}{4}) + t_2(\mu) - \tau\big), \mathcal{W}_1) \geq \frac{|\widetilde{\beta_2}\big(t_2(\mu) - \tau\big)|}{G} = \frac{r_3 \cdot \exp(\lambda_- \cdot \tau)}{2G \cdot \sqrt{d}} \geq \frac{r_3 \cdot \exp(\lambda_-)}{2G \cdot \sqrt{d}}$$

where the last inequality is due to $\lambda_- < 0$. The proof is complete by observing that $\mathrm{Dist}(A\big(t_1(\frac{\widetilde{r}(\mu)}{4}) + t_2(\mu) - \tau\big), \mathcal{W}_1) = |\beta_2\big(t_1(\frac{\widetilde{r}(\mu)}{4}) + t_2(\mu) - \tau\big)|$. $\qquad\square$

#### D.3.6 CONVERGENCE TO A GLOBAL MINIMUM

We begin this section by proving the following corollary regarding the difference between different coordinates of the points reached by the gradient flow trajectories. We will later that this ensures the objective satisfies the PL condition (Definition 13).

**Corollary 6.** *Let $\mu > 0$. Suppose we initialize at $A(0) \in \mathcal{I}_3(\frac{\widetilde{r}(\mu)}{4}) \cap \mathcal{I}_4(\mu)$ (for $\widetilde{r}$ of Corollary 4) and at $A^{ref}(0)$, and evolve $A(t)$ and $A^{ref}(t)$ according to Equation (28). For any $\tau \in [0,1]$ it holds that there exist $i, j, i^{ref}, j^{ref} \in [d]$ such that*

$$|a_i\big(t_1(\frac{\widetilde{r}(\mu)}{4}) + t_2(\mu) - \tau\big) - a_j\big(t_1(\frac{\widetilde{r}(\mu)}{4}) + t_2(\mu) - \tau\big)| \geq \frac{r_3 \cdot \exp(\lambda_-)}{2G \cdot d^{1.5}}$$

*and*

$$|a_{i^{ref}}^{ref}\big(t_1(\frac{\widetilde{r}(\mu)}{4}) + t_2^{ref}(\mu) - \tau\big) - a_{j^{ref}}^{ref}\big(t_1(\frac{\widetilde{r}(\mu)}{4}) + t_2^{ref}(\mu) - \tau\big)| \geq \frac{r_3 \cdot \exp(\lambda_-)}{2G \cdot d^{1.5}}$$

*Proof.* The claim follows from invoking Lemma 32 and plugging the lower bound on $|\beta_2|$ and $|\beta_2^{ref}|$ provided in Corollary 5. $\qquad\square$

We continue by proving that $\ell$ satisfies the PL condition (Definition 13) on the subset of points $\mathrm{Diff}(b)^{\mathcal{C}}$ (Equation (18)):

**Lemma 19.** *Let $b > 0$. $\ell|_{Diff(b)^c \cap [-3,3]^d}$ satisfies the PL condition with PL coefficient $\mu = \frac{1}{2} \min\bigg\{2d, \frac{((L-1)\widetilde{b})^2}{4}, (\frac{\widehat{b}}{\widehat{b}+2})^2\bigg\}$, for $\widetilde{b} := (\frac{b}{2})^{L-2}$ and $\widehat{b} := (\frac{b}{6})^{L-2}$.*

*Proof.* Let $A \in \text{Diff}(b)^{\mathcal{C}} \cap [-3,3]^d$. Recalling the definition of $\text{Diff}(b)^{\mathcal{C}}$ (Equation (18)) there exist $i,j \in [d]$ such that $|a_i - a_j| \geq b$. Since $a_i \neq a_j$, at least one is non-zero. Assume WLOG that $a_j \neq 0$. Hence, since $0 < L - 2 \in \mathbb{N}_{odd}$ we get per proposition 17 that

$$|a_i^{L-2} - a_j^{L-2}| \geq (\frac{b}{2})^{L-2} =: \widetilde{b}$$

$$|1 - \frac{a_i^{L-2}}{a_j^{L-2}}| \geq (\frac{b}{6})^{L-2} =: \widehat{b}$$

Denote $res_s := 1 - \sum_{k=1}^d a_k$ and $res_l := 1 - \sum_{k=1}^d a_k^{L-1}$. If $res_s = 0$, then it holds that

$$\|\nabla \ell(A)\|^2 = \sum_{k=1}^d (L-1)^2 (a_k^{L-2})^2 res_l^2 = (*)$$

By the triangle inequality, either $|a_i^{L-2}| \geq \frac{\widetilde{b}}{2}$ or $|a_j^{L-2}| \geq \frac{\widetilde{b}}{2}$. In either case,

$$(*) \geq (L-1)^2 \frac{\widetilde{b}^2}{4} res_l^2 = \frac{((L-1)\widetilde{b})^2}{4}(\frac{1}{2}res_l^2 + \frac{1}{2}res_s^2) = \frac{((L-1)\widetilde{b})^2}{4}\ell(A)$$

*i.e.*, the PL condition is satisfied with $\mu = \frac{1}{2} \cdot \frac{((L-1)\widetilde{b})^2}{4}$. If $res_l = 0$, then it holds that

$$\|\nabla \ell(A)\|^2 = \sum_{k=1}^d res_s^2 = d res_s^2 = 2d(\frac{1}{2}res_l^2 + \frac{1}{2}res_s^2) = 2d\ell(A)$$

*i.e.* the PL condition is satisfied with $\mu = \frac{1}{2} \cdot 2d$. Assume $res_l, res_s \neq 0$ and denote $\chi := \frac{-res_s}{(L-1)res_l} \neq 0$. For any $k \in [d]$ we have

$$\nabla \ell(A)_k = (L-1)a_k^{L-2}res_l + res_s = (L-1)a_k^{L-2}res_l - (L-1)res_l \cdot \chi = (L-1)(a_k^{L-2} - \chi)res_l$$

or equivalently

$$\nabla \ell(A)_k = (L-1)a_k^{L-2}res_l + res_s = -(L-1)a_k^{L-2}\frac{res_s}{(L-1)\chi} + res_s = (1 - \frac{a_k^{L-2}}{\chi})res_s$$

Squaring the above identities we obtain

$$\nabla \ell(A)_k^2 = (L-1)^2(a_k^{L-2} - \chi)^2 res_l^2 = (1 - \frac{a_k^{L-2}}{\chi})^2 res_s^2$$

By the triangle inequality

$$|a_i^{L-2} - \chi| \geq \frac{\widetilde{b}}{2}$$

or

$$|a_j^{L-2} - \chi| \geq \frac{\widetilde{b}}{2}$$

Therefore, if $res_l^2 \geq res_s^2$ we get that

$$\|\nabla \ell(A)\|^2 = \sum_{k=1}^d (L-1)^2(a_k^{L-2} - \chi)^2 res_l^2 \geq \frac{((L-1)\widetilde{b})^2}{4}res_l^2 \geq$$

$$\geq \frac{((L-1)\widetilde{b})^2}{4}(\frac{1}{2}res_l^2 + \frac{1}{2}res_s^2) = \frac{((L-1)\widetilde{b})^2}{4}\ell(A)$$

*i.e.* the PL condition is satisfied with $\mu = \frac{1}{2}\frac{((L-1)\widetilde{b})^2}{4}$. On the other hand if $res_l^2 < res_s^2$, then by proposition 18

$$|1 - \frac{a_i^{L-2}}{\chi}| \geq \frac{\widehat{b}}{\widehat{b}+2}$$

or

$$|1 - \frac{a_j^{L-2}}{\chi}| \geq \frac{\widehat{b}}{\widehat{b}+2}$$

and therefore

$$\|\nabla\ell(A)\|^2 = \sum_{k=1}^{d}(1 - \frac{a_k^{L-2}}{\chi})^2 res_s^2 \geq (\frac{\widehat{b}}{\widehat{b}+2})^2 res_s^2 \geq$$

$$\geq (\frac{\widehat{b}}{\widehat{b}+2})^2(\frac{1}{2}res_l^2 + \frac{1}{2}res_s^2) = (\frac{\widehat{b}}{\widehat{b}+2})^2\ell(A)$$

*i.e.* the PL condition is satisfied with $\mu = \frac{1}{2}(\frac{\widehat{b}}{\widehat{b}+2})^2$. Overall, we get that whenever $A \in \mathrm{Diff}(b)^{\mathcal{C}} \cap$

$[-3,3]^d$, the PL condition is satisfied with $\mu = \frac{1}{2}\min\left\{2d, \frac{((L-1)\widetilde{b})^2}{4}, (\frac{\widehat{b}}{\widehat{b}+2})^2\right\}$, as required. $\qquad\square$

The above results in the following corollary regarding the PL condition satisfied by $\ell$ at a certain set of points reached by the gradient flow trajectories:

**Corollary 7.** *Let $\mu > 0$. Suppose we initialize at $A(0) \in \mathcal{I}_3(\frac{\widetilde{r}(\mu)}{4}) \cap \mathcal{I}_4(\mu)$ (for $\widetilde{r}$ of Corollary 4) and at $A^{ref}(0)$, and evolve $A(t)$ and $A^{ref}(t)$ according to Equation (28). For any $\tau \in [0,1]$ it holds that $\ell$ satisfies the PL condition (Definition 13) at the points*

$$A\big(t_1(\frac{\widetilde{r}(\mu)}{4}) + t_2(\mu) - \tau\big)$$

*and*

$$A^{ref}\big(t_1(\frac{\widetilde{r}(\mu)}{4}) + t_2^{ref}(\mu) - \tau\big)$$

*with PL coefficient*

$$\mu_1 := \frac{1}{2}\min\left\{2d, \frac{((L-1)\widetilde{b})^2}{4}, (\frac{\widehat{b}}{\widehat{b}+2})^2\right\}$$

*for*

$$\widetilde{b} := (\frac{r_3 \cdot \exp(\lambda_-)}{4G \cdot d^{1.5}})^{L-2}, \quad \widehat{b} := (\frac{r_3 \cdot \exp(\lambda_-)}{12G \cdot d^{1.5}})^{L-2}$$

*Proof.* The claim follows from invoking Lemma 19 and plugging the bound on the coordinate difference provided in Corollary 6. $\qquad\square$

For the rest of the proof, we let $\mu = \mu_1$ from Corollary 7. We are now ready to prove the following proposition, which states that at time $t_1(\frac{\widetilde{r}(\mu_1)}{4}) + t_2(\mu_1)$, the trajectory is at a point whose value improves upon the value of $\ell$ at **s** by a constant:

**Proposition 12.** *Consider $\mu_1$ from Corollary 7. Suppose we initialize at $A(0) \in \mathcal{I}_3(\frac{\widetilde{r}(\mu_1)}{4}) \cap \mathcal{I}_4(\mu_1)$ (for $\widetilde{r}$ of Corollary 4) and at $A^{ref}(0)$, and evolve $A(t)$ and $A^{ref}(t)$ according to Equation (28). it holds that*

$$\ell(\mathbf{s}) - \ell\bigg(A\big(t_1(\frac{\widetilde{r}(\mu_1)}{4}) + t_2(\mu_1)\big)\bigg) \geq \min\{\frac{\ell(\mathbf{s})}{2}, \frac{3\mu_1 \cdot \ell(\mathbf{s})}{4}\}$$

*and*

$$\ell(\mathbf{s}) - \ell\bigg(A^{ref}\big(t_1(\frac{\widetilde{r}(\mu_1)}{4}) + t_2^{ref}(\mu_1)\big)\bigg) \geq \min\{\frac{\ell(\mathbf{s})}{2}, \frac{3\mu_1 \cdot \ell(\mathbf{s})}{4}\}$$

*Proof.* We prove the argument for $A$ (the proof is identical for $A^{ref}$). First, suppose that $\ell\left(A\left(t_1\left(\frac{\widetilde{r}(\mu_1)}{4}\right)+t_2(\mu_1)\right)\right)\le\frac{\ell(\mathbf{s})}{2}$. Then it holds that

$$\ell(\mathbf{s})-\ell\left(A\left(t_1\left(\frac{\widetilde{r}(\mu_1)}{4}\right)+t_2(\mu_1)\right)\right)\ge\ell(\mathbf{s})-\frac{\ell(\mathbf{s})}{2}=\frac{\ell(\mathbf{s})}{2}\ge\min\{\frac{\ell(\mathbf{s})}{2},\frac{3\mu_1\cdot\ell(\mathbf{s})}{4}\}$$

Next, suppose that $\ell\left(A\left(t_1\left(\frac{\widetilde{r}(\mu_1)}{4}\right)+t_2(\mu_1)\right)\right)>\frac{\ell(\mathbf{s})}{2}$. Thus per Lemma 23, for any $\tau\in[0,1]$ it holds that

$$\ell\left(A\left(t_1\left(\frac{\widetilde{r}(\mu_1)}{4}\right)+t_2(\mu_1)-\tau\right)\right)\ge\ell\left(A\left(t_1\left(\frac{\widetilde{r}(\mu_1)}{4}\right)+t_2(\mu_1)\right)\right)>\frac{\ell(\mathbf{s})}{2}$$

Next, per Corollary 7 for any $\tau\in[0,1]$ it also holds that $\ell$ satisfies the PL condition in the point $A\left(t_1\left(\frac{\widetilde{r}(\mu_1)}{4}\right)+t_2(\mu_1)-\tau\right)$ with PL coefficient $\mu_1$. Thus, by Lemma 33 it holds that

$$\ell\left(A\left(t_1\left(\frac{\widetilde{r}(\mu_1)}{4}\right)+t_2(\mu_1)-1\right)\right)-\ell\left(A\left(t_1\left(\frac{\widetilde{r}(\mu_1)}{4}\right)+t_2(\mu_1)\right)\right)\ge$$

$$\ge2\left(t_1\left(\frac{\widetilde{r}(\mu_1)}{4}\right)+t_2(\mu_1)-t_1\left(\frac{\widetilde{r}(\mu_1)}{4}\right)-t_2(\mu_1)+1\right)\cdot\mu_1\cdot\frac{\ell(\mathbf{s})}{2}=\mu_1\cdot\ell(\mathbf{s})$$

On the other hand, recall that by Corollary 4 and since gradient flow is non-increasing we have that

$$\ell\left(A\left(t_1\left(\frac{\widetilde{r}(\mu_1)}{4}\right)+t_2(\mu_1)-1\right)\right)\le\ell\left(A\left(t_1\left(\frac{\widetilde{r}(\mu_1)}{4}\right)\right)\right)\le\left(1+\frac{\mu_1}{4}\right)\ell(\mathbf{s})$$

Thus, we obtain the following:

$$\left(1+\frac{\mu_1}{4}\right)\ell(\mathbf{s})-\ell\left(A\left(t_1\left(\frac{\widetilde{r}(\mu_1)}{4}\right)+t_2(\mu_1)\right)\right)\ge\mu_1\cdot\ell(\mathbf{s})$$

Reorganizing thus yields

$$\ell(\mathbf{s})-\ell\left(A\left(t_1\left(\frac{\widetilde{r}(\mu_1)}{4}\right)+t_2(\mu_1)\right)\right)\ge\frac{3\mu_1\cdot\ell(\mathbf{s})}{4}$$

as required. $\qquad\square$

We continue to prove the following proposition, which states that after time $t_1\left(\frac{\widetilde{r}(\mu_1)}{4}\right)+t_2(\mu_1)$, the points reached by the gradient flow trajectory are ones where $\ell$ satisfies the PL condition with a certain PL coefficient:

**Proposition 13.** *Consider $\mu_1$ from Corollary 7. Suppose we initialize at $A(0)\in\mathcal{I}_3\left(\frac{\widetilde{r}(\mu_1)}{4}\right)\cap\mathcal{I}_4(\mu_1)$ (for $\widetilde{r}$ of Corollary 4) and at $A^{ref}(0)$, and evolve $A(t)$ and $A^{ref}(t)$ according to Equation (28). There exists $\nu>0$ such that for any time $t$ it holds that:*

- *If $t\ge t_1\left(\frac{\widetilde{r}(\mu_1)}{4}\right)+t_2(\mu_1)$ then $A(t)\in Diff(\nu)^{\mathcal{C}}$.*

- *If $t\ge t_1\left(\frac{\widetilde{r}(\mu_1)}{4}\right)+t_2^{ref}(\mu_1)$ then $A^{ref}(t)\in Diff(\nu)^{\mathcal{C}}$.*

*Additionally, it holds that $\ell$ satistfies the PL condition at $Diff(\nu)^{\mathcal{C}}$ with coefficient $\mu_2$ for*

$$\mu_2:=\frac{1}{2}\min\left\{2d,\frac{((L-1)\widetilde{\nu})^2}{4},\left(\frac{\widehat{\nu}}{\widehat{\nu}+2}\right)^2\right\}$$

*where*

$$\widetilde{\nu}:=\left(\frac{\nu}{2}\right)^{L-2},\ \widehat{\nu}:=\left(\frac{\nu}{6}\right)^{L-2}$$

*Proof.* We prove the argument for $A$ (the proof is identical for $A^{ref}$). Let $t \geq t_1(\frac{\widetilde{r}(\mu_1)}{4}) + t_2^{ref}(\mu_1)$. Denote for any $\psi \geq 0$ the function

$$f(\psi) := \min_{A \in \text{Diff}(\psi) \cap [-3,3]^d} \ell(A)$$

for $\text{Diff}(\psi)$ defined in Equation (17). Per Lemma 8 and since $\mathbf{s} \in [-3,3]^d$ it holds that

$$\ell(\mathbf{s}) = \min_{A \in \mathcal{W}_1} \ell(A) = \min_{A \in \mathcal{W}_1 \cap [-3,3]^d} \ell(A) = \min_{A \in \text{Diff}(0) \cap [-3,3]^d} \ell(A) = f(0)$$

Invoking Lemma 34, it holds that $f$ is right side continuous in $0$. Hence by continuity, we obtain that there exists $\nu$ such that for any $\psi \in [0, \nu]$ it holds that

$$f(\psi) \geq f(0) - \frac{1}{2} \min \left\{ \frac{\ell(\mathbf{s})}{2}, \frac{3\mu_1 \cdot \ell(\mathbf{s})}{4} \right\} = \ell(\mathbf{s}) - \frac{1}{2} \min \left\{ \frac{\ell(\mathbf{s})}{2}, \frac{3\mu_1 \cdot \ell(\mathbf{s})}{4} \right\} >$$

$$> \ell(\mathbf{s}) - \min \left\{ \frac{\ell(\mathbf{s})}{2}, \frac{3\mu_1 \cdot \ell(\mathbf{s})}{4} \right\}$$

On the other hand, per Proposition 12 and since the objective is non-increasing under gradient flow (see Lemma 23), we obtain the following:

$$\ell(\mathbf{s}) - \ell(A(t)) \geq \ell(\mathbf{s}) - \ell\left( A\big(t_1(\frac{\widetilde{r}(\mu_1)}{4}) + t_2(\mu_1)\big) \right) \geq \min\{ \frac{\ell(\mathbf{s})}{2}, \frac{3\mu_1 \cdot \ell(\mathbf{s})}{4} \}$$

Rearranging we thus obtain

$$f(\psi) > \ell(\mathbf{s}) - \min \left\{ \frac{\ell(\mathbf{s})}{2}, \frac{3\mu_1 \cdot \ell(\mathbf{s})}{4} \right\} \geq \ell\big(A(t)\big)$$

Note that per Lemma 34, $f$ is non increasing w.r.t $\psi$, thus it must hold that $A(t) \notin \text{Diff}(\psi)$ (since it is in $[-3,3]^d$). As this holds for any $\psi \in [0,\nu]$, we obtain that $A(t) \in \text{Diff}(\nu)^{\mathcal{C}}$. Hence, the argument follows from Lemma 19. $\qquad\square$

The final proposition of this section proves that the gradient flow trajectory converges to a global minimum $\widehat{A_2}$:

**Proposition 14.** *Consider $\mu_1$ from Corollary 7. Suppose we initialize at $A(0) \in \mathcal{I}_3(\frac{\widetilde{r}(\mu_1)}{4}) \cap \mathcal{I}_4(\mu_1)$ (for $\widetilde{r}$ of Corollary 4) and evolve $A(t)$ according to Equation (28). There exists $\widehat{A_2} \in \mathbb{R}^d$ such that*

$$\lim_{t \to \infty} A(t) = \widehat{A_2}$$

*and $\ell(\widehat{A_2}) = 0$*

*Proof.* Per Corollary 7, there exists $\nu > 0$ such that for any $t \geq t_1(\frac{\widetilde{r}(\mu_1)}{4}) + t_2(\mu_1)$ it holds that $A(t) \in \text{Diff}(\nu)^{\mathcal{C}}$ where $\ell$ satisfies the PL condition with coefficient $\mu_2$ (defined in Corollary 7). Next, note that per Lemma 12 and Corollary 3, $A(t)$ is always contained in $[-3,3]^d$, where $\ell$'s gradient is $N$ Lipschitz. Therefore, the claim follows from Lemma 26. $\qquad\square$

### D.3.7 Overall divergence from reference trajecory

In this section we show that one can choose a set of initializations such that the divergence between $\widehat{A_2}$ and some point on the reference trajectory is arbitrarily small. This shows that $\widehat{A_2}$ must not recover the teacher (per Lemma 4). We begin by proving the following lemma which gives explicit times at which the gradient flow trajectories reach points of arbitrary small value:

**Lemma 20.** *Let $\eta_2 \in (0, \min\{1, \frac{2P}{m}\})$. Consider $\mu_1$ from Corollary 7. Suppose we initialize at $A(0) \in \mathcal{I}_3(\frac{\widetilde{r}(\mu_1)}{4}) \cap \mathcal{I}_4(\mu_1)$ (for $\widetilde{r}$ of Corollary 4) and evolve $A(t)$ according to Equation (28). Denote $t_2^*(\mu_1) := \min\{t_2(\mu_1), t_2^{ref}(\mu_1)\}$. Denote the time $t_3(\eta_2) \geq 0$ to be*

$$t_3(\eta_2) := -\frac{\ln(\frac{m}{2P}\eta_2)}{2\mu_2} - \frac{1}{\lambda_-} \ln(\frac{G'^2 r_3}{2\sqrt{d}\alpha D_-(\mu_1)}) + \frac{1}{\lambda_-} \ln(\frac{r_3}{4G'^2 \sqrt{d}\alpha D_+(\mu_1)})$$

*It holds that*

$$\ell\left( A\big(t_1(\frac{\widetilde{r}(\mu_1)}{4}) + t_2^*(\mu_1) + t_3(\eta_2)\big) \right) \leq \eta_2$$

*Proof.* First, note that since $\lambda_- < 0 < G'$ and $0 < D_-(\mu_1) < D_+(\mu_1)$ we obtain that

$$t_3(\eta_2) = -\frac{\ln(\frac{m}{2P}\eta_2)}{2\mu_2} + \frac{1}{\lambda_-}\ln(\frac{D_-(\mu_1)}{2G'^4 D_+(\mu_1)}) \geq -\frac{\ln(\frac{m}{2P}\eta_2)}{2\mu_2}$$

The right term is positive as $0 < \frac{m}{2P}\eta_2 < 1$ and $\mu_2 > 0$. Next per Proposition 10 it holds that

$$|t_2(\mu_1) - t_2^{ref}(\mu_1)| \leq -\frac{1}{\lambda_-}\ln(\frac{G'^2 r_3}{2\sqrt{d}\alpha D_-(\mu)}) + \frac{1}{\lambda_-}\ln(\frac{r_3}{4G'^2\sqrt{d}\alpha D_+(\mu)}) =$$

$$= \frac{1}{\lambda_-}\ln(\frac{D_-(\mu_1)}{2G'^4 D_+(\mu_1)})$$

which results in

$$t_1(\frac{\widetilde{r}(\mu_1)}{4}) + t_2^*(\mu_1) + t_3(\eta_2) \geq t_1(\frac{\widetilde{r}(\mu_1)}{4}) + t_2(\mu_1) - \frac{\ln(\frac{m}{2P}\eta_2)}{2\mu_2}$$

Hence, per Lemma 23 we obtain that

$$\ell\left(A\big(t_1(\frac{\widetilde{r}(\mu_1)}{4}) + t_2^*(\mu_1) + t_3(\eta_2)\big)\right) \leq \ell\left(A\big(t_1(\frac{\widetilde{r}(\mu_1)}{4}) + t_2(\mu_1) - \frac{\ln(\frac{m}{2P}\eta_2)}{2\mu_2}\big)\right)$$

Per Proposition 13 there exists $\nu > 0$ such that for any $t \geq t_1(\frac{\widetilde{r}(\mu_1)}{4}) + t_2(\mu_1)$ it holds that $A(t) \in$ Diff$(\nu)^{\mathcal{C}}$ where $\ell$ satisfies the PL condition with coefficient $\mu_2$ (defined in Corollary 7). Thus, per Lemma 25 it holds that

$$\ell\left(A\big(t_1(\frac{\widetilde{r}(\mu_1)}{4}) + t_2(\mu_1) - \frac{\ln(\frac{m}{2P}\eta_2)}{2\mu_2}\big)\right) \leq \ell\left(A\big(t_1(\frac{\widetilde{r}(\mu_1)}{4}) + t_2(\mu_1)\big)\right) \cdot \exp(2\mu_2 \cdot \frac{\ln(\frac{m}{2P}\eta_2)}{2\mu_2}) \leq$$

$$\leq \ell(A(0)) \cdot \frac{m}{2P}\eta_2$$

where the second to last inequality stems from Lemma 23. The proof follows by noting that at initialization $\ell$'s value is no more than $\frac{2P}{m}$. $\qquad\square$

We continue to the following lemma which proves that there times at which the distance between the gradient flow trajectory and $\widehat{A_2}$ (defined in Proposition 14) is arbitrarily small:

**Lemma 21** (Distance between gradient flow trajectory and $\widehat{A_2}$)**.** *Let $\delta \in (0,1)$. Consider $\mu_1$ from Corollary 7. Suppose we initialize at $A(0) \in \mathcal{I}_3(\frac{\widetilde{r}(\mu_1)}{4}) \cap \mathcal{I}_4(\mu_1)$ (for $\widetilde{r}$ of Corollary 4). Denote $t_2^*(\mu_1) := \min\{t_2(\mu_1), t_2^{ref}(\mu_1)\}$. Then there exists $\eta_{2,\delta} > 0$ such that*

$$\|\widehat{A_2} - A\big(t_1(\frac{\widetilde{r}(\mu_1)}{4}) + t_2^*(\mu_1) + t_3(\eta_{2,\delta})\big)\|_2 \leq \delta$$

*Proof.* Denote $\mathcal{A}^* := \left\{A \in \mathbb{R}^d : \ell(A) = 0\right\}$. Per Corollary 7, there exists $\mu_2 > 0$ such that for any $t \geq t_1(\frac{\widetilde{r}(\mu_1)}{4}) + t_2(\mu_1)$ it holds that $\ell$ satisfies the PL condition in $A(t)$ with PL coefficient $\mu_2$. Next, note that per Lemma 12 and Corollary 3 it holds that $\ell$'s gradient is $N$ Lipschitz in $A(t)$. Therefore by Lemma 26 for any $t \geq 0$ it holds that

$$\|\widehat{A_2} - A\big(t_1(\frac{\widetilde{r}(\mu_1)}{4}) + t_2(\mu_1) + t\big)\|_2 \leq \sqrt{\frac{N}{\mu_2}}\text{Dist}\left(A\big(t_1(\frac{\widetilde{r}(\mu_1)}{4}) + t_2(\mu_1) + t\big), \mathcal{A}^*\right)$$

Additionally, note that since $\ell$ is continuous and non-negative, when considering its restriction to $[-3,3]^d$ we obtain that its 1 sub-level set is compact. Therefore we obtain by Lemma 36 that there exists $\eta_{2,\delta} \in (0, \min\{1, \frac{2P}{m}\})$ such that for any $A \in [-3,3]^d$ if $\ell(A) \leq \eta_{2,\delta}$ then $\text{Dist}(A, \mathcal{A}^*) \leq \sqrt{\frac{\mu_2}{N}}\delta$. It was shown in Lemma 20 that

$$t_1(\frac{\widetilde{r}(\mu_1)}{4}) + t_2^*(\mu_1) + t_3(\eta_{2,\delta}) \geq t_1(\frac{\widetilde{r}(\mu_1)}{4}) + t_2(\mu_1)$$

and so we obtain that

$$\|\widehat{A_2} - A\big(t_1(\frac{\widetilde{r}(\mu_1)}{4}) + t_2^*(\mu_1) + t_3(\eta_{2,\delta})\big)\|_2 \leq \sqrt{\frac{N}{\mu_2}}\mathrm{Dist}\bigg( A\big(t_1(\frac{\widetilde{r}(\mu_1)}{4}) + t_2^*(\mu_1) + t_3(\eta_{2,\delta})\big), \mathcal{A}^*\bigg)$$

It was also shown that

$$\ell\bigg( A\big(t_1(\frac{\widetilde{r}(\mu_1)}{4}) + t_2^*(\mu_1) + t_3(\eta_{2,\delta})\big)\bigg) \leq \eta_{2,\delta}$$

and so we obtain

$$\|\widehat{A_2} - A\big(t_1(\frac{\widetilde{r}(\mu_1)}{4}) + t_2^*(\mu_1) + t_3(\eta_{2,\delta})\big)\|_2 \leq \sqrt{\frac{N}{\mu_2}\frac{\mu_2}{N}}\delta = \delta$$

as required. $\qquad\qquad\square$

Before proving the last proposition of this section, we introduce another condition on the initialization which we denote $\mathcal{I}_5$:

**Definition 12.** Let $\delta, \eta > 0$. We use $\mathcal{I}_5(\delta, \eta)$ to denote the following subset of $\mathcal{I}_0$:

$$\mathcal{I}_5(\delta, \eta) := \left\{ A \in \mathcal{I}_0 : (1 - \zeta_2) \leq \frac{\delta}{\left(\frac{G'^2 r_3}{2\sqrt{d}D_-(\mu_1)}\right)^2 \exp\bigg( N \cdot \big(t_1(\frac{\widetilde{r}(\mu_1)}{4}) + t_3(\eta)\big)\bigg)} \cdot \alpha \right\}$$

For $r_3$, $D_-$, $G'$ and $\mu_1$ from Proposition 8, Corollaries 4 and 7, , and Definition 11 respectively.

We proceed by proving the following proposition which upper bounds the divergence between $A(t)$ and $A^{ref}(t)$:

**Proposition 15.** *Let $\delta > 0$. Consider $\mu_1$ from Corollary 7. Suppose we initialize at $A(0) \in \mathcal{I}_3(\frac{\widetilde{r}(\mu_1)}{4}) \cap \mathcal{I}_4(\mu_1) \cap \mathcal{I}_5(\frac{\delta}{2}, \eta_{2,\delta})$ (for $\eta_{2,\delta}$ of Lemma 21) and at $A^{ref}(0)$, and evolve $A(t)$ and $A^{ref}(t)$ according to Equation* (28)*. Denote $t_2^*(\mu_1) := \min\{t_2(\mu_1), t_2^{ref}(\mu_1)\}$. It holds that*

$$\|A\big(t_1(\frac{\widetilde{r}(\mu_1)}{4}) + t_2^*(\mu_1) + t_3(\eta_{2,\delta})\big) - A^{ref}\big(t_1(\frac{\widetilde{r}(\mu_1)}{4}) + t_2^*(\mu_1) + t_3(\eta_{2,\delta})\big)\|_2 \leq \frac{\delta}{2}$$

*for $\eta_{2,\delta}$ described in Lemma* 21.

*Proof.* Per Lemma 12 and Corollary 3 for any $t \geq 0$ it holds that $A(t), A^{ref}(t)$ are contained in $[-3, 3]^d$, where $\ell$'s gradient is $N$ Lipschitz . Thus per Lemma 30 it holds that

$$\|A\big(t_1(\frac{\widetilde{r}(\mu_1)}{4}) + t_2^*(\mu_1) + t_3(\eta_{2,\delta})\big) - A^{ref}\big(t_1(\frac{\widetilde{r}(\mu_1)}{4}) + t_2^*(\mu_1) + t_3(\eta_{2,\delta})\big)\|_2 \leq$$

$$\leq \|A\big(t_1(\frac{\widetilde{r}(\mu_1)}{4}) + t_2^*(\mu_1)\big) - A^{ref}\big(t_1(\frac{\widetilde{r}(\mu_1)}{4}) + t_2^*(\mu_1)\big)\|_2 \cdot \exp(N \cdot t_3(\eta_{2,\delta}))$$

Next, since $t_2^*(\mu_1) \leq t_2(\mu_1), t_2^{ref}(\mu_1)$ we obtain by Proposition 10 that for any $t \in [t_1(\frac{\widetilde{r}(\mu_1)}{4}), t_1(\frac{\widetilde{r}(\mu_1)}{4}) + t_2^*(\mu_1)]$ it holds that

$$H\big(A(t)\big), H\big(A^{ref}(t)\big) \in \overline{B_{r_3}}(\mathbf{s})$$

Invoking Proposition 8, the above results in

$$A(t), A^{ref}(t) \in \overline{B_{r_1}}(\mathbf{s})$$

By definitions of $r_1$ and $\overline{B_{r_1}}(\mathbf{s})$ (Proposition 8), the above yields the following for any $t \in [t_1(\frac{\widetilde{r}(\mu_1)}{4}), t_1(\frac{\widetilde{r}(\mu_1)}{4}) + t_2^*(\mu_1)]$ and $k \in [0, 1]$:

$$|\lambda_{\min}\bigg( \nabla^2\ell\big(k \cdot A(t) + (1 - k) \cdot A^{ref}(t)\big)\bigg)| \leq 2|\lambda_-|$$

Next, it holds that $\lambda_{\min}(\nabla^2 \ell(A)) = -\lambda_{\max}(-\nabla^2 \ell(A))$ for any $A \in \mathbb{R}^d$. Therefore, invoking Lemma 35 and plugging the above we obtain that

$$\|A\big(t_1(\tfrac{\widetilde{r}(\mu_1)}{4}) + t_2^*(\mu_1)\big) - A^{ref}\big(t_1(\tfrac{\widetilde{r}(\mu_1)}{4}) + t_2^*(\mu_1)\big)\|_2 \leq$$

$$\leq \exp\bigg(\int_0^{t_2^*(\mu_1)} 2|\lambda_-| d\tau\bigg) \cdot \|A\big(t_1(\tfrac{\widetilde{r}(\mu_1)}{4})\big) - A^{ref}\big(t_1(\tfrac{\widetilde{r}(\mu_1)}{4})\big)\|_2 =$$

$$= \exp(2t_2^*(\mu_1)|\lambda_-|) \cdot \|A\big(t_1(\tfrac{\widetilde{r}(\mu_1)}{4})\big) - A^{ref}\big(t_1(\tfrac{\widetilde{r}(\mu_1)}{4})\big)\|_2$$

Note that $\lambda_- < 0$ and so $|\lambda_-| = -\lambda_-$. Thus, recalling Proposition 10 we upper bound $t_2^*(\mu_1)$ and obtain that

$$\|A\big(t_1(\tfrac{\widetilde{r}(\mu_1)}{4}) + t_2^*(\mu_1)\big) - A^{ref}\big(t_1(\tfrac{\widetilde{r}(\mu_1)}{4}) + t_2^*(\mu_1)\big)\|_2 \leq$$

$$\leq \exp\bigg(-\frac{2}{\lambda_-} \ln\big(\frac{G'^2 r_3}{2\sqrt{d}\alpha D_-(\mu_1)}\big)|\lambda_-|\bigg) \cdot \|A\big(t_1(\tfrac{\widetilde{r}(\mu_1)}{4})\big) - A^{ref}\big(t_1(\tfrac{\widetilde{r}(\mu_1)}{4})\big)\|_2 =$$

$$= \bigg(\frac{G'^2 r_3}{2\sqrt{d}\alpha D_-(\mu_1)}\bigg)^2 \cdot \|A\big(t_1(\tfrac{\widetilde{r}(\mu_1)}{4})\big) - A^{ref}\big(t_1(\tfrac{\widetilde{r}(\mu_1)}{4})\big)\|_2$$

Applying Lemma 30 once more, we obtain that

$$\|A\big(t_1(\tfrac{\widetilde{r}(\mu_1)}{4})\big) - A^{ref}\big(t_1(\tfrac{\widetilde{r}(\mu_1)}{4})\big)\|_2 \leq \|A(0) - A^{ref}(0)\|_2 \cdot \exp(N \cdot t_1(\tfrac{\widetilde{r}(\mu_1)}{4}))$$

Finally, by Equation (30) and Definition 4 we have at initalization that

$$\|A(0) - A^{ref}(0)\|_2 = |a_2(0) - a_2^{ref}(0)| = \alpha \cdot (1 - \zeta_2)$$

Altogether we obtain the following bound on the divergence:

$$\|A\big(t_1(\tfrac{\widetilde{r}(\mu_1)}{4}) + t_2^*(\mu_1) + t_3(\eta_{2,\delta})\big) - A^{ref}\big(t_1(\tfrac{\widetilde{r}(\mu_1)}{4}) + t_2^*(\mu_1) + t_3(\eta_{2,\delta})\big)\|_2 \leq$$

$$\leq \alpha \cdot (1 - \zeta_2)\bigg(\frac{G'^2 r_3}{2\sqrt{d}\alpha D_-(\mu_1)}\bigg)^2 \exp\bigg(N \cdot \big(t_1(\tfrac{\widetilde{r}(\mu_1)}{4}) + t_3(\eta_{2,\delta})\big)\bigg)$$

The proof concludes by recalling that the initialization satisfies $A(0) \in \mathcal{I}_5(\tfrac{\delta}{2}, \eta_{2,\delta})$ and so since $\alpha > 0$ we can rewrite and obtain that

$$\alpha \cdot (1 - \zeta_2) \leq \frac{\tfrac{\delta}{2}}{\bigg(\frac{G'^2 r_3}{2\sqrt{d}\alpha D_-(\mu_1)}\bigg)^2 \exp\bigg(N \cdot \big(t_1(\tfrac{\widetilde{r}(\mu_1)}{4}) + t_3(\eta_{2,\delta})\big)\bigg)}$$

and so

$$\alpha \cdot (1 - \zeta_2)\bigg(\frac{G'^2 r_3}{2\sqrt{d}\alpha D_-(\mu_1)}\bigg)^2 \exp\bigg(N \cdot \big(t_1(\tfrac{\widetilde{r}(\mu_1)}{4}) + t_3(\eta_{2,\delta})\big)\bigg) \leq \frac{\delta}{2}$$

$$\square$$

We finish this section by proving the following corollary bounding the distance between $\widehat{A_2}$ (the point to which the gradient flow trajecory converges to) and $A^{ref}\big(t_1(\tfrac{\widetilde{r}(\mu_1)}{4}) + t_2^*(\mu_1) + t_3(\eta_{2,\delta})\big)$, a point which by Lemma 4 is far away from the teacher:

**Corollary 8.** *Let $\delta > 0$. Consider $\mu_1$ from Corollary 7. Suppose we initialize at $A(0) \in \mathcal{I}_3(\tfrac{\widetilde{r}(\mu_1)}{4}) \cap \mathcal{I}_4(\mu_1) \cap \mathcal{I}_5(\tfrac{\delta}{2}, \eta_{2,\delta})$ (for $\eta_{2,\delta}$ of Lemma 21) and at $A^{ref}(0)$, and evolve $A(t)$ and $A^{ref}(t)$ according to Equation (28). Denote $t_2^*(\mu_1) := \min\{t_2(\mu_1), t_2^{ref}(\mu_1)\}$. It holds that*

$$\|\widehat{A_2} - A^{ref}\big(t_1(\tfrac{\widetilde{r}(\mu_1)}{4}) + t_2^*(\mu_1) + t_3(\eta_{2,\delta})\big)\|_2 \leq \delta$$

*where $A(t)$ converges to $\widehat{A_2}$ (see Proposition 14).*

*Proof.* Per Lemma 21, it holds that

$$\|\widehat{A_2} - A\big(t_1(\frac{\widetilde{r}(\mu_1)}{4}) + t_2^*(\mu_1) + t_3(\eta_{2,\delta})\big)\|_2 \le \frac{\delta}{2}$$

Per Proposition 15, it holds that

$$\|A\big(t_1(\frac{\widetilde{r}(\mu_1)}{4}) + t_2^*(\mu_1) + t_3(\eta_{2,\delta})\big) - A^{ref}\big(t_1(\frac{\widetilde{r}(\mu_1)}{4}) + t_2^*(\mu_1) + t_3(\eta_{2,\delta})\big)\|_2 \le \frac{\delta}{2}$$

The claim thus follows from the triangle inequality. $\qquad\square$

Let $\mathcal{I}_2(\delta_2)$ be the initialization subset defined above, *i.e.*

$$I_2(\delta_2) := \mathcal{I}_3(\frac{\widetilde{r}(\mu_1)}{4}) \cap \mathcal{I}_4(\mu_1) \cap \mathcal{I}_5(\frac{\delta_2}{2}, \eta_{2,\delta_2})$$

for $\delta_2$ of Lemma 4. Invoking the lemma we obtain that for any $L' \ge L+2$ it holds that

$$Gen_{L'}(\widehat{A_2}) \ge \frac{1}{2}\min\{0.1, 1/(9d) \cdot (1-(0.6)^{1/(L-1)})$$

which concludes our proof of the fact that gradient flow under $\mathcal{S}_2$ converges to a non-generalizing solution when initialized at $\mathcal{I}_2(\delta)$.

### D.4 INITIALIZATION SUBSETS INTERSECT

In Proposition 4 we showed that when initializing at $\mathcal{I}_1(\epsilon)$ gradient flow converges to a point $\widehat{A_1}$ which satisfies $Gen_{L'}(\widehat{A_1}) \le \epsilon$. In Corollary 8 we showed that when initializing at $\mathcal{I}_2(\delta_2)$ gradient flow converges to a point $\widehat{A_2}$ which satisfies $Gen_{L'}(\widehat{A_2}) \ge \frac{1}{2}\min\{0.1, 1/(9d) \cdot (1-(1/2)^{1/(L-1)})$.

In this section we show that not only do the initialization subsets $\mathcal{I}_1(\epsilon)$ and $\mathcal{I}_2(\delta_2)$ intersect but also that their intersection contains an open subset. For convenience of the reader, we rewrite the full requirements as they appear in the statements of Sections D.1 to D.3 and state their respective arguments. The base initialization set (Equation (30)) we consider is

$$\mathcal{I}_0 = \left\{ \alpha \cdot (\zeta_1, \dots, \zeta_d)^\top \in \mathbb{R}^d : \alpha \in (0, \frac{1}{2d}), 1 = \zeta_1 > \zeta_2 > \cdots > \zeta_d > 0 \right\}$$

$\mathcal{I}_1(\epsilon)$ (Definition 3) was defined as

$$\mathcal{I}_1(\epsilon) = \left\{ A \in \mathcal{I}_0 : \forall j \in \{2, \dots, d\}. \; \alpha \le \left(\frac{1 - (1-\eta_{1,\delta_1})^{L-1} - \eta_{1,\delta_1}\sqrt{\frac{n}{P}}}{d-1}\right)^{\frac{1}{L-1}} \frac{1}{\zeta_j} (1-\zeta_j^{L-3})^{\frac{1}{L-3}} \right\}$$

for $\eta_{1,\delta_1}$ of Remark 1. $\mathcal{I}_3(\frac{\widetilde{r}(\mu_1)}{4})$ (Definition 10) was defined as

$$\mathcal{I}_3(\frac{\widetilde{r}(\mu_1)}{4}) := \left\{ A \in \mathcal{I}_0 : \alpha \le \frac{\min\{\frac{\widetilde{r}(\mu_1)}{4}, \|A^Z\big(t_1(\frac{\widetilde{r}(\mu_1)}{4})\big) - A^-\big(t_1(\frac{\widetilde{r}(\mu_1)}{4})\big)\|_2\}}{6d} e^{-N \cdot t_1(\frac{\widetilde{r}(\mu_1)}{4})}, \; \zeta_d \le \frac{1}{2} \right\}$$

for $\widetilde{r}(\cdot)$ and $\mu_1$ of Corollaries 4 and 7 respectively. $\mathcal{I}_4(\mu_1)$ (Definition 11) was defined as

$$\mathcal{I}_4(\mu_1) = \left\{ A \in \mathcal{I}_0 : \alpha \le \frac{r_3}{4\max\{2, \exp(-2\lambda_-)\} \cdot G'^2 \sqrt{d}D_+(\mu_1)} \right\}$$

$\mathcal{I}_5(\delta_2, \eta_{2,\delta_2})$ (Definition 12) was defined as

$$\mathcal{I}_5(\delta_2, \eta_{2,\delta_2}) = \left\{ A \in \mathcal{I}_0 : (1-\zeta_2) \le \frac{\delta_2}{\left(\frac{G'^2 r_3}{2\sqrt{d}D_-(\mu_1)}\right)^2 \exp\left(N \cdot \big(t_1(\frac{\widetilde{r}(\mu_1)}{4}) + t_3(\eta_{2,\delta_2})\big)\right)} \cdot \alpha \right\}$$

for $\eta_{2,\delta_2}$ of Lemma 21. We begin by observing the following simplication:

$$\mathcal{I}_1(\epsilon) \cap \mathcal{I}_0 = \mathcal{I}_0 \cap \left\{ A \in \mathcal{I}_0 : \alpha \le \left(\frac{1 - (1-\eta_{1,\delta_1})^{L-1} - \eta_{1,\delta_1}\sqrt{\frac{n}{P}}}{d-1}\right)^{\frac{1}{L-1}} \frac{1}{\zeta_2} (1-\zeta_2^{L-3})^{\frac{1}{L-3}} \right\}$$

since the right hand side is monotonically decreasing in $\zeta$ and since $\zeta_2 > \zeta_j$ for any $j \in \{3, \ldots d\}$. Next, we also require that $\frac{1}{2} \geq \zeta_3$. Note that this requirement satisfies the requirement of $\mathcal{I}_3$ on the magnitude of $\zeta_d$ (since $\zeta_3 > \zeta_4 > \cdots > \zeta_d$). Moreover, note that $\mathcal{I}_3$ and $\mathcal{I}_4$ impose upper bounds on $\alpha$ which are not related to $\zeta_2, \ldots, \zeta_d$. Therefore, there exists some $\alpha^* > 0$ such that if $\alpha \in (0, \alpha^*)$ then all of these conditions are satisfied. Moving on to the conditions that involve $\alpha$ and $\zeta_2$ we first observe that there exists constants $S, T > 0$ such that

$$\alpha \leq \frac{S}{\zeta_2}(1 - \zeta_2^{L-3})^{\frac{1}{L-3}}$$

is equivalent to the condition from $\mathcal{I}_1$ and

$$(1 - \zeta_2) \leq T\alpha$$

is equivalent to the condition from $\mathcal{I}_5$. Invoking Lemma 37 we obtain that there exist constants $q_1, w_1 \in (0, 1)$ and $q_2, w_2 \in (\frac{1}{2}, 1)$ such that taking $\alpha \in (q_1, w_1)$ and $\zeta_2 \in (q_2, w_2)$ satisfies the two conditions involving $\alpha$ and $\zeta_2$. The above discussion is summarized in the following proposition:

**Proposition 16.** *For any $\epsilon > 0$ there exist constants $q_1, w_1 \in (0, 1)$ and $q_2, w_2 \in (\frac{1}{2}, 1)$ such that the set*

$$\left\{ \alpha \cdot (1, \zeta_2, ..., \zeta_d)^\top \; : \; (\alpha, \zeta_2) \in (q_1, w_1) \times (q_2, w_2), \; \frac{1}{2} \geq \zeta_3... > \zeta_d \right\}$$

*is contained in the intersection of initialization subsets given by*

$$\mathcal{I}_1(\epsilon) \cap \mathcal{I}_2(\delta_2)$$

Now that we have characterized a set of initializations which satisfy all our requirements, we will show that this set contains an open subset.

**Lemma 22.** *The set of initializations satisfying all our requirements, namely*

$$\left\{ \alpha \cdot (1, \zeta_2, ..., \zeta_d)^\top \; : \; (\alpha, \zeta_2) \in (q_1, w_1) \times (q_2, w_2), \; \frac{1}{2} \geq \zeta_3... > \zeta_d \right\}$$

*contains an open set, namely a set of the form $(a_1, b_1) \times (a_2, b_2) \times ... \times (a_d, b_d)$.*

*Proof.* We begin by restricting $\zeta_3, \ldots, \zeta_d$ by requiring that for $3 \leq j \leq d$, $\zeta_j \in [e_j, f_j]$ where the closed intervals $\{[e_j, f_j]\}_{2 \leq j \leq d}$ satisfy

$$\frac{1}{2} > f_2 > e_2 > f_3 > e_3 > \cdots > f_d > e_d > 0$$

We can restrict $\alpha$ further, by requiring that $\alpha \in (a_1, b_1)$, where $a_1, b_1$ are chosen such that for all $2 \leq j \leq d$ we have

$$a_1 f_j > b_1 e_j$$

We now claim that for all $2 \leq j \leq d$

$$(b_1 e_j, a_1 f_j) \subseteq \bigcap_{\alpha \in (a_1, b_1)} [\alpha e_j, \alpha f_j]$$

Indeed, for any $\alpha \in (a_1, b_1)$ we have $b_1 e_j > \alpha e_j$ and $a_1 f_j < \alpha f_j$. It follows that we can take $a_j = b_1 e_j, b_j = a_1 f_j$ and obtain that the set $(a_1, b_1) \times (a_2, b_2) \times ... \times (a_d, b_d)$ is contained within

$$\{\alpha(1, \zeta_2, \ldots, \zeta_d) \; : \; (\alpha, \zeta_2) \in (q_1, w_1) \times (q_2, w_2), \; \frac{1}{2} > \zeta_3 > \cdots > \zeta_d\}$$

as required. □

**Lemma 23.** *Let $f : \mathbb{R}^d \to \mathbb{R}$ be some differentiable function. Suppose we optimize over $f$ by initializing $\mathbf{x}(0) := \mathbf{x_0}$ for some $\mathbf{x_0} \in \mathbb{R}^d$ and updating using gradient flow, i.e.:*

$$\dot{\mathbf{x}}(t) := \frac{d}{dt}\mathbf{x}(t) = -\nabla f(\mathbf{x}(t))$$

*Then the objective is non-increasing w.r.t time, i.e. for any $t \geq 0$ it holds that*

$$\frac{d}{dt}f(\mathbf{x}(t)) \leq 0$$

*Proof.* Applying the chain rule, we obtain the following:

$$\frac{d}{dt}f(\mathbf{x}(t)) = \nabla f(\mathbf{x}(t))^\top \frac{d}{dt}\mathbf{x}(t) = f(\mathbf{x}(t))^\top(-f(\mathbf{x}(t))) = -\|f(\mathbf{x}(t))\|_2^2 \leq 0$$

$\square$

**Lemma 24.** *Let $f : \mathbb{R}^d \to \mathbb{R}$ be some continuously differentiable function, which is also coercive, namely*

$$\lim_{\|x\| \to \infty} f(x) = \infty$$

*Suppose we optimize over $f$ by initializing $\mathbf{x}(0) := \mathbf{x_0}$ for some $\mathbf{x_0} \in \mathbb{R}^d$ and updating using gradient flow, i.e.:*

$$\dot{\mathbf{x}}(t) := \frac{d}{dt}\mathbf{x}(t) = -\nabla f(\mathbf{x}(t)) \tag{31}$$

*Then there exists a global solution to the above ODE, namely a curve $\mathbf{x}(t)$ which satisfies the above equation for all $t \geq 0$.*

*Proof.* By Lemma 23 and the coercivity of $f$, the trajectories of gradient flow cannot escape from some compact set $K := K(\mathbf{x_0})$. Because $f$ is continuously differentiable $\nabla f$ has some finite Lipschitz constant on $K$. Existence of the solution for all $t \geq 0$ now follows from the Picard–Lindelöf theorem (see Teschl (2024)). $\square$

**Theorem 4.** *Let $\mathcal{V} \subseteq \mathbb{R}^d$ be an open set. Let $f : \mathcal{V} \to \mathbb{R}$ be a non-negative differentiable function satisfying the following conditions:*

- *The set $X^* := \{\mathbf{x} \in \mathcal{V} : f(\mathbf{x}) = 0\}$ is not empty.*

- *There exists $\mu > 0$ such that for any $\mathbf{x} \in \mathcal{V}$ it holds that*

$$\|\nabla f(\mathbf{x})\|_2^2 \geq 2\mu f(\mathbf{x})$$

- *There exists $M > 0$ such that $\nabla f(\mathbf{x})$ is $M$-Lipschitz in $\mathcal{V}$.*

*Suppose we optimize over $f$ by initializing $\mathbf{x}(0) := \mathbf{x_0}$ for some $\mathbf{x_0} \in \mathcal{V}$ and evolving via gradient flow, i.e. via the update rule*

$$\dot{\mathbf{x}}(t) := \frac{d}{dt}\mathbf{x}(t) = -\nabla f(\mathbf{x(t)})$$

*Assume the set $\mathcal{V}$ is not escaped, i.e. for any time $t \geq 0$ it holds that $\mathbf{x}(t) \in \mathcal{V}$. Then it holds that*

$$\int_0^\infty \|\dot{\mathbf{x}}(t)\|_2 dt = \int_0^\infty \|\nabla f(\mathbf{x}(t))\|_2 dt \leq \sqrt{\frac{M}{\mu}} Dist(\mathbf{x_0}, X^*)$$

*Proof.* The theorem is a restatement of theorem 9 in Gupta et al. (2021). $\square$

**Lemma 25.** *Let $\mathcal{V} \subseteq \mathbb{R}^d$ be an open set. Let $f : \mathcal{V} \to \mathbb{R}$ be a non-negative differentiable function satisfying the following conditions:*

- *The set $\mathbf{x}^* := \{\mathbf{x} \in \mathcal{V} : f(\mathbf{x}) = 0\}$ is not empty.*

- *PL condition - there exists $\mu > 0$ such that for any $\mathbf{x} \in \mathcal{V}$ it holds that*

$$\|\nabla f(\mathbf{x})\|^2 \geq 2\mu f(\mathbf{x})$$

*Suppose we optimize over $f$ by initializing $\mathbf{x}(0) := \mathbf{x_0}$ for some $\mathbf{x_0} \in \mathcal{V}$ and evolving via gradient flow, i.e. via the update rule*

$$\dot{\mathbf{x}}(t) := \frac{d}{dt}\mathbf{x}(t) = -\nabla f(\mathbf{x(t)})$$

*Assume the set $\mathcal{V}$ is not escaped, i.e. for any time $t \geq 0$ it holds that $\mathbf{x}(t) \in \mathcal{V}$. Then for any $t \geq 0$ it holds that*

$$f(\mathbf{x}(t)) \leq f(\mathbf{x}(0)) \cdot \exp(-2\mu \cdot t)$$

*Namely, it holds that*

$$\lim_{t \to \infty} f(\mathbf{x}(t)) = 0$$

*Proof.* Let $t \geq 0$. By the chain rule, it holds that

$$\frac{d}{dt}f(\mathbf{x}(t)) = \nabla f(\mathbf{x}(t))^\top \frac{d}{dt}\mathbf{x}(t) = f(\mathbf{x}(t))^\top(-f(\mathbf{x}(t))) = -\|f(\mathbf{x}(t))\|_2^2$$

By the PL condition and since $\mathcal{V}$ is not escaped, we have that

$$\frac{d}{dt}f(\mathbf{x}(t)) = -\|\nabla f(\mathbf{x}(t))\|_2^2 \leq -2\mu f(\mathbf{x}(t))$$

Therefore, by Grönwall's inequality (Gronwall (1919)) we have that

$$f(\mathbf{x}(t)) \leq f(\mathbf{x}(0)) \cdot \exp\left(\int_0^t -2\mu d\tau\right) = f(\mathbf{x}(0)) \cdot \exp(-2\mu \cdot t)$$

Taking the limit as $t \to \infty$ completes the proof. $\qquad\square$

**Definition 13.** Let $\mathcal{V} \subseteq \mathbb{R}^d$ be an open set. Let $f : \mathcal{V} \to \mathbb{R}$ be a differentiable function. We say that $f$ satisfies the *Polyak-Lojasiewicz condition* with coefficient $\mu > 0$ at $\mathbf{x} \in \mathcal{V}$

$$\|\nabla f(\mathbf{x})\|_2^2 \geq 2\mu(f(\mathbf{x}) - \min_{\mathbf{y} \in \mathbb{R}^d} f(\mathbf{y}))$$

If the above holds for all $\mathbf{x} \in \mathcal{V}$ we say that $f$ satisfies the PL condition in $\mathcal{V}$.

**Lemma 26.** *Let $\mathcal{V} \subseteq \mathbb{R}^d$ be an open set. Let $f : \mathcal{V} \to \mathbb{R}$ be a non-negative differentiable function satisfying the following conditions:*

- *The set $X^* := \{\mathbf{x} \in \mathcal{V} : f(\mathbf{x}) = 0\}$ is not empty.*

- *$f$ satisfies the PL condition with coefficient $\mu > 0$ (see Definition 13).*

- *Lipschitz gradient - there exists $M > 0$ such that $\nabla f(\mathbf{x})$ is $M$-Lipschitz in $\mathcal{V}$.*

*Suppose we optimize over $f$ by initializing $\mathbf{x}(0) := \mathbf{x_0}$ for some $\mathbf{x_0} \in \mathcal{V}$ and evolving via gradient flow, i.e. via the update rule*

$$\dot{\mathbf{x}}(t) := \frac{d}{dt}\mathbf{x}(t) = -\nabla f(\mathbf{x(t)})$$

*Assume the set $\mathcal{V}$ is not escaped, i.e. for any time $t \geq 0$ it holds that $\mathbf{x}(t) \in \mathcal{V}$. Then the limit $\lim_{t \to \infty} \mathbf{x}(t) = \mathbf{x}^*$ exists and satisfies $f(\mathbf{x}^*) = 0$ and*

$$\|\mathbf{x}^* - \mathbf{x_0}\|_2 \leq \sqrt{\frac{M}{\mu}}Dist(\mathbf{x_0}, X^*)$$

*Proof.* Let $\epsilon > 0$. By theorem 4, it holds that

$$\int_0^\infty \|\dot{\mathbf{x}}(\tau)\|_2 d\tau \leq \sqrt{\frac{M}{\mu}} \text{Dist}(\mathbf{x_0}, X^*)$$

which is finite since $X^*$ is not empty. Hence, there exists $t^* \geq 0$ such that for any $t \geq t^*$ it holds that

$$\int_t^\infty \|\dot{\mathbf{x}}(\tau)\|_2 d\tau \leq \epsilon$$

Therefore, for any $t_2 \geq t_1 \geq t^*$ it holds by the fundamental theorem of calculus and by the triangle inequality that

$$\|\mathbf{x}(t_2) - \mathbf{x}(t_1)\|_2 = \|\mathbf{x_0} + \int_0^{t_2} \dot{\mathbf{x}}(\tau)d\tau - \mathbf{x_0} - \int_0^{t_1} \dot{\mathbf{x}}(\tau)d\tau\|_2 =$$

$$= \|\int_{t_1}^{t_2} \dot{\mathbf{x}}(\tau)d\tau\|_2 \leq \int_{t_1}^{t_2} \|\dot{\mathbf{x}}(\tau)\|_2 d\tau \leq \int_{t_1}^\infty \|\dot{\mathbf{x}}(\tau)\|_2 d\tau \leq \epsilon$$

Thus, the Cauchy convergence criterion is met and so the limit $\lim_{t\to\infty} \mathbf{x}(t) = \mathbf{x}^*$ exists. Plugging $f$'s continuity and lemma 25 yields the following

$$f(\mathbf{x}^*) = f(\lim_{t\to\infty} \mathbf{x}(t)) = \lim_{t\to\infty} f(\mathbf{x}(t)) = 0$$

Finally, by continuity and by the triangle inequality it holds that

$$\|\mathbf{x}^* - \mathbf{x_0}\|_2 = \|\mathbf{x_0} + \int_0^\infty \dot{\mathbf{x}}(\tau)d\tau - \mathbf{x_0}\|_2 = \|\int_0^\infty \dot{\mathbf{x}}(\tau)d\tau\|_2 \leq$$

$$\leq \int_0^\infty \|\dot{\mathbf{x}}(t)\|_2 \leq \sqrt{\frac{M}{\mu}} \text{Dist}(\mathbf{x_0}, X^*)$$

as required. $\square$

**Lemma 27.** *Let $a, b \in \mathbb{R}$. An eigendecomposition of the matrix $(a-b)I_d + b\mathbf{1}_{d\times d}$ is the following:*

- *The eigenvector $\mathbf{1}_d$ with the eigenvalue $a + (d-1)b$.*

- *For $j \in \{2, \ldots, d\}$ the eigenvector $\mathbf{e_1} - \mathbf{e_j}$ with the eigenvalue $a - b$.*

*Proof.* First, it holds that

$$[(a-b)I_d + b\mathbf{1}_{d\times d}]\mathbf{1}_d = (a-b)\mathbf{1}_d + b \cdot d\mathbf{1}_d = (a + (d-1)b)\mathbf{1}_d$$

hence $\mathbf{1}_d$ is an eigenvector with the eigenvalue $a + (d-1)b$. Next, note that for any $j \in \{2, \ldots, d\}$ we have

$$[(a-b)I_d + b\mathbf{1}_{d\times d}](\mathbf{e_1} - \mathbf{e_j}) = (a-b)\mathbf{e_1} + b\mathbf{1}_d - (a-b)\mathbf{e_j} - b\mathbf{1}_d = (a-b)(\mathbf{e_1} - \mathbf{e_j})$$

hence $\mathbf{e_1} - \mathbf{e_j}$ is an eigenvector with the eigenvalue $a - b$. Finally, note that the set $\{\mathbf{1}_d, \mathbf{e_1} - \mathbf{e_2}, \ldots, \mathbf{e_1} - \mathbf{e_d}\}$ is linearly independent and thus spans $\mathbb{R}^d$. Therefore, the above eigenvectors and eigenvalues constitute and eigendecomposition of $(a-b)I_d + b\mathbf{1}_{d\times d}$. $\square$

**Lemma 28.** *Let $W \in \mathbb{R}^{d\times d}$ be a symmetric matrix and $\mathbf{b} \in \mathbb{R}^d$ be a vector. The solution of the linear dynamical system*

$$\dot{\mathbf{y}}(t) = -W(\mathbf{y}(t) - \mathbf{b})$$

*is given by*

$$\mathbf{y}(t) = \exp(-W)(\mathbf{y}(0) - \mathbf{b}) = Q\exp(-t \cdot \Lambda)Q^\top(\mathbf{y}(0) - \mathbf{b}) + \mathbf{b}$$

*where $W = Q\Lambda Q^\top$ is any orthogonal eigendecomposition of $W$.*

*Proof.* Using the change of variables $\mathbf{z}(t) = \mathbf{y}(t) - \mathbf{b}$, the given system simplifies to

$$\dot{\mathbf{z}}(t) = -W\mathbf{z}(t)$$

whose solution is given by

$$\mathbf{z}(t) = \exp(-t \cdot W)\mathbf{z}(0)$$

Reversing the change of variables and reorganizing yields

$$\mathbf{y}(t) = \exp(-t \cdot W)(\mathbf{y}(0) - \mathbf{b}) + \mathbf{b}$$

Let $W = Q\Lambda Q^\top$ be an orthogonal eigendecomposition of the symmetric $W$. Then we have by the definition of matrix exponential that

$$\mathbf{y}(t) = Q\exp(-t \cdot \Lambda)Q^\top(\mathbf{y}(0) - \mathbf{b}) + \mathbf{b}$$

as required. $\qquad\square$

**Lemma 29.** *Let $\mathbf{x_0} \in \mathbb{R}^d$. Let $\mathcal{V}_1, \mathcal{U}_1 \subseteq \mathbb{R}^d$ be neighborhoods of $\mathbf{x_0}$. Let $H : \mathcal{V}_1 \to \mathcal{U}_1$ be a $C^3$ diffeomorphism. There exists $r > 0$ such that for any $r_1 \in (0, r]$ there exist $r_2 \in (0, r_1)$ and $r_3 > 0$ for which*

- *$H[\overline{B_{r_2}}(\mathbf{s})] \subseteq \overline{B_{r_3}}(\mathbf{s}) \subseteq H[\overline{B_{r_1}}(\mathbf{s})]$*

- *$H|_{\overline{B_{r_1}}(\mathbf{s})}$ is Lipschitz*

- *$H^{-1}|_{\overline{B_{r_3}}(\mathbf{s})}$ is Lipschitz*

*Proof.* $\mathcal{V}_1$ and $\mathcal{U}_1$ are neighborhoods of $\mathbf{x_0}$ and so there exist $r', r'' > 0$ for which $\overline{B_{r'}}(\mathbf{x_0}) \subseteq \mathcal{V}_1$ and $\overline{B_{r''}}(\mathbf{x_0}) =: \mathcal{U}_2 \subseteq \mathcal{U}_1$. Hence, by $H$'s continuity there exists some small enough $r > 0$ for which $\overline{B_r}(\mathbf{x_0}) \subseteq \mathcal{V}_1$ is mapped by $H$ to $H[\overline{B_r}(\mathbf{x_0})] \subseteq \mathcal{U}_2$. Fix $r_1 \in (0, r]$. Then it holds that $\overline{B_{r_1}}(\mathbf{x_0})$ satisfies

$$\overline{B_{r_1}}(\mathbf{x_0}) \subseteq \overline{B_r}(\mathbf{x_0}) \subseteq \mathcal{V}_1$$

and is mapped by $H$ to

$$H[\overline{B_{r_1}}(\mathbf{x_0})] \subseteq H[\overline{B_r}(\mathbf{x_0})] \subseteq \mathcal{U}_2$$

Since $\overline{B_{r_1}}(\mathbf{x_0})$ is a compact ball and since $H$ is $C^3$, we obtain that $H$ is Lipschitz over $\overline{B_{r_1}}(\mathbf{x_0})$, *i.e.* $H|_{\overline{B_{r_1}}(\mathbf{x_0})}$ is Lipschitz. Similarly, we obtain that $H^{-1}$ is Lipschitz over $\mathcal{U}_2$, *i.e.* $H^{-1}|_{\mathcal{U}_2}$ is Lipschitz. Therefore since $H[\overline{B_{r_1}}(\mathbf{x_0})] \subseteq \mathcal{U}_2$ we obtain that $H^{-1}|_{H[\overline{B_{r_1}}(\mathbf{x_0})]}$ is Lipschitz. Next, note that for any $r_2 \in (0, r_1)$ the compact ball $\overline{B_{r_2}}(\mathbf{x_0})$ satisfies $H[\overline{B_{r_2}}(\mathbf{x_0})] \subseteq H[\overline{B_{r_1}}(\mathbf{x_0})]$. Hence, by taking a small enough $r_2$ we can guarantee by $H$'s continuity that there exists some $r_3 > 0$ for which $\overline{B_{r_3}}(\mathbf{x_0})$ satisfies

$$H[\overline{B_{r_2}}(\mathbf{x_0})] \subseteq \overline{B_{r_3}}(\mathbf{x_0}) \subseteq H[\overline{B_{r_1}}(\mathbf{x_0})]$$

Since $H^{-1}|_{H[\overline{B_{r_1}}(\mathbf{x_0})]}$ is Lipschitz and $\overline{B_{r_3}}(\mathbf{x_0}) \subseteq H[\overline{B_{r_1}}(\mathbf{x_0})]$ we obtain that $H^{-1}|_{\overline{B_{r_3}}(\mathbf{x_0})}$ is Lipschitz. $\qquad\square$

**Lemma 30.** *Let $f : \mathbb{R}^d \to \mathbb{R}^d$ be a vector field and let $B \subseteq \mathbb{R}^d$ be a bounded and compact space. Suppose $f$ is $N$-Lipschitz within $B$ for some constant $N > 0$. Consider the following system of ODEs:*

$$\dot{\mathbf{x}}(t) = f(\mathbf{x}(t))$$

*Consider two initialization points $\mathbf{x_1}(0), \mathbf{x_2}(0) \in B$. Suppose we evolve $\mathbf{x_1}(t), \mathbf{x_2}(t)$ according to the above system. If for any $t \geq 0$ it holds that $\mathbf{x_1}(t), \mathbf{x_2}(t) \in B$, then*

$$\|\mathbf{x_1}(0) - \mathbf{x_2}(0)\|_2 \cdot \exp(-N \cdot t) \leq \|\mathbf{x_1}(t) - \mathbf{x_2}(t)\|_2 \leq \|\mathbf{x_1}(0) - \mathbf{x_2}(0)\|_2 \cdot \exp(N \cdot t)$$

*Proof.* Let $t \geq 0$. Applying the chain rule, we obtain the following:

$$\frac{d}{dt}\|\mathbf{x_1}(t) - \mathbf{x_2}(t)\|_2 = \frac{d}{dt}\sqrt{\left(\mathbf{x_1}(t) - \mathbf{x_2}(t)\right)^\top \left(\mathbf{x_1}(t) - \mathbf{x_2}(t)\right)} =$$

$$= \frac{1}{2\|\mathbf{x_1}(t) - \mathbf{x_2}(t)\|_2} \frac{d}{dt}\left(\mathbf{x_1}(t) - \mathbf{x_2}(t)\right)^\top \left(\mathbf{x_1}(t) - \mathbf{x_2}(t)\right) =$$

$$= \frac{2}{2\|\mathbf{x_1}(t) - \mathbf{x_2}(t)\|_2} \cdot \left(\mathbf{x_1}(t) - \mathbf{x_2}(t)\right)^\top \frac{d}{dt}\left(\mathbf{x_1}(t) - \mathbf{x_2}(t)\right) =$$

$$= \frac{1}{\|\mathbf{x_1}(t) - \mathbf{x_2}(t)\|_2} \cdot \left(\mathbf{x_1}(t) - \mathbf{x_2}(t)\right)^\top \left(f(\mathbf{x_1}(t)) - f(\mathbf{x_2}(t))\right)$$

Thus, applying the Cauchy Schwarz inequality we obtain

$$\frac{d}{dt}\|\mathbf{x_1}(t) - \mathbf{x_2}(t)\|_2 \leq \frac{1}{\|\mathbf{x_1}(t) - \mathbf{x_2}(t)\|_2} \cdot \|\mathbf{x_1}(t) - \mathbf{x_2}(t)\|_2 \cdot \|f(\mathbf{x_1}(t)) - f(\mathbf{x_2}(t))\|_2 =$$

$$= \|f(\mathbf{x_1}(t)) - f(\mathbf{x_2}(t))\|_2$$

$f$ is $N$-Lipschitz within $B$, and so since $\mathbf{x_1}(t), \mathbf{x_2}(t) \in B$ we obtain

$$\frac{d}{dt}\|\mathbf{x_1}(t) - \mathbf{x_2}(t)\|_2 \leq \|f(\mathbf{x_1}(t)) - f(\mathbf{x_2}(t))\|_2 \leq N \cdot \|\mathbf{x_1}(t) - \mathbf{x_2}(t)\|_2$$

Finally, plugging Grönwall's inequality (Gronwall (1919)) results in

$$\|\mathbf{x_1}(t) - \mathbf{x_2}(t)\|_2 \leq \|\mathbf{x_1}(0) - \mathbf{x_2}(0)\|_2 \cdot \exp(N \cdot t)$$

Next, consider the following system of ODEs which we coin the *reversal of $f$*:

$$\dot{\overline{\mathbf{x}}}(t) = -f(\overline{\mathbf{x}}(t))$$

Consider the initialization points $\overline{\mathbf{x_1}}(0) = \mathbf{x_1}(t)$ and $\overline{\mathbf{x_2}}(0) = \mathbf{x_2}(t)$. Suppose we evolve $\overline{\mathbf{x_1}}(t), \overline{\mathbf{x_2}}(t)$ according to the reversal of $f$. Then it holds that for any time $\bar{t} \in [0, t]$ and any $i \in [2]$ we have

$$\overline{\mathbf{x_i}}(\bar{t}) = \mathbf{x_i}(t - \bar{t})$$

hence $\overline{\mathbf{x_i}}(\bar{t}) \in B$. As Lipschitz continuity is invariant to sign, $-f$ is $N$-Lipschitz within $B$. Therefore, we can apply the above claim on the reversal of $f$, and obtain that

$$\|\mathbf{x_1}(0) - \mathbf{x_2}(0)\|_2 = \|\overline{\mathbf{x_1}}(t) - \overline{\mathbf{x_2}}(t)\|_2 \leq \|\overline{\mathbf{x_1}}(0) - \overline{\mathbf{x_2}}(0)\|_2 \cdot \exp(N \cdot t) =$$

$$= \|\mathbf{x_1}(t) - \mathbf{x_2}(t)\|_2 \cdot \exp(N \cdot t)$$

The proof concludes by rearranging of the left and right hand side. $\qquad\square$

**Lemma 31** (ODE solutions do not cross). *Let $f : \mathbb{R}^d \to \mathbb{R}^d$ be a Lipschitz continuous vector field. Consider the following system of ODEs:*

$$\dot{\mathbf{x}}(t) = f(\mathbf{x}(t))$$

*Consider two initialization points $\mathbf{x_1}(0), \mathbf{x_2}(0) \in \mathbb{R}^d$. Suppose we evolve $\mathbf{x_1}(t), \mathbf{x_2}(t)$ according to the above system. If $\mathbf{x_1}(0) \neq \mathbf{x_2}(0)$ then for any $t \in \mathbb{R}$ it holds that $\mathbf{x_1}(t) \neq \mathbf{x_2}(t)$*

*Proof.* The argument follows from the Picard–Lindelöf existence and uniqueness theorem, which states that for a given initialization $\mathbf{x}(0)$, there exists a unique solution $\mathbf{x}(t)$ to the ODE system

$$\dot{\mathbf{x}}(t) = f(\mathbf{x}(t))$$

$\qquad\square$

**Lemma 32.** *Let $\mathbf{x} \in \mathbb{R}^d$. Denote $\mathbf{x}$'s representation with the orthogonal subspaces $\mathcal{W}_1$ and $\mathcal{W}_2$ to be*

$$\mathbf{x} = \beta_1 \cdot \mathbf{1}_d + \beta_2 \cdot \mathbf{v}$$

*where $\mathbf{v} \in \mathcal{W}_2$ is a unit vector. There exist $i, j \in [d]$ such that*

$$|x_i - x_j| \geq \frac{|\beta_2|}{d}$$

*Proof.* Denote $\mathbf{v}$'s representation with the basis vectors of $\mathcal{W}_2$ to be

$$\mathbf{v} = \sum_{k=2}^{d} \lambda_k(\mathbf{e_1} - \mathbf{e_k}) = \begin{pmatrix} \sum_{k=2}^{d} \lambda_k \\ -\lambda_2 \\ \dots \\ -\lambda_d \end{pmatrix}$$

where $\lambda_2, \ldots, \lambda_d \in \mathbb{R}$. Denote $\Lambda := (\lambda_2, \ldots, \lambda_d)^\top \in \mathbb{R}^{d-1}$. As $\mathbf{v}$ is a unit vector, it holds that

$$1 = \|\mathbf{v}\|_2^2 = (\sum_{k=2}^{d} \lambda_k)^2 + \sum_{k=2}^{d}(-\lambda_k)^2 = (\sum_{k=2}^{d} \lambda_k)^2 + \sum_{k=2}^{d} \lambda_k^2$$

Hence,

$$1 - (\sum_{k=2}^{d} \lambda_k)^2 = \sum_{k=2}^{d} \lambda_k^2$$

Applying the Cauchy-Schwartz inequality we obtain the following:

$$(\sum_{k=2}^{d} \lambda_k)^2 = (\sum_{k=2}^{d} 1 \cdot \lambda_k)^2 = \langle \mathbf{1}_{d-1}, \Lambda \rangle \leq \|\mathbf{1}_{d-1}\|_2^2 \cdot \|\Lambda\|^2 = (d-1)\sum_{k=2}^{d} \lambda_k^2$$

Therefore, we obtain that

$$\sum_{k=2}^{d} \lambda_k^2 = 1 - (\sum_{k=2}^{d} \lambda_k)^2 \geq 1 - (d-1)\sum_{k=2}^{d} \lambda_k^2 \implies d\sum_{k=2}^{d} \lambda_k^2 \geq 1 \implies \sum_{k=2}^{d} \lambda_k^2 \geq \frac{1}{d}$$

Therefore, there exists $i^* \in \{2, \ldots, d\}$ for which $\lambda_{i^*}^2 \geq \frac{1}{d(d-1)}$ and thus $|\lambda_{i^*}| \geq \frac{1}{\sqrt{d(d-1)}}$. If there exists $j \in \{2, \ldots d\}$ for which $\lambda_j$ has a distinct sign than $\lambda_{i^*}$, then it holds that

$$|v_{i^*} - v_j| = |-\lambda_{i^*} + \lambda_j| \geq |\lambda_{i^*} - 0| \geq \frac{1}{\sqrt{d(d-1)}}$$

Otherwise, all entries of $\Lambda$ share the same sign, and so it holds that for any $j \in \{2, \ldots, d\}$

$$|v_1 - v_j| = |\sum_{k=2}^{d} \lambda_k - (-\lambda_j)| = \sum_{k=2}^{d} |\lambda_k| + |\lambda_j| \geq |\lambda_{i^*}| \geq \frac{1}{\sqrt{d(d-1)}}$$

Therefore, there must exist $i, j \in [d]$ for which $|v_i - v_j| \geq \frac{1}{\sqrt{d(d-1)}}$, resulting in the following:

$$|x_i - x_j| = |\beta_1 + \beta_2 \cdot v_i - \beta_1 - \beta_2 \cdot v_j| = |\beta_2| \cdot |v_i - v_j| \geq \frac{|\beta_2|}{\sqrt{d(d-1)}}$$

$\square$

**Proposition 17.** *Let $x, y \in [-s, s]$ for some $s > 0$ such that $|x - y| \geq b$ for some $b > 0$. Then for any $k \in \mathbb{N}_{odd}$ it holds that $|x^k - y^k| \geq (\frac{b}{2})^k$. Additionally, if $y \neq 0$ then $|1 - \frac{x^k}{y^k}| \geq (\frac{b}{2s})^k$.*

*Proof.* Suppose WLOG that $x \geq y$. By the triangle inequality it holds that $\max\{|x|, |y|\} \geq \frac{b}{2}$. If $x \geq 0 \geq y$, then since $k \in \mathbb{N}_{odd}$ it holds that

$$|x^k - y^k| = |x|^k + |y|^k \geq (\frac{b}{2})^k$$

Now suppose that $x, y \geq 0$. Then we have that

$$|x^k - y^k| = x^k - y^k \geq (y + b)^k - y^k \geq b^k \geq (\frac{b}{2})^k$$

The case of $x, y \leq 0$ is identical. As for the second inequality, If $y \neq 0$, we get

$$|1 - \frac{x^k}{y^k}| = |\frac{y^k - x^k}{y^k}| = \frac{|x^k - y^k|}{|y^k|} \geq \frac{(\frac{b}{2})^k}{|y^k|}$$

Since $y \in [-s, s]$ we get that $|y^k| \leq s^k$ and so

$$\frac{(\frac{b}{2})^k}{|y^k|} \geq (\frac{b}{2s})^k$$

$\square$

**Proposition 18.** *Let* $x, y, z \in \mathbb{R}$ *such that* $y, x.z \neq 0$, *and let* $\widehat{b} > 0$. *If* $|1 - \frac{x}{y}| \geq \widehat{b}$ *then either* $|1 - \frac{x}{z}| \geq \frac{\widehat{b}}{\widehat{b}+2}$ *or* $|1 - \frac{y}{z}| \geq \frac{\widehat{b}}{\widehat{b}+2}$.

*Proof.* Assume to the contrary that both $|1 - \frac{x}{z}| < \frac{\widehat{b}}{\widehat{b}+2}$ and $|1 - \frac{y}{z}| < \frac{\widehat{b}}{\widehat{b}+2}$. This implies that

$$0 < 1 - \frac{\widehat{b}}{\widehat{b}+2} \leq \frac{x}{z}, \frac{y}{z} < 1 + \frac{\widehat{b}}{\widehat{b}+2}$$

Hence, we get that

$$\frac{1}{\widehat{b}+1} = \frac{2}{2\widehat{b}+2} = \frac{\widehat{b}+2-\widehat{b}}{\widehat{b}+2+\widehat{b}} = \frac{1 - \frac{\widehat{b}}{\widehat{b}+2}}{1 + \frac{\widehat{b}}{\widehat{b}+2}} < \frac{\frac{x}{z}}{\frac{y}{z}} = \frac{x}{y} < \frac{1 + \frac{\widehat{b}}{\widehat{b}+2}}{1 - \frac{\widehat{b}}{\widehat{b}+2}} = \frac{\widehat{b}+2+\widehat{b}}{\widehat{b}+2-\widehat{b}} = \frac{2\widehat{b}+2}{2} = \widehat{b}+1$$

rearranging we obtain

$$-\widehat{b} < \frac{-\widehat{b}}{\widehat{b}+1} = \frac{1}{\widehat{b}+1} - 1 < \frac{x}{y} - 1 < \widehat{b}$$

which implies $|\frac{x}{y} - 1| < \widehat{b}$, contradicting our assumption. $\square$

**Lemma 33.** *Let* $f : \mathbb{R}^d \to \mathbb{R}$ *be a differentiable function. Consider the gradient flow dynamics induced by* $f$, *namely:*

$$\dot{\mathbf{x}}(t) = -\nabla f(\mathbf{x}(t))$$

*Initialized at some* $\mathbf{x_0} \in \mathbb{R}^d$. *Let* $t_1 < t_2$ *be times such that*

$$\{\mathbf{x}(t) : t \in [t_1, t_2]\} \subseteq \{\mathbf{z} \in \mathbb{R}^d : f(z) \geq \min_{\mathbf{y} \in \mathbb{R}^d}(f(\mathbf{y})) + c\}$$

*for some* $c \geq 0$ *and* $f$ *satisfies the PL condition with some coefficient* $\mu > 0$ *in* $\{\mathbf{x}(t) : t \in [t_1, t_2]\}$. *Then*

$$f(\mathbf{x}(t_1)) - f(\mathbf{x}(t_2)) \geq 2(t_2 - t_1) \cdot \mu \cdot c$$

*Proof.* By the fundamental theorem for line integrals we have

$$f(\mathbf{x}(t_1)) - f(\mathbf{x}(t_2)) = -\int_{t_1}^{t_2} \langle \nabla f(\mathbf{x}(\tau)), \dot{\mathbf{x}}(\tau) \rangle d\tau = \int_{t_1}^{t_2} \|\nabla f(\mathbf{x}(\tau))\|^2 d\tau$$

applying the PL condition and Equation (31) we get the required result. $\square$

**Lemma 34.** *Let* $g : \mathbb{R}^d \to \mathbb{R}$ *be a continuous function and let* $r > 0$ *and let* $B$ *be a compact set such that* $\mathbf{0}_d \in B$. *For any* $\psi \geq 0$ *denote the minimum value of* $g$ *over Diff($\psi$)* $\cap B$ *as*

$$f(\psi) := \min_{\mathbf{x} \in Diff(\psi) \cap B} g(\mathbf{x})$$

*It holds that* $f$ *is right side continuous in* $\psi = 0$.

*Proof.* Recalling the definition of $\mathrm{Diff}(\psi)$ (Equation (17)), we have that for any $\psi \geq 0$ the set $\mathrm{Diff}(\psi) \cap B$ is compact, thus $f(\psi)$ is properly defined for any $\psi \geq 0$ (as by continuity $g$ attains a minimum over the set). Next, note that $f$ is non-increasing since for any $\psi_2 \geq \psi_1 \geq 0$ it holds that

$$\mathrm{Diff}(\psi_1) \cap B \subseteq \mathrm{Diff}(\psi_2) \cap B \implies f(\psi_1) = \min_{\mathbf{x} \in \mathrm{Diff}(\psi_1) \cap B} g(\mathbf{x}) \geq \min_{\mathbf{x} \in \mathrm{Diff}(\psi_2) \cap B} g(\mathbf{x}) = f(\psi_2)$$

Let $\{\psi_n\}_{n=1}^\infty$ be a non-increasing sequence of non-negative reals for which

$$\lim_{n \to \infty} \psi_n = 0$$

As $f$ is non-increasing, the sequence $\{f(\psi_n)\}_{n=1}^\infty$ is monotonically non-decreasing and upper bounded by $f(0)$. Hence, the limit $R := \lim_{n \to \infty} f(\psi_n)$ exists and satisfies $R \leq f(0)$. For any $\psi \geq 0$ we let $\mathbf{x}_\psi$ be a minimizer of $g$ over $\mathrm{Diff}(\psi) \cap B$, *i.e.*

$$\mathbf{x}_\psi \in \operatorname*{argmin}_{\mathbf{x} \in \mathrm{Diff}(\psi) \cap B} g(\mathbf{x})$$

The set $B$ is compact and so the sequence $\{\mathbf{x}_{\psi_n}\}_{n=1}^\infty$ has a convergent subsequence $\{\mathbf{x}_{\psi_{n_k}}\}_{k=1}^\infty$. Denote its limit

$$\lim_{k \to \infty} \mathbf{x}_{\psi_{n_k}} =: \mathbf{x}^* \in \overline{B_r}(\mathbf{0}_d)$$

Assume on the contrary that $\mathbf{x}^* \notin \mathrm{Diff}(0)$. Hence by definition of $\mathrm{Diff}(0)$ it holds that

$$\widetilde{\psi} := \max_{i,j \in [d]} |x_i^* - x_j^*| > 0$$

However, since $\lim_{n \to \infty} \psi_n = 0$ there exists $\widetilde{k} \in \mathbb{N}$ such that for any $k \in \mathbb{N}$ such that $k \geq \widetilde{k}$ it holds that $\psi_{n_k} \leq \frac{\widetilde{\psi}}{2}$ and thus

$$\mathbf{x}_{\psi_{n_k}} \in \mathrm{Diff}(\psi_{n_k}) \subseteq \mathrm{Diff}(\frac{\widetilde{\psi}}{2}) \wedge \mathbf{x}^* \notin \mathrm{Diff}(\frac{\widetilde{\psi}}{2})$$

This results in $\lim_{k \to \infty} \mathbf{x}_{\psi_{n_k}} \neq \mathbf{x}^*$ in contradiction. Therefore we obtain $\mathbf{x}^* \in \mathrm{Diff}(0) \cap B$. By $g$'s continuity and $f$'s definition we thus obtain the following

$$\lim_{k \to \infty} g(\mathbf{x}_{\psi_{n_k}}) = g(\lim_{k \to \infty} \mathbf{x}_{\psi_{n_k}}) = g(\mathbf{x}^*) \geq f(0)$$

Per $\mathbf{x}_\psi$'s definition and since $\lim_{n \to \infty} f(\psi_n) = R$ we also obtain

$$\lim_{k \to \infty} g(\mathbf{x}_{\psi_{n_k}}) = \lim_{k \to \infty} f(\psi_{n_k}) = R$$

*i.e.*, $R \geq f(0)$. Overall we obtain that $R \leq f(0) \leq R$, hence $R = f(0)$. The above result is satisfied for any sequence $\{\psi_n\}_{n=1}^\infty$, hence $\lim_{\psi \to 0^+} f(\psi) = f(0)$ as required. $\qquad\square$

**Lemma 35.** *Let $f : \mathbb{R}^d \to \mathbb{R}^d$ be a $C^1$ and Lipschitz vector field. Consider the system of ODEs given by*

$$\dot{\mathbf{x}}(t) = f(\mathbf{x}(t))$$

*Consider two initialization points $\mathbf{x_1}(0), \mathbf{x_2}(0) \in \mathbb{R}^d$. For any $t \geq 0$ we use $\lambda_{max}(t)$ to denote the maximum over the line segment between $\mathbf{x_1}(t)$ and $\mathbf{x_2}(t)$ of the maximal eigenvalue of the jacobian of $f$, i.e.*

$$\lambda_{max}(t) := \max_{k \in [0,1]} \lambda_{max}\left(\nabla f\big(k \cdot \mathbf{x_1}(t) + (1 - k) \cdot \mathbf{x_2}(t)\big)\right)$$

*For any $t \geq 0$ it holds that*

$$\|\mathbf{x_1}(t) - \mathbf{x_2}(t)\|_2 \leq \exp\left(\int_0^t \lambda_{max}(\tau)d\tau\right) \cdot \|\mathbf{x_1}(0) - \mathbf{x_2}(0)\|_2$$

*Proof.* First note that since $f$ is Lipschitz, $\lambda_{max}(t)$ is defined for any $t \geq 0$. Next, if $\mathbf{x_1}(0) = \mathbf{x_2}(0)$ then the trajectories coincide and so the claim trivially follows. Suppose $\mathbf{x_1}(0) \neq \mathbf{x_2}(0)$. By definition, we have that

$$\frac{d}{dt}(\mathbf{x_1}(t) - \mathbf{x_2}(t)) = f(\mathbf{x_1}(t)) - f(\mathbf{x_2}(t))$$

As $f$ is $C^1$, we obtain by the mean value theorem (see Sahoo and Riedel (1998)) that there exists $k \in [0,1]$ for which

$$f(\mathbf{x_1}(t)) - f(\mathbf{x_2}(t)) = \nabla f(k \cdot \mathbf{x_1}(t) + (1-k) \cdot \mathbf{x_2}(t))(\mathbf{x_1}(t) - \mathbf{x_2}(t))$$

Additionally, by the chain rule we also have

$$\frac{d}{dt}\|\mathbf{x_1}(t) - \mathbf{x_2}(t)\|_2 = \frac{d}{dt}\sqrt{(\mathbf{x_1}(t) - \mathbf{x_2}(t))^\top (\mathbf{x_1}(t) - \mathbf{x_2}(t))} =$$

$$= \frac{1}{2\|\mathbf{x_1}(t) - \mathbf{x_2}(t)\|_2} \frac{d}{dt}(\mathbf{x_1}(t) - \mathbf{x_2}(t))^\top (\mathbf{x_1}(t) - \mathbf{x_2}(t)) =$$

$$= \frac{2}{2\|\mathbf{x_1}(t) - \mathbf{x_2}(t)\|_2} \cdot (\mathbf{x_1}(t) - \mathbf{x_2}(t))^\top \frac{d}{dt}(\mathbf{x_1}(t) - \mathbf{x_2}(t)) =$$

$$= \frac{1}{\|\mathbf{x_1}(t) - \mathbf{x_2}(t)\|_2} \cdot (\mathbf{x_1}(t) - \mathbf{x_2}(t))^\top \left( f(\mathbf{x_1}(t)) - f(\mathbf{x_2}(t)) \right)$$

Plugging the above yields

$$\frac{d}{dt}\|\mathbf{x_1}(t) - \mathbf{x_2}(t)\|_2 = \frac{(\mathbf{x_1}(t) - \mathbf{x_2}(t))^\top \nabla f(k \cdot \mathbf{x_1}(t) + (1-k) \cdot \mathbf{x_2}(t))(\mathbf{x_1}(t) - \mathbf{x_2}(t))}{\|\mathbf{x_1}(t) - \mathbf{x_2}(t)\|_2} =$$

$$= \|\mathbf{x_1}(t) - \mathbf{x_2}(t)\|_2 \cdot \frac{(\mathbf{x_1}(t) - \mathbf{x_2}(t))^\top \nabla f(k \cdot \mathbf{x_1}(t) + (1-k) \cdot \mathbf{x_2}(t))(\mathbf{x_1}(t) - \mathbf{x_2}(t))}{(\mathbf{x_1}(t) - \mathbf{x_2}(t))^\top (\mathbf{x_1}(t) - \mathbf{x_2}(t))} = (*)$$

The right term is bound by the Rayleigh quotient (see Horn and Johnson (1985)), and so the above can be bound by

$$(*) \leq \|\mathbf{x_1}(t) - \mathbf{x_2}(t)\|_2 \cdot \lambda_{max}\left(\nabla f(k \cdot \mathbf{x_1}(t) + (1-k) \cdot \mathbf{x_2}(t))\right) \leq \|\mathbf{x_1}(t) - \mathbf{x_2}(t)\|_2 \cdot \lambda_{max}(t)$$

where the last inequality stems from $\lambda_{max}(t)$'s definition. Dividing both sides by $\|\mathbf{x_1}(t) - \mathbf{x_2}(t)\|_2$ and integrating w.r.t time yields the following

$$\ln(\|\mathbf{x_1}(t) - \mathbf{x_2}(t)\|_2) - \ln(\|\mathbf{x_1}(0) - \mathbf{x_2}(0)\|_2) = \int_0^t \frac{\frac{d}{d\tau}\|\mathbf{x_1}(\tau) - \mathbf{x_2}(\tau)\|_2}{\|\mathbf{x_1}(\tau) - \mathbf{x_2}(\tau)\|_2} d\tau \leq \int_0^t \lambda_{max}(\tau)d\tau$$

Reorganizing the inequality and taking exponents yields

$$\|\mathbf{x_1}(t) - \mathbf{x_2}(t)\|_2 \leq \exp\left(\int_0^t \lambda_{max}(\tau)d\tau + \ln(\|\mathbf{x_1}(0) - \mathbf{x_2}(0)\|_2)\right) =$$

$$= \exp\left(\int_0^t \lambda_{max}(\tau)d\tau\right) \cdot \|\mathbf{x_1}(0) - \mathbf{x_2}(0)\|_2$$

as required. $\qquad\square$

**Lemma 36.** *Let $f : \mathbb{R}^d \to \mathbb{R}$ be a continuous non-negative function with $\min_{\mathbf{x} \in \mathbb{R}^d} f(\mathbf{x}) = 0$. Denote $X^* := \{\mathbf{x} \in \mathbb{R}^d : f(\mathbf{x}) = 0\}$. Suppose that the 1 sub-level set of $f$ defined as $\mathcal{L}_1(f) := \{\mathbf{x} \in \mathbb{R}^d : f(\mathbf{x}) \leq 1\}$ is compact. Then for any $\delta > 0$ there exists $\eta > 0$ such that for any $\mathbf{x} \in \mathbb{R}^d$ if $f(\mathbf{x}) \leq \eta$ then $Dist(\mathbf{x}, X^*) \leq \delta$.*

*Proof.* Assume on the contrary that there exists $\delta > 0$ such that for any $\epsilon > 0$ there exists $\mathbf{x}_\eta \in \mathbb{R}^d$ for which $f(\mathbf{x}_\eta) \leq \delta$ and $Dist(\mathbf{x}_\eta, X^*) > \delta$. Consider the sequence $\{\mathbf{x}_{\frac{1}{n}}\}_{n=1}^\infty$. For any $n \in \mathbb{N}$ it holds that $f(\mathbf{x}_{\frac{1}{n}}) \leq \frac{1}{n}$ and $Dist(\mathbf{x}_{\frac{1}{n}}, X^*) > \delta$, therefore it holds that

$$\lim_{n \to \infty} f(\mathbf{x}_{\frac{1}{n}}) = 0$$

The sub-level set $\mathcal{L}_1(f)$ is compact and satisfies $\mathbf{x}_{\frac{1}{n}} \in \mathcal{L}_1(f)$ for any $n \in \mathbb{N}$, hence the sequence $\{\mathbf{x}_{\frac{1}{n}}\}_{n=1}^{\infty}$ is bounded. Therefore, the sequence has a convergent subsequence $\{\mathbf{x}_{\frac{1}{n_k}}\}_{k=1}^{\infty}$ with some limit $\mathbf{x}^* := \lim_{k \to \infty} \mathbf{x}_{n_k}$. By $f$'s continuity we get that

$$f(\mathbf{x}^*) = f(\lim_{k\to\infty} \mathbf{x}_{n_k}) = \lim_{k\to\infty} f(\mathbf{x}_{n_k}) = \lim_{n\to\infty} f(\mathbf{x}_n) = 0$$

*i.e.*, $\mathbf{x}^* \in X^*$. This is a contradiction since all $\mathbf{x}_{\frac{1}{n}}$ must remain at distance at least $\delta$ from $\mathbf{x}^*$ on the one hand, and $\mathbf{x}_{\frac{1}{n_k}}$ converges to $\mathbf{x}^*$ on the other hand. $\qquad\square$

**Lemma 37.** *Let $T, S > 0$, $n \in \mathbb{N}$ and $x^* \in (0,1)$. There exist $q_1, w_1 \in (0, x^*)$ and $q_2, w_2 \in (\frac{1}{2}, 1)$ such that for any $x \in (q_1, w_1)$ and $y \in (q_2, w_2)$ it holds that*

$$1 - y \leq Tx$$

*and*

$$x \leq \frac{S}{y}(1 - y^n)^{\frac{1}{n}}$$

*Proof.* First note that since $T > 0$, the first requirement is equivalent to having

$$\frac{1-y}{T} \leq x$$

Let $y \in (0,1)$. It holds that

$$\lim_{y\to 1^-} \frac{S}{y}(1-y^n)^{\frac{1}{n}} = \lim_{y\to 1^-} \frac{S}{y}\left((1-y)\sum_{i=0}^{n-1} y^i\right)^{\frac{1}{n}} = S \cdot n^{\frac{1}{n}} \cdot \lim_{y\to 1^-}(1-y)^{\frac{1}{n}} = 0$$

Therefore, there exists $y' \in (0,1)$ such that for any $y \in [y', 1)$ it holds that

$$\frac{S}{y}(1-y^n)^{\frac{1}{n}} \leq x^*$$

On the other hand, it also holds that

$$\frac{\frac{S}{\frac{y+1}{2}}\left(1 - (\frac{y+1}{2})^n\right)^{\frac{1}{n}}}{\frac{1-y}{T}} = \frac{2TS}{y+1} \cdot \frac{\left((1 - \frac{y+1}{2})\sum_{i=0}^{n-1}(\frac{y+1}{2})^i\right)^{\frac{1}{n}}}{1-y} =$$

$$= \frac{2TS\left(\sum_{i=0}^{n-1}(\frac{y+1}{2})^i\right)^{\frac{1}{n}}}{y+1} \cdot \left(\frac{1}{2}\right)^{\frac{1}{n}} \cdot \frac{(1-y)^{\frac{1}{n}}}{1-y}$$

Hence, taking the limit as $y \to 1^-$ we obtain that

$$\lim_{y\to 1^-} \frac{\frac{S}{\frac{y+1}{2}}\left(1 - (\frac{y+1}{2})^n\right)^{\frac{1}{n}}}{\frac{1-y}{T}} = \lim_{y\to 1^-} \frac{2TS\left(\sum_{i=0}^{n-1}(\frac{y+1}{2})^i\right)^{\frac{1}{n}}}{y+1} \cdot \left(\frac{1}{2}\right)^{\frac{1}{n}} \cdot \frac{(1-y)^{\frac{1}{n}}}{1-y} =$$

$$= TS \cdot n^{\frac{1}{n}} \cdot \left(\frac{1}{2}\right)^{\frac{1}{n}} \cdot \lim_{y\to 1^-} \frac{1}{(1-y)^{\frac{n-1}{n}}} = \infty$$

Therefore, there exists $y'' \in (0,1)$ such that for any $y \in [y'', 1)$ it holds that

$$\frac{\frac{S}{\frac{y+1}{2}}\left(1 - (\frac{y+1}{2})^n\right)^{\frac{1}{n}}}{\frac{1-y}{T}} > 2$$

and so $\frac{1-y}{T} < \frac{S}{\frac{y+1}{2}}\left(1 - (\frac{y+1}{2})^n\right)^{\frac{1}{n}}$. Thus, setting $y^* = \max\{\frac{1}{2}, y', y''\}$ we obtain that the interval $\left(\frac{1-y^*}{T}, \frac{S}{\frac{y^*+1}{2}}\left(1 - (\frac{y^*+1}{2})^n\right)^{\frac{1}{n}}\right)$ is not empty and upper bounded by $x^*$. Additionally, for any $y \in (y^*, \frac{y^*+1}{2})$ the following holds:

$$y^* < y \implies \frac{1-y^*}{T} > \frac{1-y}{T}$$

$$\frac{y^* + 1}{2} > y \implies \frac{S}{\frac{y^*+1}{2}} \big(1 - (\frac{y^*+1}{2})^n\big)^{\frac{1}{n}} < \frac{S}{y}(1 - y^n)^{\frac{1}{n}}$$

Hence the interval $\left(\frac{1-y^*}{T}, \frac{S}{\frac{y^*+1}{2}}\big(1 - (\frac{y^*+1}{2})^n\big)^{\frac{1}{n}}\right)$ is contained within the interval $\big(\frac{1-y}{T}, \frac{S}{y}(1 - (y^n)^{\frac{1}{n}}\big)$. Noting that $\frac{y^*+1}{2} < 1$, we complete the proof by setting

$$q_1 = \frac{1-y^*}{T}, \ w_1 = \frac{S}{\frac{y^*+1}{2}}\big(1 - (\frac{y^*+1}{2})^n\big)^{\frac{1}{n}}$$

$$q_2 = y^*, \ w_2 = \frac{y^*+1}{2}$$

$\square$

## E    EXTENSIONS OF THEOREM 1

In this appendix, we outline extensions of Theorem 1 (Section 3.2) to settings in which: *(i)* the teacher SSM is of arbitrary dimension $d^* \geq 2$; *(ii)* the input and output matrices of the teacher SSM vary; *(iii)* the input and output matrices of the student SSM are learned (as opposed to being fixed throughout training); and *(iv)* the training set $\mathcal{S}$ varies. We also account for limitations of the above extensions.

**Teacher of arbitrary dimension.**    For any $d^* \geq 2$, consider the following parameter assignments for the teacher SSM:

$$A^* = \begin{pmatrix} 1 & 0 & \cdots & 0 \\ 0 & 0 & \cdots & 0 \\ \vdots & \vdots & \ddots & \vdots \\ 0 & 0 & \cdots & 0 \end{pmatrix} \in \mathbb{R}^{d^*,d^*} \ , \ \ B^* = \begin{pmatrix} 1 \\ \sqrt{\frac{d-1}{d^*-1}} \\ \sqrt{\frac{d-1}{d^*-1}} \\ \vdots \\ \sqrt{\frac{d-1}{d^*-1}} \end{pmatrix} \in \mathbb{R}^{d^*,1} \ , \ \ C^* = \begin{pmatrix} 1 \\ \sqrt{\frac{d-1}{d^*-1}} \\ \sqrt{\frac{d-1}{d^*-1}} \\ \vdots \\ \sqrt{\frac{d-1}{d^*-1}} \end{pmatrix}^{\top} \in \mathbb{R}^{1,d^*}.$$

In this setting, the mapping $\phi_{(A^*,B^*,C^*)}(\cdot)$ realized by the teacher SSM is the same as it is in the setting defined by Equation (33) (where the teacher has dimension $d^* = 2$). Accordingly, Theorem 1 and its proof apply as is to the current setting.

**Varying teacher input and output matrices.**    Given any teacher SSM $(A^*, B^*, C^*)$ with which Theorem 1 holds (including a high dimensional teacher as described above), a similar result holds with the teacher SSM $(A^*, \alpha_1 B^*, \alpha_2 C^*)$, where $\alpha_1, \alpha_2 \in \mathbb{R}_{\neq 0}$ are arbitrary. Indeed, if we likewise scale the values of the (fixed) student parameters $B$ and $C$, *i.e.* we replace $B$ by $\alpha_1 B$ and $C$ by $\alpha_2 C$, then for every sequence $\mathbf{x}$:

$$\phi_{(A^*,\alpha_1 B^*,\alpha_2 C^*)}(\mathbf{x}) = \alpha_1 \alpha_2 \phi_{(A^*,B^*,C^*)}(\mathbf{x})$$

and likewise:

$$\phi_{(A,\alpha_1 B,\alpha_2 C)}(\mathbf{x}) = \alpha_1 \alpha_2 \phi_{(A,B,C)}(\mathbf{x}).$$

The training loss and its derivatives thus scale by a positive factor, and so do generalization errors (Definition 1). Accordingly, the proof of Theorem 1 carries through.

**Learned student input and output matrices.**    Below we outline a modification of Theorem 1 that accounts for a setting in which the input and output matrices of the student SSM are learned. Suppose these input and output matrices—$B$ and $C$, respectively—are learned with a learning rate (step size) that may be different from the learning rate of the student's state transition matrix $A$. Formally,

suppose the optimization trajectory $(A(\cdot), B(\cdot), C(\cdot))$ is governed by the following dynamics:

$$\dot{A}(t) = -\frac{\partial}{\partial A}\ell(A(t), B(t), C(t); \mathcal{S})$$

$$\dot{B}(t) = -\eta \cdot \frac{\partial}{\partial B}\ell(A(t), B(t), C(t); \mathcal{S}) \ , \ \ t \in \mathbb{R}_{\geq 0}\,, \tag{32}$$

$$\dot{C}(t) = -\eta \cdot \frac{\partial}{\partial C}\ell(A(t), B(t), C(t); \mathcal{S})$$

where $\eta > 0$ represents the ratio between the learning rate of $B$ and $C$, and the learning rate of $A$. Consider a trajectory induced by Equation (32), and a corresponding trajectory that emanates from the same initialization, but where only $A$ is learned (or equivalently, where $\eta$ in Equation (32) is replaced by zero). Arguments similar to those used in the proof of Theorem 1 can be used to show that the divergence between these two trajectories is upper bounded by a quantity that depends on $\eta$, and in particular tends to zero as $\eta$ does. Accordingly, if $\eta$ is sufficiently small, generalization errors attained when $A$, $B$ and $C$ are learned jointly (*i.e.*, when optimization is governed by Equation (32)), are close to those attained when only $A$ is learned. Theorem 1—which applies to a setting where only $A$ is learned—thus translates to a result that applies to a setting where $B$ and $C$ are also learned.

**Varying training set.** Theorem 1 proves existence of a specific training set $\mathcal{S}$ under which gradient flow converges to a generalizing solution. As we show below, one can extend this result to a much larger class of training sets.

**Theorem 5.** *Consider the teacher SSM given by*

$$A^* = \begin{pmatrix} a^* & 0 \\ 0 & 0 \end{pmatrix} \ , \ \ B^* = \begin{pmatrix} 1 & \sqrt{d-1} \end{pmatrix}^\top \ , \ \ C^* = \begin{pmatrix} 1 & \sqrt{d-1} \end{pmatrix}. \tag{33}$$

*Suppose we learn the transition matrix $A$ of the student SSM via gradient flow, and its input and output matrices $B(\cdot)$ and $C(\cdot)$ are fixed at $\mathbf{1}_d$ and $\mathbf{1}_d^\top$, respectively. Let $\mathcal{S} = \{(\mathbf{x}_i, y_i)\}_{i=1}^n$ be a training set such that $(\mathbf{x}^{(i)}, y^{(i)}) \in \mathbb{R}^\kappa \times \mathbb{R}$, where for all $i \in [n]$ the last two entries of $\mathbf{x}^{(i)}$ equal zero, and the rest are positive. Then, for any $\kappa' \in \mathbb{N}$ and $\epsilon, \delta > 0$, there exists a time $T := T(\epsilon, \delta) > 0$ and an open set $\mathcal{I} := \mathcal{I}(\epsilon, \delta)$ such that gradient flow initialized in $\mathcal{I}$ satisfies:*

$$\ell(A(T)) < \delta \ \text{ and } \ Gen_{L'}(A(T)) < \epsilon.$$

*Proof.* Consider the point $A_0 = (a_0, 0, , , ., 0)$ where $0 < a_0 < a^*$. We will first show that if we initialize at $A_0$, gradient flow will converge to $(a^*, 0, , , ., 0)$, and therefore achieve perfect generalization. Indeed, writing 3 in terms of the entries of $A$ we get:

$$\ell(A(t)) = \frac{1}{n}\sum_{i=1}^n \left( \sum_{l=2}^{\kappa-1}(a^*)^l x_{\kappa-l}^{(i)} - \sum_{l=2}^{\kappa-1}\left( \sum_{j=1}^d a_j(t)^l \right) x_{\kappa-l}^{(i)} \right)^2$$

The derivative of $\ell(A(t))$ with respect to $a_p$ is therefore:

$$\frac{\partial \ell}{\partial a_p} = -\frac{2}{n}\sum_{i=1}^n \left( \sum_{l=2}^{\kappa-1}(a^*)^l x_{\kappa-l}^{(i)} - \sum_{l=2}^{\kappa-1}\left( \sum_{j=1}^d a_j^l \right) x_{\kappa-l}^{(i)} \right) \left( \sum_{l=2}^{\kappa-1} l\, a_p^{l-1} x_{\kappa-l}^{(i)} \right)$$

For $j > 2$, $a_j(0) = 0$ and thus $\dot{a}_j(0) = -\frac{\partial \ell}{\partial a_j}(0) = 0$. Therefore for all $j > 2$, $a_j(t) = 0$ for all $t > 0$. Hence it suffices to show that $a_1(t)$ converges to $a^*$ as $t \to \infty$. To see this, note that because $a_j(t) = 0$ for all $t > 0$ the dynamics simplify to

$$\dot{a}_1(t) = -\frac{\partial \ell}{\partial a_1}(t) = \frac{2}{n}\sum_{i=1}^n \left( \sum_{l=2}^{\kappa-1} x_{\kappa-l}^{(i)}((a^*)^l - a_1(t)^l) \right) \left( \sum_{l=2}^{\kappa-1} l\, a_1(t)^{l-1} x_{\kappa-l}^{(i)} \right)$$

For all $i \in [n]$, at $t = 0$ it holds that

$$\left( \sum_{l=2}^{\kappa-1} x_{L-l}^{(i)}((a^*)^l - a_1(t)^l) \right) > 0$$

Additionally, by the positivity of the (non zero) entries of $\mathbf{x}^{(i)}$, any of the above terms equals zero if and only if $a_1 = a^*$ (in which case the derivative vanishes). Therefore they must remain positive for all $t > 0$. The term

$$\left( \sum_{l=2}^{\kappa-1} l\, a_1(t)^{l-1} x_{\kappa-l}^{(i)} \right)$$

is likewise positive by a similar argument. It follows that $a_1(t)$ is monotonically increasing and bounded from above, thus it converges. Furthermore, the limit must be a point where the derivative vanishes, and therefore it must equal $a^*$.

Because $A(t)$ converges to $(a^*, ...., 0)$ when initialized at $A_0$, for any $\epsilon, \delta > 0$ there exists $T := T(\epsilon, \delta) > 0$ such that $\ell(A(T)) < \frac{\epsilon}{2}$, $Gen_{\kappa'}(A(T)) < \frac{\delta}{2}$. Now by the continuity of $\ell$, $Gen_{\kappa'}$ and by Lemma 30, there exists an open set $\mathcal{I} := \mathcal{I}(\epsilon, \delta)$ such that, if we initialize gradient flow from $\widetilde{A}(0) \in \mathcal{I}$, resulting in the trajectory $\widetilde{A}(t)$, we get that $\|A(T) - \widetilde{A}(T)\|_2$ is sufficiently small to ensure that

$$\ell(\widetilde{A}(T)) < \delta$$
$$Gen_{L'}(\widetilde{A}(T)) < \epsilon$$

as required. $\qquad\square$

**Limitations.** While the abovedescribed extensions of Theorem 1 broaden its scope, they still entail limitations which are important to acknowledge. In general, Theorem 1 is an existence result, and even under the extensions above, it applies to specific settings. More specifically, it does not account for: large initializations and large learning rates (for $A(\cdot)$, and even more so for $B(\cdot)$ and $C(\cdot)$); many values for the teacher parameters $(A^*, B^*, C^*)$; non-diagonal SSMs; and more. Further extending Theorem 1 is regarded as an important direction for future research.

# F  FURTHER EXPERIMENTS

## F.1  DYNAMICAL CHARACTERIZATION

In Section 4.1 we provided experiments that corroborate the implication of the dynamical characterization presented in Section 3.1 and demonstrate the implicit bias to greedy low rank learning of the state transition matrix $A$ under some, but not all, choices of training sequences. In this appendix we report additional experiments, including other settings, that demonstrate this phenomenon. Figure 3 extends the experiments reported in Figure 2 to longer training sequences. Figures 4 to 6 showcase similar experiments to the ones in Figure 2, where the training sequences are labeled by teachers of higher ranks. Figures 7 and 8 report the results achieved with different random seeds in the settings of Figures 2 and 3, respectively. A classical continuous surrogate for the matrix rank is the *effective rank* (Roy and Vetterli, 2007). Figures 9 to 15 present the effective rank of the transition matrix $A$ throughout optimization in the settings of Figures 2 to 8, respectively, underlining the low effective rank caused by greedy low rank learning. Finally, Figures 16 to 19 report the values of $\gamma^{(0)}(t)$ (as defined in Proposition 1) observed during optimization in the settings of Figures 2, 3, 7, and 8, respectively, showcasing that larger absolute values of $\gamma^{(0)}(t)$ do not correspond to greedy low rank learning, whereas lower absolute values do.

## F.2  CLEAN-LABEL POISONING

In Section 4.2 we provided experiments which corroborate our theory in Section 3.2 and emphasize the potential generalization failures SSMs are susceptible to when adding special training sequences. In this appendix we report additional experiments demonstrating this phenomenon. Table 3 demonstrates clean-label poisoning of SSMs in the same settings as the ones in Table 1, except that the sequences used to train the models were longer.

# G  IMPLEMENTATION DETAILS

This appendix provides implementation details omitted from Sections 4 and F. Code for reproducing all of our experiments will be made publicly available.

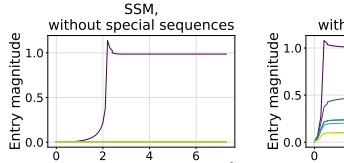 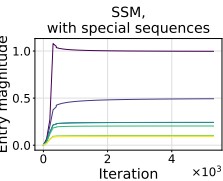 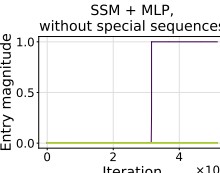 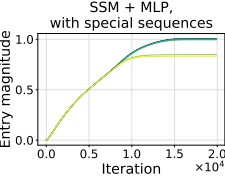

Figure 3: Demonstration of the dynamical characterization derived in Proposition 1—optimization of an SSM, trained individually or as part of a non-linear neural network, implicitly induces greedy learning of the (diagonal) entries of the state transition matrix $A$ under some, but not all, choices of training sequences. This figure is identical to Figure 2, except that the sequences used to train the models were longer, namely, of sequence length 10 as opposed to 6. For further details see Figure 2 as well as Section G.1.

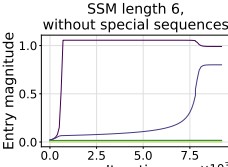 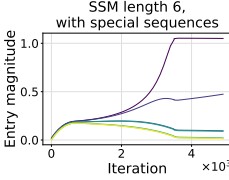 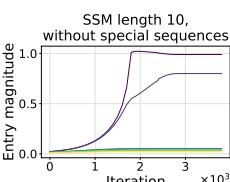 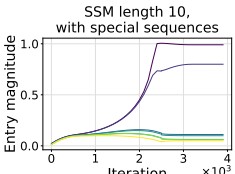

Figure 4: Demonstration of the dynamical characterization derived in Proposition 1—optimization of an individually trained SSM implicitly induces greedy learning of the (diagonal) entries of the state transition matrix $A$ under some, but not all, choices of training sequences. First two plots (left) and last two plots are identical to the first two plots in Figures 2 and 3 respectively, except that the teacher used to label the training sequences is of dimension $d^* = 2$ (as opposed to $d^* = 1$). For further details see Figures 2 and 3 as well as Section G.1.

### G.1 DYNAMICAL CHARACTERIZATION

In this Appendix we provide implementation details for the experiments provided in Sections 4.1 and F.1. All experiments were implemented using Keras (Chollet et al., 2015) and were carried out using a single Nvidia RTX 2080 Ti GPU.

#### G.1.1 STANDALONE SSM

**Models.** In the experiments reported in Figures 2, 3, 7, and 8 where a standalone SSM was trained, we used a teacher SSM model of dimension $d^* = 1$ that was set with the parameters

$$A^* = 1 \ , \ B^* = 1 \ , \ C^* = 1$$

We used student SSM models that were trained end to end (*i.e.* $B(\cdot)$ and $C(\cdot)$ were not fixed). The student models had dimension $d = 10$ in the original experiments (Figures 2 and 7), and dimension $d = 20$ in the experiments with longer sequences (Figures 3 and 8).

Next, we detail the models used in the experiments with teachers of higher ranks (Figures 4 to 6). In the experiments reported in Figure 4 we used a teacher SSM model of dimension 2 that was set with the parameters

$$A^* = \begin{pmatrix} 0.99 & 0 \\ 0 & 0.8 \end{pmatrix} \ , \ B^* = \begin{pmatrix} 1 & 1 \end{pmatrix}^\top \ , \ C^* = \begin{pmatrix} 1 & 1 \end{pmatrix}$$

In the experiments reported in Figure 5 we used a teacher SSM model of dimension 3 that was set with the parameters

$$A^* = \begin{pmatrix} 0.99 & 0 & 0 \\ 0 & 0.8 & 0 \\ 0 & 0 & 0.5 \end{pmatrix} \ , \ B^* = \begin{pmatrix} 1 & 1 & 1 \end{pmatrix}^\top \ , \ C^* = \begin{pmatrix} 1 & 1 & 1 \end{pmatrix}$$

In the experiments reported in Figure 6 we used a teacher SSM model of dimension 3 that was set with the parameters

$$A^* = \begin{pmatrix} 0.99 & 0 & 0 & 0 \\ 0 & 0.8 & 0 & 0 \\ 0 & 0 & 0.5 & 0 \\ 0 & 0 & 0 & 0.3 \end{pmatrix} \ , \ B^* = \begin{pmatrix} 1 & 1 & 1 & 1 \end{pmatrix}^\top \ , \ C^* = \begin{pmatrix} 1 & 1 & 1 & 1 \end{pmatrix}$$

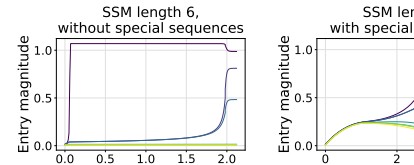 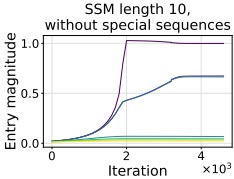 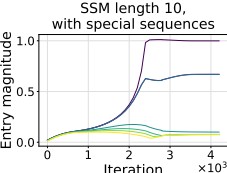

Figure 5: Demonstration of the dynamical characterization derived in Proposition 1. This figure is identical to Figure 4 except that the teacher used to label the training sequences is of dimension $d^* = 3$. For further details see Figure 4 and Section G.1.

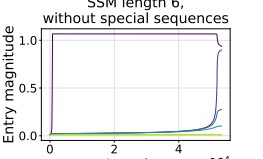 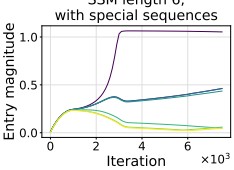 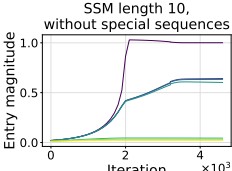 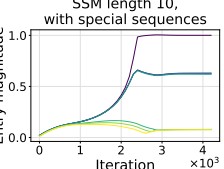

Figure 6: Demonstration of the dynamical characterization derived in Proposition 1. This figure is identical to Figure 4 except that the teacher used to label the training sequences is of dimension $d^* = 4$. For further details see Figure 4 and Section G.1.

We used student SSM models whose input and output matrices $B(\cdot)$ and $C(\cdot)$ were fixed at $\mathbf{1}_d$ and $\mathbf{1}_d^\top$ respectively (due to computational limitations). The student models had dimension $d = 10$ in the experiments of shorter sequence length and dimension $d = 20$ in the experiments of longer sequence length.

**Data.** In all experiments we used the respective teachers to generate the labels for training sequences. Additionally, we used training sequences of one of two types ("baseline" *vs.* "special"), where each type had designated indices of non-zero entries. Table 4 specifies which non-zero indices were present in each sequence type for each experiment. Training sequences of both types had their non-zero entries sampled i.i.d from $\mathcal{N}(0, 1)$. Table 5 specifies how many training sequences of each type were used in each experiment.

**Initialization.** In all experiments we initialized the student's $A$, $B$ and $C$ parameter matrices in a manner that was inspired by the initialization set $\mathcal{I}$ of Theorem 1.

To initialize (the diagonal) $A$ in each experiment we first sampled $d$ i.i.d entries from $\mathcal{N}(0, \texttt{sd\_A})$, took their absolute values and then arranged them in descending order. Then, we set the second entry to have the first entry's value minus a constant $\texttt{diff}$. This was done to reflect the near zero initialization on the one hand and the proximity to the reference trajectory on the other hand. In the experiments reported in Figures 5 and 6 we naturally extended this procedure by setting the third entry to have the first entry's value minus $1.01 \cdot \texttt{diff}$ in both experiments, and the fourth entry to have the first entry's value minus $1.05 \cdot \texttt{diff}$ in the latter. Table 6 report the values of $\texttt{sd\_A}$ and $\texttt{diff}$ used in each experiment.

To initialize $B$ in each experiment we first sampled $d$ i.i.d entries from $\mathcal{N}(0, \texttt{sd\_B\_C})$, took their absolute values and then arranged them in descending order. Then, we set the second entry to have the first entry's value minus constant $\texttt{diff}$. This was done to reflect the near zero initialization on the one hand and the proximity to the reference trajectory on the other hand. In the experiments reported in Figures 5 and 6 we naturally extended this procedure by setting the third entry to have the first entry's value minus $1.01 \cdot \texttt{diff}$ in both experiments, and the fourth entry to have the first entry's value minus $1.05 \cdot \texttt{diff}$ in the latter. To initialize $C$ we followed the same procedure, without modifying the second to potentially fourth entries. Note that in the experiments reported in Figures 4 to 6 the input and output matrices $B(\cdot)$ and $C(\cdot)$ were not trained. Table 6 report the values of $\texttt{sd\_B\_C}$ used in each experiment.

**Optimization.** In all of the experiments we trained using the empirical mean squared error as a loss function and optimized over full batches of the training sets. In order to facilitate more efficient

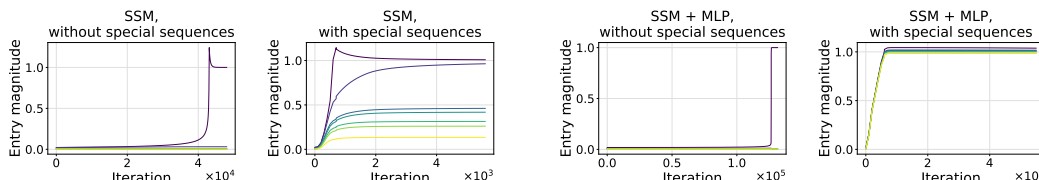

Figure 7: Demonstration of the dynamical characterization derived in Proposition 1. This figure is identical to Figure 2 except that a different random seed was used.

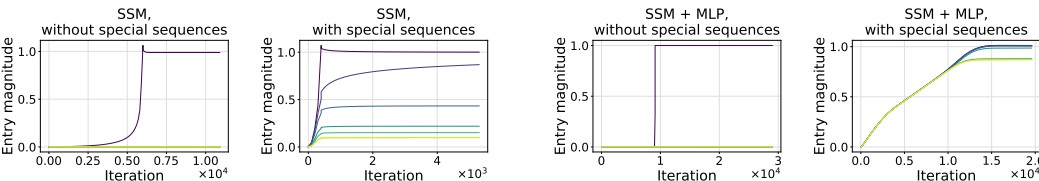

Figure 8: Demonstration of the dynamical characterization derived in Proposition 1. This figure is identical to Figure 3 except that a different random seed was used.

experimentation in the experiments where a standalone SSM is trained, we optimized using gradient descent with an adaptive learning rate scheme, where at each iteration a base learning rate is divided by the square root of an exponential moving average of squared gradient norms (see appendix D.2 in Razin et al. (2022) for more details). We used a weighted average coefficient of $\beta = 0.8$ and a softening constant of $10^{-6}$. Note that only the learning rate (step size) is affected by this scheme, not the direction of movement. Comparisons between the adaptive scheme and optimization with a fixed learning rate showed no significant difference in terms of the dynamics, while run times of the former were considerably shorter. Table 7 specifies the base learning rate used in each experiment.

### G.1.2 SSM IN A NON-LINEAR NEURAL NETWORK

**Models.** In the experiments reported in Figures 2, 3, 7, and 8 where an SSM was trained as a part of a non-linear neural network, we used neural networks comprised of an SSM module whose output was fed to a 2 hidden layers MLP module. Overall, the models used realize the following function:

$$D_{out} \cdot \sigma\left( D_{hidden} \cdot \sigma(D_{in} \cdot \phi_{A,B,C}(\mathbf{x})) \right)$$

where $d_h \in \mathbb{N}$ is the hidden MLP width, $D_{in} \in \mathbb{R}^{d_h \times 1}$, $D_{hidden} \in \mathbb{R}^{d_h \times d_h}$ and $D_{out} \in \mathbb{R}^{1 \times d_h}$ are the MLP's parameter matrices, $\sigma(x) := \max\{0, x\}$ is the MLP's activation function and $\phi_{A,B,C}(\mathbf{x})$ is the output of an SSM with the parameter matrices $A, B, C$. All teacher models used had SSM modules of dimension $d^* = 1$ that were set with the parameters

$$A^* = 1 \ , \ \ B^* = 1 \ , \ \ C^* = 1$$

The teacher models in Figures 2 and 7 had hidden MLP width of $d_h^* = 15$ while the teacher models in Figures 3 and 8 had hidden MLP width of $d_h^* = 25$. In both cases, the teacher models had MLP modules that were set with the following parameter matrices:

$$D_{in}^* = \mathbf{1}_{d_h^*} \ , \ \ D_{hidden}^* = I_{d_h^*} \ , \ \ D_{out}^* = \frac{1}{2} \cdot \mathbf{1}_{d_h^*}^\top$$

We trained all of the SSM and MLP parameter matrices of our student models. The student models in Figures 2 and 7 had SSM dimension of $d = 10$ and hidden MLP width of $d_h = 15$, while the student models in Figures 3 and 8 had SSM dimension of $d = 20$ and hidden MLP width of $d_h = 25$.

**Data.** Data for the experiments were generated in the same manner as in Section G.1.1. Table 4 specifies which non-zero indices were present in each sequence type for each experiment. Training sequences of both types had their non-zero entries sampled i.i.d from $\mathcal{N}(0, 1)$. Table 5 specifies how many training sequences of each type were used in each experiment.

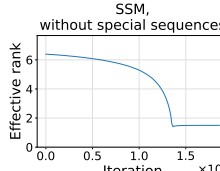 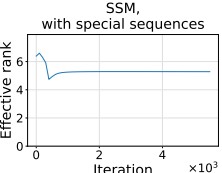 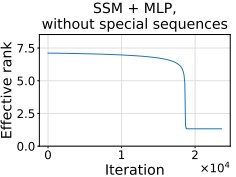 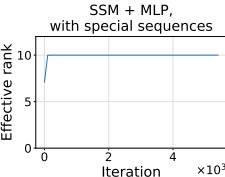

Figure 9: Demonstration of the dynamical characterization derived in Proposition 1 through the lens of the effective rank of $A$—introduction of special sequences to the training set results in significantly larger effective rank, in compliance with the disruption of greedy low rank learning. Each plot shows the effective rank of $A$ during the training process reported in Figure 2.

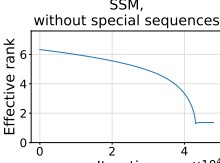 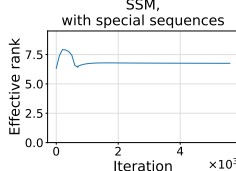 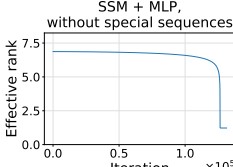 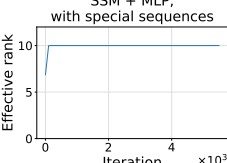

Figure 10: Demonstration of the dynamical characterization derived in Proposition 1 through the lens of the effective rank of $A$. This figure is identical to Figure 9 except that the setting considered is that of Figure 7.

**Initialization.** In all experiments we initialized the student's $A$, $B$ and $C$ parameter matrices identically to the standalone SSM experiments (Section G.1.1). Table 6 report the values of `sd_A`, `sd_B_C` and `diff` used in each experiment.

To initialize the MLP layers, we initialized all parameter matrices by sampling i.i.d values from $\mathcal{N}(0, \texttt{sd\_D})$. We used $\texttt{sd\_D} = 0.03$ in the original experiments (Figures 2 and 7) and $\texttt{sd\_D} = 0.1$ in the experiments with longer sequences (Figures 3 and 8).

**Optimization.** To speed up the optimization we trained using the default Keras implementation of Adam (Kingma, 2014) with its default hyperparameters. Table 7 report the base learning rates used in each of the experiments.

## G.2 CLEAN-LABEL POISONING

In this Appendix we provide implementation details for the experiments provided in Sections 4.2 and F.2. All synthetic experiments were implemented using Keras (Chollet et al., 2015) and were carried out using a single Nvidia RTX 2080 Ti GPU. The real-world experiments reported in Table 2 were implemented using PyTorch (Paszke et al., 2019) and were carried out using a cluster of 8 Nvidia RTX 2080 Ti GPUs.

### G.2.1 SSM PER THEOREM 1

The main goal of the experiments in the first poisoning setting (standalone SSM per Theorem 1) was to approximate the solution to the system of ODEs induced by gradient flow (Equation (4)) in order to demonstrate the results of Theorem 1. To do so we employed the use of the `odeint` function of SciPy (Virtanen et al., 2020) which is a numerical solver for systems of ODEs based on `lsoda` from the FORTRAN library odepack (Hindmarsh, 1983). `odeint`'s arguments are the initial point in parameter space $A(0)$, the timestamps at which the solution is required, and a function which specifies the system by intaking a timestamp $t$ and a point in parameter space $A$ and outputting the derivative in time $t$ at $A$. `odeint` outputs a set of numerical approximations for the solution of the system at the required timestamps.

**Models.** We use teacher and student models according to the setting described in Section 3.2. We used a teacher SSM of dimension $d^* = 2$ that was set with the parameters

$$A^* = \begin{pmatrix} 1 & 0 \\ 0 & 0 \end{pmatrix} \ , \ B^* = \begin{pmatrix} 1 & \sqrt{d-1} \end{pmatrix}^\top \ , \ C^* = \begin{pmatrix} 1 & \sqrt{d-1} \end{pmatrix}$$

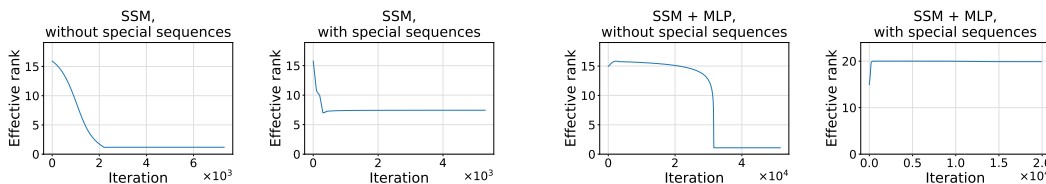

Figure 11: Demonstration of the dynamical characterization derived in Proposition 1 through the lens of the effective rank of $A$. This figure is identical to Figure 9 except that the setting considered is that of Figure 3.

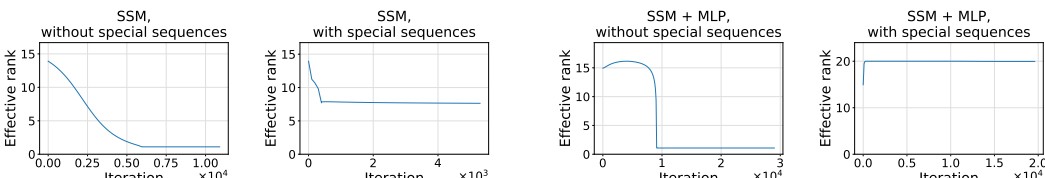

Figure 12: Demonstration of the dynamical characterization derived in Proposition 1 through the lens of the effective rank of $A$. This figure is identical to Figure 9 except that the setting considered is that of Figure 8.

We used student SSM models whose input and output matrices $B(\cdot)$ and $C(\cdot)$ were fixed at $\mathbf{1}_d$ and $\mathbf{1}_d^\top$ respectively. The student models had dimension $d = 10$ in the original poisoning experiments (Table 1), and dimension $d = 20$ when training over longer sequences (Table 3).

**Data.** In all experiments we used the respective teachers to generate the labels for all training sequences. We trained using 5 "baseline" sequences of the form $x \cdot \mathbf{e_1} \in \mathbb{R}^k$ and a single "special" sequence of the form $x \cdot \mathbf{e_{k-1}} \in \mathbb{R}^k$. Given some positive scalar $P$ we determined the "baseline" sequences by first sampling 5 i.i.d values from $\mathcal{N}(0, 1)$, and then scaling them such that their sum of squares is equal $P$. The single "special sequence" was set with $x = \sqrt{P}$. This is done to satisfy a technical requirement of Theorem 1 (See Section D for details on this requirement). The reported results use $P = 10$, and we saw similar results qualitatively when using other positive values for $P$ and other amounts of "baseline" and "special" sequences. The relevant experiments reported in Tables 1 and 3 were trained using sequences of lengths 7 and 9 respectively.

**Initialization.** We initialized the student's diagonal matrix $A$ in a manner that was inspired by the initialization set $\mathcal{I}$ of Theorem 1. In each experiment we first sampled $d$ i.i.d entries from $\mathcal{N}(0, \texttt{sd\_A})$ and then arranged them in descending order. We then set the second entry to have the first's value minus a constant $\texttt{diff}$. This was done to reflect the near zero initialization on the one hand and the proximity to the reference trajectory on the other hand. Table 8 reports the values of $\texttt{sd\_A}$ and $\texttt{diff}$ used in each experiment.

**Optimization.** We input $\texttt{odeint}$ a custom function which computes $-\nabla\ell(A)$ (Equation (4)) given the point $A$ to be used for derivative computations. Table 9 reports the timestamps simulated in each experiment. All experiments reached training loss values of less than $10^{-5}$ and stable generalization errors.

**Generalization evaluation.** In the first poisoning setting (standalone SSM per Theorem 1) generalization errors were measured via impulse responses of length 40 as defined in Definition 1, divided by the $\ell_\infty$ norm of the teacher's length 40 impulse response (this normalization is chosen so that the zero mapping has generalization error 1).

### G.2.2 SSM Beyond Theorem 1

**Models.** We used the same teacher models as described in Section G.1.1. We used student SSM models that were trained end to end (*i.e.* $B(\cdot)$ and $C(\cdot)$ were not fixed). The student models had dimension $d = 10$ in the original poisoning experiments (Table 1), and dimension $d = 20$ in the experiments with longer sequences (Table 3).

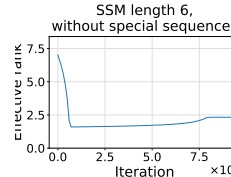 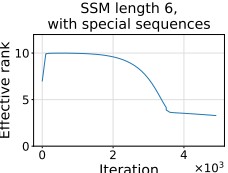 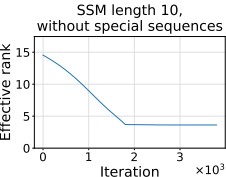 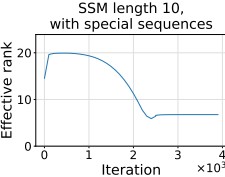

Figure 13: Demonstration of the dynamical characterization derived in Proposition 1 through the lens of the effective rank of $A$. This figure is identical to Figure 9 except that the setting considered is that of Figure 4.

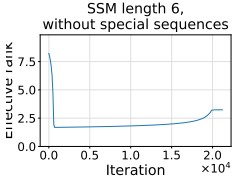 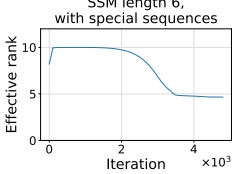 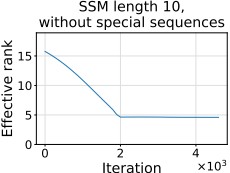 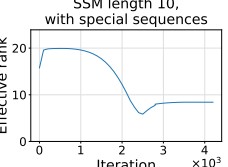

Figure 14: Demonstration of the dynamical characterization derived in Proposition 1 through the lens of the effective rank of $A$. This figure is identical to Figure 9 except that the setting considered is that of Figure 5.

**Data.** Data for the experiments were generated in the same manner as in Section G.1.1. Table 10 specifies which non-zero indices were present in each sequence type for each experiment. Training sequences of both types had their non-zero entries sampled i.i.d from $\mathcal{N}(0, 1)$. Table 11 specifies how many training sequences of each type were used in each experiment.

**Initialization.** To speed up optimization we modified the initialized employed in Section G.1.1 by adding a small constant of $10^{-1}$ to all entries of $A$ and $B$ at initialization. This modification showed no significant differences in terms of the generalization error achieved by the models when compared to without it, while run times of the former were considerably shorter. Table 12 report the values of `sd_A`, `sd_B_C` and `diff` used in each experiment.

**Optimization.** We followed a training scheme identical to Section G.1.1. Table 13 report the base learning rates used in each of the experiments.

We optimized all models to reach a training loss under $0.01$. To verify the generalization errors we report were stable, we trained for additional iterations after reaching sub $0.01$ training loss. We trained the standalone SSM models for $1500$ more iterations, and the models with additional layers for $5000$ more iterations.

**Generalization evaluation.** Generalization errors were measured via impulse responses of length $40$ as defined in Definition 1, divided by the $\ell_\infty$ norm of the teacher's length $40$ impulse response (such that the zero mapping has error of one). The same evaluation procedures were used in both the original experiments (Table 1) and in the longer experiments (Table 3).

### G.2.3 SSM IN NON-LINEAR NEURAL NETWORK

**Models.** We used the same teacher models as described in Section G.1.2. We used student SSM models that were trained end to end (*i.e.* $B(\cdot)$ and $C(\cdot)$ were not fixed). The student models had dimension $d = 10$ in the original poisoning experiments (Table 1), and dimension $d = 20$ in the experiments with longer sequences (Table 3).

**Data.** The data used is identical to that of Section G.2.2. Table 11 specify how many training sequences of each type were used in each experiment.

**Initialization.** We initialized the student models identically to Section G.1.2. To speed up optimization we modified the initialized employed in Section G.1.2 by adding a small constant of $10^{-3}$ to all entries of $A$ and $B$ at initialization. This modification showed no significant differences in terms of the generalization error achieved by the models when compared to without it, while run times of the former were considerably shorter. Table 12 report the values of `sd_A`, `sd_B_C` and `diff` used in each experiment.

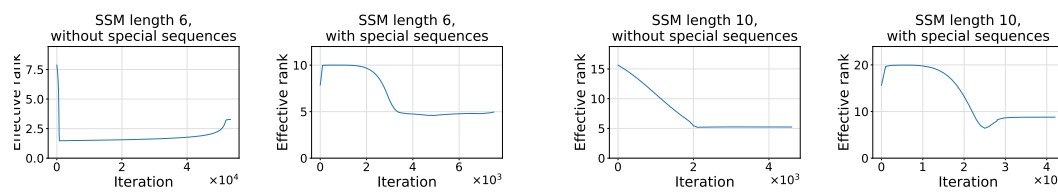

Figure 15: Demonstration of the dynamical characterization derived in Proposition 1 through the lens of the effective rank of $A$. This figure is identical to Figure 9 except that the setting considered is that of Figure 6.

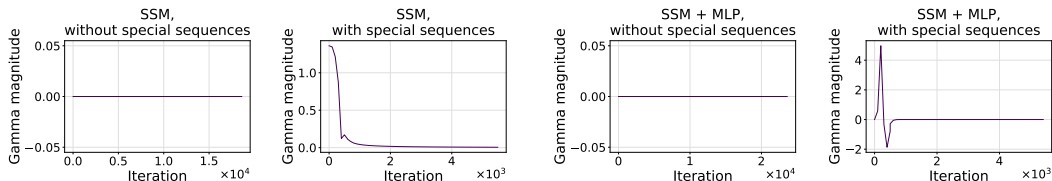

Figure 16: Evolution of $\gamma^{(0)}(t)$ (as defined in Proposition 1) during training. Each plot shows the values of $\gamma^{(0)}(t)$ during optimization in the setting of Figure 2. As can be seen, in compliance with our interpretation of Proposition 1, larger absolute values of $\gamma^{(0)}(t)$ do not correspond to greedy low rank learning, whereas lower absolute values do.

**Optimization.** We followed a training scheme identical to Section G.1.2. Table 13 report the base learning rates used in each of the experiments.

We optimized all models to reach a training loss under $0.01$. To verify the generalization errors we report were stable, we trained for additional iterations after reaching sub $0.01$ training loss. We trained the standalone SSM models for $1500$ more iterations, and the models with additional layers for $5000$ more iterations.

**Generalization evaluation.** Generalization errors were measured via the root mean square error of a held-out test set of $2000$ correctly labeled sequences of length $40$, divided by the $\ell_2$ norm of the teacher's outputs vector (such that the zero mapping has error of one). The same evaluation procedures were used in both the original experiments (Table 1) and in the longer experiments (Table 3).

### G.2.4 SSM IN REAL-WORLD SETTING

The SSM-based S4 neural network adheres to the implementation provided in `https://github.com/state-spaces/s4`, utilizing the "minimalist S4" configuration available in `s4d.py` and `s4.py`. In all experiments, the S4 neural network had four layers with a hidden dimension of $256$. For the poisoning process, we adapted the Gradient Matching method provided in `https://github.com/JonasGeiping/poisoning-gradient-matching`. The adaptation involved: reshaping the MNIST image input into a sequence format compatible with the S4 architecture; and introducing regularization that encourages the last elements of an injected noise sequence to be relatively large.[14] Following this adaptation, apart from varying the number of target test instances and the percentage of poisonous examples, all hyperparameters were kept at their default values.

---

[14]Regularization comprised weight decay of $28 - i$ applied to the noise entries corresponding to the $i$th row of an input image, where $i \in [28]$ (recall that MNIST images are of size $28 \times 28$).

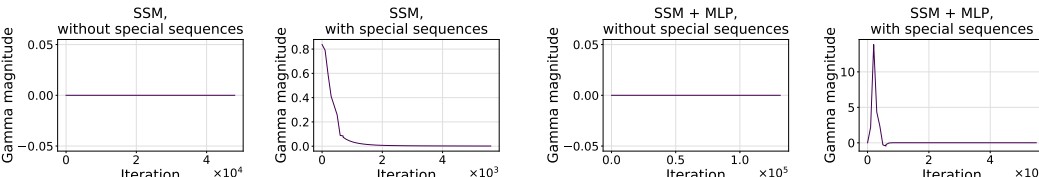

Figure 17: Evolution of $\gamma^{(0)}(t)$ (as defined in Proposition 1) during training. This figure is identical to Figure 16 except that the setting considered is that of Figure 7.

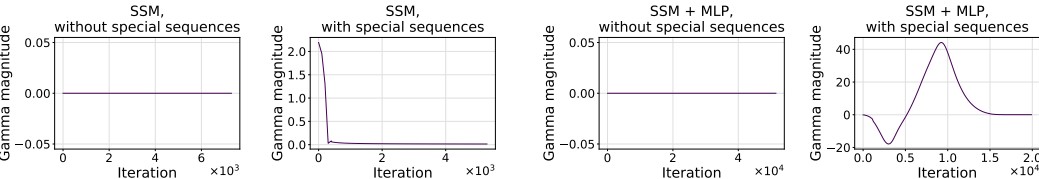

Figure 18: Evolution of $\gamma^{(0)}(t)$ (as defined in Proposition 1) during training. This figure is identical to Figure 16 except that the setting considered is that of Figure 3.

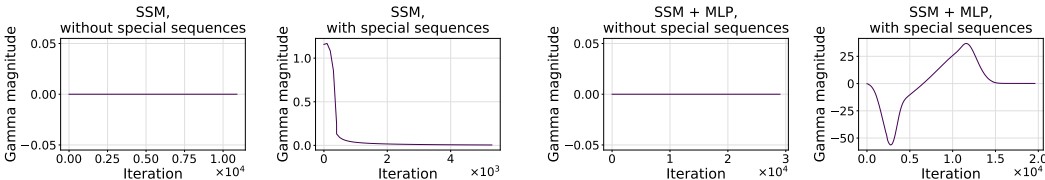

Figure 19: Evolution of $\gamma^{(0)}(t)$ (as defined in Proposition 1) during training. This figure is identical to Figure 16 except that the setting considered is that of Figure 8.

Table 3: Demonstration of clean-label poisoning of SSMs. The table is identical to Table 1, except that the sequences used to train the models were longer, namely, of sequence length 10 as opposed to 6. Notice that across all settings, special training sequences significantly deteriorate generalization. For further details see Table 1 as well as Section G.2.

| Setting | Without special sequences | With special sequences |
|---|---|---|
| SSM per Theorem 1 | $7.34 \times 10^{-3}$ | $3.51 \times 10^{-2}$ |
| SSM beyond Theorem 1 | $1.22 \times 10^{-1}$ | $1.2$ |
| SSM in non-linear neural network | $4.67 \times 10^{-2}$ | $8.93 \times 10^{-2}$ |

Table 4: Types of sequences used in dynamics experiments (Figures 2 to 8). Last two columns (right) indicate the non-zero indices for each sequence type.

| Setting | Length | Baseline indices | Special indices |
|---|---|---|---|
| SSM / SSM + MLP (Figures 2 and 7) | 6 | $1, 2$ | $5, 6$ |
| SSM / SSM + MLP longer (Figures 3 and 8) | 10 | $1, 2, \ldots, 7$ | $9, 10$ |
| SSM higher rank (Figures 4 to 6) | 6 | $1, 2$ | $5$ |
| SSM higher rank longer (Figures 4 to 6) | 10 | $1, 2, \ldots, 7$ | $9$ |

Table 5: Amount of sequences of each type used in dynamics experiments (Figures 2 to 8).

| Setting | Baseline amount | Special amount |
|---|---|---|
| SSM (Figures 2, 3, 7, and 8) | 8 | 10 |
| SSM + MLP (Figures 2, 3, 7, and 8) | 20 | 20 |
| SSM higher rank (Figures 4 to 6) | 8 | 10 |

Table 6: Values of `sd_A`, `sd_B_C` and `diff` used in the dynamics experiments (Figures 2 to 8).

| Setting | sd_A | sd_B_C | diff |
|---|---|---|---|
| SSM (Figures 2 and 7) | $10^{-2}$ | $10^{-2}$ | $0.05 \times \exp\big(20 \cdot \log_{10}(\texttt{sd\_A})\big)$ |
| SSM longer (Figures 3 and 8) | $10^{-3}$ | $5 \times 10^{-2}$ | $0.05 \times \exp\big(20 \cdot \log_{10}(\texttt{sd\_A})\big)$ |
| SSM + MLP (Figure 7) | $10^{-2}$ | $10^{-2}$ | $0.05 \times \exp\big(0.5 \cdot \log_{10}(\texttt{sd\_A})\big)$ |
| SSM + MLP longer (Figure 8) | $10^{-3}$ | $10^{-3}$ | $0.05 \times \exp\big(2 \cdot \log_{10}(\texttt{sd\_A})\big)$ |
| SSM higher rank (Figures 4 to 6) | $10^{-2}$ | — | $0.05 \times \exp\big(2 \cdot \log_{10}(\texttt{sd\_A})\big)$ |
| SSM higher rank longer (Figures 4 to 6) | $10^{-2}$ | — | $0.05 \times \exp\big(3 \cdot \log_{10}(\texttt{sd\_A})\big)$ |

Table 7: Base learning rates used in dynamics experiments (Figures 2 to 8). Last two columns (right) indicate the base learning rate used in the experiments without the addition of "special" sequences and with their addition respectively.

| Setting | W/o special sequences | W/ special sequences |
|---|---|---|
| SSM (Figures 2 and 7) | 0.01 | 0.01 |
| SSM longer (Figures 3 and 8) | 0.01 | 0.01 |
| SSM + MLP (Figures 2 and 7) | 0.01 | 0.001 |
| SSM + MLP longer (Figures 3 and 8) | 0.01 | $5 \times 10^{-5}$ |
| SSM higher rank (Figures 4 to 6) | 0.01 | 0.001 |
| SSM higher rank longer (Figures 4 to 6) | 0.001 | 0.001 |

Table 8: Values of `sd_A` and `diff` used in the experiments of the first poisoning setting (Tables 1 and 3).

| Setting | sd_A | diff |
|---|---|---|
| SSM per Theorem 1 (Table 1) | $10^{-3}$ | $0.05 \times \exp\big(5 \cdot \log_{10}(\texttt{sd\_A})\big)$ |
| SSM per Theorem 1 longer (Table 3) | $5 \times 10^{-3}$ | $0.05 \times \exp\big(10 \cdot \log_{10}(\texttt{sd\_A})\big)$ |

Table 9: Timestamps simulated for the experiments of the first poisoning setting (Tables 1 and 3). The timestamps used for each experiment are the endpoints of intervals obtained by evenly partitioning the range $(0, \texttt{last\_timestamp})$ into `timestamp_amount` segments.

| Setting | last_timestamp | timestamp_amount |
|---|---|---|
| SSM per Theorem 1 (Table 1) w/o special | $10^{11}$ | 1000 |
| SSM per Theorem 1 (Table 1) w/ special | $10^{4}$ | 1000 |
| SSM per Theorem 1 longer (Table 3) w/o special | $10^{12}$ | 10000 |
| SSM per Theorem 1 longer (Table 3) w/ special | $10^{6}$ | 1000 |

Table 10: Types of sequences used in the experiments of the second and third poisoning settings (Tables 1 and 3). Last two columns (right) indicate the non-zero indices for each sequence type.

| Setting | Length | Baseline indices | Special indices |
|---|---|---|---|
| $2^{nd}$ and $3^{rd}$ settings of Table 1 | 6 | $1, 2$ | 5 |
| $2^{nd}$ and $3^{rd}$ settings of Table 3 | 10 | $1, 2, \ldots, 7$ | 9 |

Table 11: Amount of sequences of each type used in in the experiments of the second and third poisoning settings (Tables 1 and 3).

| Setting | Baseline amount | Special amount |
|---|---|---|
| SSM beyond Theorem 1 (Tables 1 and 3) | 8 | 10 |
| SSM in non-linear NN (Tables 1 and 3) | 20 | 20 |

Table 12: Values of `sd_A`, `sd_B_C` and `diff` used in the experiments of the second and third poisoning settings (Tables 1 and 3).

| Setting | sd_A | sd_B_C | diff |
|---|---|---|---|
| SSM beyond Theorem 1 (Table 1) | $10^{-3}$ | $10^{-3}$ | $0.05 \times \exp\big(5 \cdot \log_{10}(\texttt{sd\_A})\big)$ |
| SSM beyond Theorem 1 longer (Table 3) | $10^{-2}$ | $10^{-3}$ | $0.05 \times \exp\big(3 \cdot \log_{10}(\texttt{sd\_A})\big)$ |
| SSM in non-linear NN (Table 1) | $10^{-2}$ | $10^{-2}$ | $0.05 \times \exp\big(0.5 \cdot \log_{10}(\texttt{sd\_A})\big)$ |
| SSM in non-linear NN longer (Table 3) | $10^{-3}$ | $10^{-3}$ | $0.05 \times \exp\big(2 \cdot \log_{10}(\texttt{sd\_A})\big)$ |

Table 13: Base learning rates used in the experiments of the second and third poisoning settings (Tables 1 and 3). Last two columns (right) indicate the base learning rate used in the experiments without the addition of "special" sequences and with their addition respectively.

| Setting | W/o special sequences | W/ special sequences |
|---|---|---|
| SSM beyond Theorem 1 (Table 1) | 0.01 | 0.01 |
| SSM beyond Theorem 1 longer (Table 3) | 0.001 | 0.001 |
| SSM in non-linear NN (Table 1) | 0.01 | 0.01 |
| SSM in non-linear NN longer (Table 3) | 0.01 | $5 \times 10^{-5}$ |

