# OpenReview forum: "The Implicit Bias of Structured State Space Models Can Be Poisoned With Clean Labels"
_ICLR.cc/2025/Conference — Submitted to ICLR 2025_

### Official Review · Reviewer_mXtt · 2024-11-01

**Soundness:** 2
**Presentation:** 2
**Contribution:** 1
**Rating:** 3
**Confidence:** 5

**Summary:**

This paper shows that when learning certain linear state-space models, gradient flow converges to solutions that generalize well with certain training trajectories but to those that do not when some specific trajectory is appended.

**Strengths:**

Data poisoning attacks are one of the practical concerns in training machine learning models, this paper studies some theoretical aspects of this issue for state-space models.

**Weaknesses:**

The result shown in this paper comes from an extremely contrived example and the reviewer has strong concerns regarding its practical significance.

1. Theorem 1 states that the learned student SSM generalizes "under various choices of training sequence" (line 258), the truth is these training sequences are $\left\lbrace p_i\mathbf{e_1},p_i\right\rbrace_{i=1}^n$, where $p_i\in\mathbb{R}$ (line 857, Definition 2 in Appendix), i.e., the input sequences to the teacher SSM are the same up to some scaling. Since the SSM to be learned is linear, these $n$ training sequences are effectively single sequences. It is not a big surprise if student SSM generalizes when this single training trajectory is crafted to guide the GF to the generalizing solutions, and then it is even less surprising if another crafted training trajectory is added to steer GF to other solutions. There are effectively only two training trajectories, after all.

2. Another reason this example is contrived is that there is no persistence of excitation (Green&Moore, 1986) in the input sequence used for training, which is fundamentally needed to identify linear systems.

3. The state matrix $A$ is assumed diagonal, the $B,C$ matrices are fixed during training. However, these are of less concern than the previous points.

4. The reviewer does not appreciate the presentation in this paper where assumptions ($A$ being diagonal) are not stated as proper assumptions (line 112), and key definitions (precise definitions of the training trajectories) are written in the Appendix.

**References**:

M Green and J Moore, Persistence of excitation in linear systems, Systems & Control Letters, 1986

**Questions:**

See "Weakness".

---

> ### Author Response · Authors · 2024-11-16
> **Author Response (Part 1/2)**
>
> Thank you for your review. Below we address each of your points. In light of our response, we would greatly appreciate it if you would consider raising your score.
>
> >The result shown in this paper comes from an extremely contrived example and the reviewer has strong concerns regarding its practical significance.
>
> Please note that (as stated in the introduction, line 67) our theory comprises two main results:
> 1. **Dynamical characterization (Section 3.1):** This result applies to a very broad range of settings (including ones where the SSM is trained as part of a non-linear neural network).  Dynamical characterizations have proven to be central to the theory of various deep learning models (see, e.g, [1,2,3]), and we believe our dynamical characterization may similarly become central to the theory of SSMs.
> 2. **Demonstration of clean-label poisoning (Section 3.2):** This theoretical demonstration indeed applies to a specific setting.  Its purpose is to prove existence of clean-label poisoning in SSMs, similarly to numerous existence proofs in deep learning theory (see, e.g, [4,5,6]).  Moreover, at least part of the analyzed setting—namely, the cleanly labeled training example leading generalization to fail—is expected to be specific, as in practice, coming up with such examples requires substantial computational efforts (see, e.g., [7,8,9,10]).  Following your feedback, we will generalize other aspects of the analyzed setting, in particular the training examples with which generalization succeeds—see details below.
>
> >Theorem 1 states that the learned student SSM generalizes "under various choices of training sequence" (line 258), the truth is these training sequences are $\lbrace p_i\mathbf{e_1},p_i\rbrace_{i=1}^n$, where $p_i∈\mathbb{R}$ (line 857, Definition 2 in Appendix), i.e., the input sequences to the teacher SSM are the same up to some scaling. Since the SSM to be learned is linear, these $n$ training sequences are effectively single sequences.
>
> Note that Theorem 1 applies to a generic (odd) $\kappa$, thus it actually implies generalization under any training sequence $\mathbf{e}_j$ where $j$ is odd and no greater than $\kappa - 6$.  Moreover, Theorem 1 can easily be extended to establish generalization under any set of training sequences with positive entries that hold zeros in their last two entries.  We are now adding this extension to the paper.
>
> >It is not a big surprise if student SSM generalizes when this single training trajectory is crafted to guide the GF to the generalizing solutions, and then it is even less surprising if another crafted training trajectory is added to steer GF to other solutions. There are effectively only two training trajectories, after all.
>
> We respectfully disagree, and believe there may be a misunderstanding.
>
> First of all, showing that a training sequence such as $\mathbf{e}_1$ leads to generalization is non-trivial, and requires our dynamical characterization (Section 3.1).  Secondly, we cannot “add a trajectory” to GF, we can only add a training sequence, and that leads to the GF trajectory changing in a highly non-trivial manner.  Indeed, in Theorem 1, adding the training sequence $\mathbf{x}^\dagger$ changes the GF trajectory in a way that is highly complex (e.g., as discussed in the proof sketch of Theorem 1, it leads the trajectory to encounter saddle points)—far more complex than “adding” a trajectory associated with $\mathbf{x}^\dagger$. Finally, there are not only two training trajectories: any scaling of a training sequence, or a slight shift in the initialization (our proof accounts for infinitely many initializations), lead to highly non-trivial changes in the GF trajectory.
>
> We fear you may be underestimating the complexity of the correspondence between training sequences and the GF trajectories they lead to.  As the proof sketch of Theorem 1 emphasizes, this correspondence is extremely non-trivial.
>
> >Another reason this example is contrived is that there is no persistence of excitation (Green&Moore, 1986) in the input sequence used for training, which is fundamentally needed to identify linear systems.
>
> Here too, we believe there may be a misunderstanding.
>
> When studying implicit bias in a teacher-student setting, the whole point is that the training data will not support identifying the teacher. Rather, there should be multiple solutions that fit the training data, some generalize (recover the teacher), while others do not. The question of interest is then whether there is some implicit bias leading to generalization. Please let us know if this point is not clear and we will happily elaborate.

---

> > ### Author Response · Authors · 2024-11-16
> > **Author Response (Part 2/2)**
> >
> > >The state matrix $A$ is assumed diagonal, the $B,C$ matrices are fixed during training. However, these are of less concern than the previous points.
> >
> > As explicitly stated in lines 109-112, the most common form of structure imposed on SSMs is diagonality of $A$, and we accordingly study this structure.
> >
> > Theorem 1 indeed assumes that $B$ and $C$ are fixed throughout training. This is stated explicitly right before Theorem 1 (line 254), and then again in our account for the limitations of our work (line 440).  Note that our dynamical characterization (Section 3.1) is not limited by this assumption.
> >
> > >The reviewer does not appreciate the presentation in this paper where assumptions ($A$ being diagonal) are not stated as proper assumptions (line 112), and key definitions (precise definitions of the training trajectories) are written in the Appendix.
> >
> > As far as we know, our theoretical presentation is fully rigorous. We chose to embed assumptions in the body of the text (rather than placing them in separate environments) solely for the sake of readability, which was commended by other reviewers. Following your feedback, in order to support easy tracking of assumptions, while still maintaining readability, we are adding to the paper an appendix listing our assumptions.
> >
> > As for the definition of training sequences used for proving Theorem 1, following your feedback, we will add them to the proof sketch that appears in the body of the paper.
> >
> > **References**
> >
> > [1] Saxe, Andrew M., James L. McClelland, and Surya Ganguli. "Exact solutions to the nonlinear dynamics of learning in deep linear neural networks." arXiv preprint arXiv:1312.6120 (2013).
> >
> > [2] Ji, Ziwei and Matus Telgarsky. “The implicit bias of gradient descent on nonseparable data.” Annual Conference Computational Learning Theory (2019).
> >
> > [3] Razin, Noam, Asaf Maman, and Nadav Cohen. "Implicit regularization in hierarchical tensor factorization and deep convolutional neural networks." International Conference on Machine Learning. PMLR, 2022.
> >
> > [4] Telgarsky, Matus. "Benefits of depth in neural networks." Conference on learning theory. PMLR, 2016.
> >
> > [5] Vardi, Gal, and Ohad Shamir. "Implicit regularization in relu networks with the square loss." Conference on Learning Theory. PMLR, 2021.
> >
> > [6] Arjevani, Yossi, et al. "Lower bounds for non-convex stochastic optimization." Mathematical Programming 199.1 (2023): 165-214.
> >
> > [7] Huang, W. Ronny, et al. "Metapoison: Practical general-purpose clean-label data poisoning." Advances in Neural Information Processing Systems 33 (2020): 12080-12091.
> >
> > [8] Geiping, Jonas, et al. "Witches' brew: Industrial scale data poisoning via gradient matching." arXiv preprint arXiv:2009.02276 (2020).
> >
> > [9] Shafahi, Ali, et al. "Poison frogs! targeted clean-label poisoning attacks on neural networks." Advances in neural information processing systems 31 (2018).
> >
> > [10] Aghakhani, Hojjat, et al. "Bullseye polytope: A scalable clean-label poisoning attack with improved transferability." 2021 IEEE European symposium on security and privacy (EuroS&P). IEEE, 2021.

---

> > > ### Comment · Reviewer_mXtt · 2024-11-21
> > >
> > > Thanks for the response, and thanks for the lesson(?) on why studying GF is hard, even though the reviewer's concern does not include "the analysis might be too simple."
> > >
> > > The reviewer's concern is not addressed.
> > >
> > > 1. Contrived example
> > >
> > >  >Quote from the reviewer: It is not a big surprise if student SSM generalizes when this single training trajectory is crafted to guide the GF to the generalizing solutions, and then it is even less surprising if another crafted training trajectory is added to steer GF to other solutions. There are effectively only two training trajectories, after all.
> > >
> > > The reviewer apologizes for the inconsistency in writing "training trajectory" and "training sequences" in different parts of the review, but they both mean $\left\lbrace p_i\textbf{e}_j,p_i\right\rbrace$ for some $p_i$. Therefore, this initial reviewer says that this paper studies a learning problem trained with a single data point, showing generalization, and then shows non-generalization when adding one more data point. Based on this, the reviewer thinks the example is contrived. The promised revision (using a different $j$) does not resolve the issue; it just uses a different data point.
> > >
> > > The motivation for understanding the clean-label poisoning attack from a theoretical perspective is good, but the studied example is so different from a practical learning setting. Justification of its practical relevance is needed.
> > >
> > > 2. Writtings
> > >
> > > >Quote from authors: We chose to embed assumptions in the body of the text (rather than placing them in separate environments) solely for the sake of readability, which was commended by other reviewers.
> > >
> > > I don't think saying this paper "is well written" is commending the "embeded assumptions in the body of the text." In fact, I think not stating Assumptions properly and hiding key definitions in the appendix are hindering the readability because all that information is needed for any reader to fully appreciate and assess the results.

---

> > > > ### Author Response · Authors · 2024-11-23
> > > > **Author Response to Reviewer Reply (Part 1/2)**
> > > >
> > > > Thank you for responding. Below we address the two remaining concerns that you have listed. In light of our response, and given how low your scores currently are, we kindly ask that you consider raising them.
> > > > >The reviewer apologizes for the inconsistency in writing "training trajectory" and "training sequences" in different parts of the review, but they both mean $\lbrace p_i\mathbf{e}_j,p_i \rbrace$ for some $p_i$. Therefore, this initial reviewer says that this paper studies a learning problem trained with a single data point, showing generalization, and then shows non-generalization when adding one more data point. Based on this, the reviewer thinks the example is contrived. The promised revision (using a different j) does not resolve the issue; it just uses a different data point.
> > > >
> > > > >The motivation for understanding the clean-label poisoning attack from a theoretical perspective is good, but the studied example is so different from a practical learning setting. Justification of its practical relevance is needed.
> > > >
> > > > Thank you for supporting the motivation behind our work.
> > > >
> > > > We reiterate that our first main theoretical result (dynamical characterization in Section 3.1), as well as our experiments, apply to a broad range of settings, including ones where an SSM is trained as part of a non-linear neural network.
> > > >
> > > > With regards to the theoretical result you are referring to (Theorem 1):
> > > > 1. Please note that a cleanly labeled training example leading generalization to fail is expected to be specific, as in practice, coming up with such examples requires substantial computational efforts.
> > > >
> > > > 2. As for training examples leading generalization to succeed, there can be multiple of those, provided their last two entries are zero, and their remaining entries are positive.  We are adding this clarification to the paper.
> > > >
> > > > 3. To our knowledge, Theorem 1 is the first result to formally establish clean-label poisoning with a deep learning model. In the field of deep learning theory, it is very common to establish new results on simplified settings with specific examples (see, e.g., [1, 2, 3, 4]).  If such results were disqualified from publication at ICLR, then in our humble opinion the conference would not fulfill its purpose.

---

> > > > > ### Author Response · Authors · 2024-11-23
> > > > > **Author Response to Reviewer Reply (Part 2/2)**
> > > > >
> > > > > >I don't think saying this paper "is well written" is commending the "embeded assumptions in the body of the text." In fact, I think not stating Assumptions properly and hiding key definitions in the appendix are hindering the readability because all that information is needed for any reader to fully appreciate and assess the results.
> > > > >
> > > > > We reiterate that, to our knowledge, our theoretical presentation is fully rigorous, and in particular, includes all assumptions in the body of the paper.  In order to accommodate your request for assumptions to appear in dedicated environments, we are adding to the paper an appendix that does so.  We are also specifying in the proof sketch of Theorem 1 the specific training examples we use to prove the theorem (i.e., to prove existence of clean-label poisoning).
> > > > >
> > > > > With regards to our mathematical writing style, please note that various guides for mathematical writing (e.g., [5, 6]) strongly emphasize the importance of a streamlined text without verbose technical statements.  Our writing style of streamlining text by embedding theoretical assumptions in its body (rather than placing them in separate environments) is adopted by important publications in deep learning theory (e.g., [7, 8, 9, 10]).  In our humble opinion, ICLR readership should have the opportunity to accommodate such papers.
> > > > >
> > > > > **References**
> > > > >
> > > > > [1] Frei, Spencer, et al. "Implicit bias in leaky relu networks trained on high-dimensional data." arXiv preprint arXiv:2210.07082 (2022).
> > > > >
> > > > > [2] Arora, Sanjeev, Nadav Cohen, and Elad Hazan. "On the optimization of deep networks: Implicit acceleration by overparameterization." International conference on machine learning. PMLR, 2018.
> > > > >
> > > > > [3] Li, Yuanzhi, Tengyu Ma, and Hongyang Zhang. "Algorithmic regularization in over-parameterized matrix sensing and neural networks with quadratic activations." Conference On Learning Theory. PMLR, 2018.
> > > > >
> > > > > [4] Razin, Noam, and Nadav Cohen. "Implicit regularization in deep learning may not be explainable by norms." Advances in neural information processing systems 33 (2020): 21174-21187.
> > > > >
> > > > > [5] Krantz, Steven G. A primer of mathematical writing. Vol. 243. American Mathematical Soc., 2017
> > > > >
> > > > > [6] Higham, Nicolas J. Handbook of writing for the mathematical sciences. Society for Industrial and Applied Mathematics, 2020.
> > > > >
> > > > > [7] Arora, Sanjeev, et al. "Generalization and equilibrium in generative adversarial nets (gans)." International conference on machine learning. PMLR, 2017.
> > > > >
> > > > > [8] Belkin, Mikhail, Siyuan Ma, and Soumik Mandal. "To understand deep learning we need to understand kernel learning." International Conference on Machine Learning. PMLR, 2018.
> > > > >
> > > > > [9] Anandkumar, Animashree, et al. "Tensor decompositions for learning latent variable models." J. Mach. Learn. Res. 15.1 (2014): 2773-2832.
> > > > >
> > > > > [10] Hanin, Boris, and David Rolnick. "How to start training: The effect of initialization and architecture." Advances in neural information processing systems 31 (2018).

---

### Official Review · Reviewer_KJ3P · 2024-11-01

**Soundness:** 3
**Presentation:** 3
**Contribution:** 3
**Rating:** 6
**Confidence:** 3

**Summary:**

This paper investigates the implicit bias of Structured State Space Models (SSMs), an emerging and efficient alternative to transformers. Previous research suggested that the implicit bias of SSMs enables good generalization when data is generated by a low-dimensional teacher model. However, this study revisits that assumption and uncovers a previously undetected phenomenon: while implicit bias generally promotes generalization, certain special training examples can completely distort this bias, causing generalization to fail. Notably, this failure occurs even when these examples have clean labels provided by the teacher model. The authors empirically demonstrate this effect in SSMs trained both independently and within non-linear neural networks. In adversarial machine learning, disrupting generalization with cleanly labeled examples is known as clean-label poisoning. Given the growing use of SSMs, especially in large language models, the authors emphasize the need for further research into their vulnerability to clean-label poisoning and the development of strategies to mitigate this susceptibility.

**Strengths:**

1. This paper presents a solid theoretical work, extending the results on the gradient descent (GD) implicit bias of Structured State Space Models (SSMs) from the population risk setup, as discussed in [1], to the finite empirical risk setup. The authors demonstrate that training a student SSM on sequences labeled by a low-dimensional teacher SSM exhibits an implicit bias conducive to generalization. Their dynamical analysis also establishes a connection with greedy low-rank learning.

2. Using an advanced tool from dynamical systems theory—the non-resonance linearization theorem—the authors prove that adding a single sequence to the training set, even if labeled by the teacher SSM (i.e., with a clean label), can completely distort the implicit bias to the point where generalization fails. This finding highlights the need for significant research efforts to better understand SSMs' vulnerability to clean-label poisoning and to develop effective methods to counteract this issue.

3. Overall, the paper is very well written.

**Reference**

[1] Cohen-Karlik, E., Menuhin-Gruman, I., Giryes, R., Cohen, N., & Globerson, A. (2022). Learning Low Dimensional State Spaces with Overparameterized Recurrent Neural Nets. arXiv preprint arXiv:2210.14064.

**Weaknesses:**

1. The assumptions in Theorem 1 are overly restrictive, particularly as the structures of $ A^* $, $ B^* $, and $ C^* $ seem to be very simple, and there is a lack of detailed explanation as to why such a simplified setup is justified in this context.

2. Regarding the special sequence data, Theorem 1 only provides an existence result without specifying a concrete construction method or offering a more detailed characterization, making it difficult for me to understand why the introduction of these clean data points leads to a failure in generalization.

3. The experiments in the paper seem to be conducted solely on synthetic datasets, which greatly limits their persuasiveness. The authors should consider using more real-world datasets to demonstrate the generality of the phenomenon they describe.

**Questions:**

1. Could the authors provide a more formal statement regarding greedy low-rank learning and clarify whether the theoretical analysis can be observed in experiments on real-world datasets?

2. For real-world datasets, can the theoretical analysis and conclusions of this paper help design an algorithm to identify the so-called special sequence for poisoning?

---

> ### Author Response · Authors · 2024-11-19
> **Author Response (Part 1/2)**
>
> Thank you for your review, and for highlighting the solidity of our theory and the clarity of our writing.  Thank you also for your constructive suggestions!  We address your comments and questions below.
>
> >The assumptions in Theorem 1 are overly restrictive, particularly as the structures of $A^*$, $B^*$, and $C^*$ seem to be very simple, and there is a lack of detailed explanation as to why such a simplified setup is justified in this context.
>
> The setting of Theorem 1 is indeed specific and relatively simple.  The importance of the theorem is that it formally proves existence  of clean-label poisoning, analogously to various existence proofs in deep learning theory (see, e.g, [1,2,3]).
>
> Despite the specificity and relative simplicity of its setting, proving Theorem 1 is highly non-trivial.  Indeed, our proof spans nearly 40 pages (Appendix C), and employs advanced tools from dynamical systems theory (see proof sketch, lines 288-323).  It is nonetheless possible to extend Theorem 1 in several ways, including accommodation of other values for $A^*$, $B^*$ and $C^*$.  We are now adding these extensions to the paper.  Thank you for raising the matter!
>
> >Regarding the special sequence data, Theorem 1 only provides an existence result without specifying a concrete construction method or offering a more detailed characterization, making it difficult for me to understand why the introduction of these clean data points leads to a failure in generalization.
>
> We refer as special sequences to ones in which the last elements are relatively large.  As discussed in the interpretation of our dynamical characterization (Section 3.1), such sequences can disrupt greedy low rank learning—a sufficient condition for generalization with a low dimensional teacher.  This is the intuition behind Theorem 1: the added cleanly labeled sequence $\mathbf{x}^\dagger$ has a large penultimate element, thus disrupts greedy low rank learning, which in turn can be shown to fail generalization.  Following your feedback, we will make this point clearer in the text.  Thank you!
>
> >The experiments in the paper seem to be conducted solely on synthetic datasets, which greatly limits their persuasiveness. The authors should consider using more real-world datasets to demonstrate the generality of the phenomenon they describe.
>
> We are now adding to the paper experiments demonstrating clean-label poisoning of SSMs in real-world scenarios.  Thank you for the feedback!

---

> > ### Author Response · Authors · 2024-11-19
> > **Author Response (Part 2/2)**
> >
> > >Could the authors provide a more formal statement regarding greedy low-rank learning and clarify whether the theoretical analysis can be observed in experiments on real-world datasets?
> >
> > In the context of other (non-SSM) deep learning models, dynamical characterizations analogous to ours (i.e., that reveal a greedy low rank learning phenomenon) have been used to formally prove approximation of low rank solutions (see, e.g, [4,5]).  It is possible to do the same with our dynamical characterization, and in fact, this is precisely what we do in proving the first bullet of Theorem 1.  Following your feedback, we will highlight this point in the text.
> >
> > With regards to experiments on real-world datasets, we are adding such experiments to the paper (as discussed above).  These showcase different aspects of our theory.
> >
> > Thank you for your fruitful questions!
> >
> > >For real-world datasets, can the theoretical analysis and conclusions of this paper help design an algorithm to identify the so-called special sequence for poisoning?
> >
> > Since real-world datasets deviate from the analyzed setting of a low dimensional teacher SSM, our theory does not directly apply to them.  Nonetheless, various studies argued that real-world datasets often admit “low dimensional structure” (see, e.g, [6,7]).  We thus believe it may be possible to draw on our conclusions for identifying cleanly labeled sequences that poison SSMs in real-world settings.  For example, it may be that in certain real-world settings, clean-label poisoning may be achieved by increasing the last elements of certain training sequences in certain ways.  Exploration of this prospect is a very interesting direction for future work.  We will mention it in the paper.  Thank you for asking!
> >
> > **References**
> >
> > [1] Telgarsky, Matus. "Benefits of depth in neural networks." Conference on learning theory. PMLR, 2016.
> >
> > [2] Vardi, Gal, and Ohad Shamir. "Implicit regularization in relu networks with the square loss." Conference on Learning Theory. PMLR, 2021.
> >
> > [3] Arjevani, Yossi, et al. "Lower bounds for non-convex stochastic optimization." Mathematical Programming 199.1 (2023): 165-214.
> >
> > [4] Li, Zhiyuan, Yuping Luo, and Kaifeng Lyu. "Towards resolving the implicit bias of gradient descent for matrix factorization: Greedy low-rank learning." arXiv preprint arXiv:2012.09839 (2020).
> >
> > [5] Razin, Noam, Asaf Maman, and Nadav Cohen. "Implicit regularization in hierarchical tensor factorization and deep convolutional neural networks." International Conference on Machine Learning. PMLR, 2022.
> >
> > [6] Pope, Phillip, et al. "The intrinsic dimension of images and its impact on learning." arXiv preprint arXiv:2104.08894 (2021).
> >
> > [7] Brown, Bradley CA, et al. "Verifying the union of manifolds hypothesis for image data." arXiv preprint arXiv:2207.02862 (2022).

---

> > > ### Author Response · Authors · 2024-11-28
> > > **Updates to Paper Following Reviewer Feedback**
> > >
> > > Dear reviewer,
> > >
> > > We have uploaded a new version of the paper that includes all changes we committed to (marked blue in the PDF). Below is a list of changes relevant to your review. In light of this update, we would greatly appreciate it if you would consider raising your score.
> > >
> > > Best wishes, and thank you for the illuminating feedback!
> > >
> > > Authors
> > >
> > > —
> > >
> > > List of relevant changes:
> > >
> > > - New experiment (reported in Section 4.2) demonstrates clean-label poisoning of SSMs in a real-world (non-synthetic) setting: SSM-based S4 neural network trained over sequential MNIST dataset.  In this experiment, we do not have access to a teacher (i.e., to a ground truth labeling function), and accordingly, cleanly labeled poisonous examples are generated from given training examples by introducing human-imperceptible noise to input sequences, while keeping their labels intact. The noise introduced for generating **a cleanly labeled poisonous example has its last entries relatively large, in line with our theory.**
> > >
> > > - The proof sketch of Theorem 1 further explains the intuition behind our construction, and its relation to our dynamical characterization (Section 3.1).  Specifically, the proof sketch now highlights the fact that the greedy low rank learning phenomenon established by our dynamical characterization is used to formally prove approximation of low rank solutions.
> > >
> > > - New Appendix E extends Theorem 1 in several ways, including accommodation of other values for $A^*$, $B^*$ and $C^*$.

---

> > > > ### Comment · Reviewer_KJ3P · 2024-11-28
> > > > **Response to Update from Authors**
> > > >
> > > > Thanks for your update! While I appreciate the efforts of authors for updating the revision, I still think the forms of $A^*, B^*, C^*$ in Theorem 1 extended by Appendix E are very simple due to the very sparse property of $A^*$ and the repeated coordinate values of $B^*$ and $C^*$. Thus, I will keep my scores.

---

### Official Review · Reviewer_Jomp · 2024-11-03

**Soundness:** 3
**Presentation:** 3
**Contribution:** 3
**Rating:** 6
**Confidence:** 1

**Summary:**

This paper examines the vulnerability of SSMs to clean-label poisoning. While SSMs generally possess an implicit bias that promotes generalization when trained on low-dimensional data from a teacher model, this study demonstrates that the inclusion of carefully chosen, correctly labeled training examples can disrupt this bias, resulting in a failure to generalize.

**Strengths:**

1). This paper discovered clean-label poisoning of SSMs, which is a vulnerability unrecognized by previous works, highlighting a potential risk in the safety, robustness, and reliability of SSMs.

2). This paper provides a strong theoretical foundation and empirical validation to show that generalization can be ruined by introducing certain clean-labeled sequences.

**Weaknesses:**

This paper falls outside my current area of expertise.

**Questions:**

This paper falls outside my current area of expertise.

---

> ### Author Response · Authors · 2024-11-19
> **Author Response**
>
> Thank you for your review, and for highlighting the strength of our theory and experiments!

---

### Official Review · Reviewer_Qi9A · 2024-11-03

**Soundness:** 3
**Presentation:** 4
**Contribution:** 3
**Rating:** 8
**Confidence:** 2

**Summary:**

This paper presents a fundamental understanding of SSMs by investigating its implicit bias both theoretically and empirically. From the extended setting of the previous work [1], the authors address several theoretical findings which are corroborated with experimental evidence. First, the authors provide a dynamical characterization of gradient flow over SSM, which reveals that greedy learning can be implicitly induced under many choices of training sequences. Second, based on the first point, the authors prove the followings; (1) a collection of sequences labeled by a low dimensional teacher SSM leads to the implicit bias of the student SSM, (2) adding a certain single sequence labeled by the teacher to the training set ruins the implicit bias which fails the generalization. These analyses are supported by simple experiments that are well consistent with the theories.


[1] Edo Cohen-Karlik, Itamar Menuhin-Gruman, Raja Giryes, Nadav Cohen, and Amir Globerson. Learning low dimensional state spaces with overparameterized recurrent neural nets. In International Conference on Learning Representations (ICLR), 2023

**Strengths:**

The theoretical analysis of the implicit bias of SSMs is a strength of this work. Providing a basic understanding of SSMs, which are often considered as an alternative to transformers, is indeed important for the community.

The paper is nicely written, while I'm not an expert in this area, I could easily follow the logical steps of the paper based on proper interpretation and the sketches of the mathematical details.

The experimental results that corroborates the theoretical analyses are a strength of this work. While the authors acknowledge the limitations of their work (to name a few, the experiments pertain near-zero initialization, the dimension of the teacher SSM is one in most settings), the identification of the phenomena and a theoretical modeling to explain such things is a meaningful contribution.

**Weaknesses:**

Again, I'm not an expert in this area, but I'd like to make a few points that might be helpful to the authors and also clarify my understanding.

1. Is it possible to separately decompose and plot $\gamma^{(0)}(t)$ along with Figure 2? If so, I think it could give a more concrete explanation to aid the Interpretation part of Section 3.1.
2. Can the authors come up with a measure to qualitatively distinguish between the scenarios of the (leftmost) and (second) subplot of Figure 2? I'm aware that this might be an abrupt question, but it could be seen that the (second) subplot does not correspond to the case of 'not greedy', maybe close to 'less greedy' (especially for the case of Figure 4 in the supplementary material).
3. (Slightly related to 2.) Adding results of more individual runs/seeds for Figure 2 could more strengthen the authors' explanation. Could be an indirect way to show this, but showing only one trial as confirmation to the theory seems less sufficient.
4. Out of the scope of the teacher-student setting (i.e. in the real-world scenario), what could be the interpretation or example for $(\mathbf{x}^{\dagger}, y^{\dagger})$ of Theorem 1 that harms the student from generalization?
5. Adding a short discussion of the limitations with respect to the corresponding results from the supplementary materials would make the paper more complete. In addition, could the authors provide more experimental results beyond the limitation setting made in the supplementary materials (i.e., results for the teacher dimension greater than 2)? Providing these additional results is not necessary, and I do not want this point to be an extensive burden. But I believe that effectively showing such limitations is indeed important from a completeness perspective.

**Questions:**

Please refer to the weaknesses above.

---

> ### Author Response · Authors · 2024-11-19
> **Author Response**
>
> Thank you for your review, and for highlighting the strength of our theory and experiments, as well as the clarity of our writing.  Thank you also for your constructive suggestions!  We address your comments below.
>
> >Is it possible to separately decompose and plot $γ^{ (0) }(t)$ along with Figure 2? If so, I think it could give a more concrete explanation to aid the Interpretation part of Section 3.1.
>
> We are adding to the paper plots of $\gamma^{ (0) } ( t )$ in the context of Figure 2.  Thank you for the suggestion!
>
> >Can the authors come up with a measure to qualitatively distinguish between the scenarios of the (leftmost) and (second) subplot of Figure 2? I'm aware that this might be an abrupt question, but it could be seen that the (second) subplot does not correspond to the case of 'not greedy', maybe close to 'less greedy' (especially for the case of Figure 4 in the supplementary material).
>
> For assessing the extent to which greedy low rank learning occurs, one may apply to the state transition matrix $A$ the measure of effective rank, which can be seen as a continuous version of rank [1].  For a given teacher model, the stronger the effect of greedy low rank learning is, the lower the effective rank throughout learning will be.  This becomes apparent when plotting the effective rank in the context of Figures 2 and 4.  Following your feedback, we are adding such plots to the paper.  Thank you very much for the suggestion!
>
> >(Slightly related to 2.) Adding results of more individual runs/seeds for Figure 2 could more strengthen the authors' explanation. Could be an indirect way to show this, but showing only one trial as confirmation to the theory seems less sufficient.
>
> We are adding to the appendix additional demonstrations of our dynamical characterization, obtained with different random seeds.  Thank you!
>
> >Out of the scope of the teacher-student setting (i.e. in the real-world scenario), what could be the interpretation or example for $(x^\dagger,y^\dagger)$ of Theorem 1 that harms the student from generalization?
>
> In real-world scenarios, it is often very difficult to interpret cleanly labeled training examples that lead generalization to fail.  Indeed, coming up with such examples typically requires substantial computational efforts (see, e.g., [2,3,4,5]). While our theory identifies sequences in which the last elements are relatively large to be ones that alter the implicit bias (Section 3.1), we prove that such sequences can lead generalization to fail only in teacher-student settings, not in real-world scenarios.  To gain insight into the kind of sequences leading generalization to fail in real-word scenarios, we are adding to the paper real-world experiments demonstrating clean-label poisoning of SSMs.  Thank you for raising the matter!
>
> >Adding a short discussion of the limitations with respect to the corresponding results from the supplementary materials would make the paper more complete. In addition, could the authors provide more experimental results beyond the limitation setting made in the supplementary materials (i.e., results for the teacher dimension greater than 2)? Providing these additional results is not necessary, and I do not want this point to be an extensive burden. But I believe that effectively showing such limitations is indeed important from a completeness perspective.
>
> We are adding to the appendix an account of its limitations, as well as additional experiments (e.g., with a teacher of dimension greater than two, and with real-world data).
>
> **References**
>
> [1] O. Roy and M. Vetterli, "The effective rank: A measure of effective dimensionality," 2007 15th European Signal Processing Conference, Poznan, Poland, 2007, pp. 606-610.
>
> [2] Huang, W. Ronny, et al. "Metapoison: Practical general-purpose clean-label data poisoning." Advances in Neural Information Processing Systems 33 (2020): 12080-12091.
>
> [3] Geiping, Jonas, et al. "Witches' brew: Industrial scale data poisoning via gradient matching." arXiv preprint arXiv:2009.02276 (2020).
>
> [4] Shafahi, Ali, et al. "Poison frogs! targeted clean-label poisoning attacks on neural networks." Advances in neural information processing systems 31 (2018).
>
> [5] Aghakhani, Hojjat, et al. "Bullseye polytope: A scalable clean-label poisoning attack with improved transferability." 2021 IEEE European symposium on security and privacy (EuroS&P). IEEE, 2021.

---

> > ### Comment · Reviewer_Qi9A · 2024-11-25
> >
> > Thank you for your thoughtful response. While the authors have indicated that they will update their manuscript, I would appreciate seeing the primary results for particularly addressing weaknesses 1 and 2 (and, if possible, 4). I would be happy to update my score after confirming that the additional results align with the authors' claims.

---

> > > ### Author Response · Authors · 2024-11-28
> > > **Author Response to Reviewer Reply**
> > >
> > > Dear Reviewer,
> > >
> > > Thank you for the engagement, and for the willingness to increase your score!
> > >
> > > We have uploaded a new version of the paper that includes all changes we committed to (marked blue in the PDF).  In particular, this new version treats weaknesses 1, 2 and 4 in your review, through the changes respectively specified below.
> > >
> > > Thank you for your enlightening feedback throughout this process.
> > >
> > > Authors
> > >
> > > —
> > > - New Figures 16 to 19 (in Appendix F.1) plot $\gamma^{ 0 } ( t )$ in the context of Figures 2, 3 and other (new) figures demonstrating our dynamical characterization.  The plots clearly align with our interpretation of Proposition 1: when $\gamma^{ 0 } ( t )$ has large magnitude, greedy low rank learning does not take place, and vice versa.
> > >
> > > - New Figures 9 to 15 (in Appendix F.1) plot the effective rank in the context of Figures 2, 3, 4 and other (new) figures demonstrating our dynamical characterization.  The plots clearly show that including special sequences in the training set increases the effective rank.  This accords with our interpretation of such sequences impeding greedy low rank learning.
> > >
> > > - New experiment (reported in Section 4.2) demonstrates clean-label poisoning of SSMs in a real-world (non-synthetic) setting: SSM-based S4 neural network trained over sequential MNIST dataset.  In this experiment, we do not have access to a teacher (i.e., to a ground truth labeling function), and accordingly, cleanly labeled poisonous examples are generated from given training examples by introducing human-imperceptible noise to input sequences, while keeping their labels intact.  The noise introduced for generating **a cleanly labeled poisonous example has its last entries relatively large, in line with our theory.**

---

> > > > ### Author Response · Authors · 2024-12-01
> > > > **Reminder**
> > > >
> > > > Dear Reviewer,
> > > >
> > > > We are eagerly awaiting your response.
> > > >
> > > > As stated in our previous message, the newly uploaded version of the paper includes treatment of weaknesses 1, 2 and 4 in your review (as well as all other changes we committed to).
> > > >
> > > > We believe your feedback has greatly improved our work.  Thank you very much for this improvement!  Thank you also for being willing to acknowledge it by updating your score.
> > > >
> > > > Best wishes,
> > > >
> > > > Authors

---

> ### Comment · Reviewer_Qi9A · 2024-12-02
>
> I apologize for the late response. First of all, thank you for clarifying my questions with additional experiments. After reading the comment from the authors, including the additional results and the updated manuscript, I'm glad to see that the newly added results stay align to the authors original claims. I have updated my score accordingly.

---

> > ### Author Response · Authors · 2024-12-02
> > **Final Response**
> >
> > Thank you for your thoughtful feedback and for taking the time to reconsider your score. Your insights have greatly improved the quality of our work, and we sincerely appreciate your efforts.
> >
> > Best regards,
> >
> > Authors

---

### Meta-Review · Area_Chair_CY8c · 2024-12-23

**Metareview:**

Summary of Contributions:
This paper addresses the implicit bias in Structured State Space Models (SSMs) and demonstrates a theoretical phenomenon: while such models typically generalize when trained using sequences generated by a low-dimensional teacher, their generalization can break down upon adding certain carefully crafted but cleanly labeled examples. The authors provide a dynamical characterization of gradient flow on SSMs and prove a theoretical result that identifies the existence of so-called “clean-label poisoning” examples. Additionally, they support their theory with synthetic experiments and attempt to highlight relevance to scenarios outside simple teacher-student setups by experimenting on a real-world (sequential MNIST) setting.

Strengths:
	1.	Novelty and Insight: The paper sheds light on an under-explored aspect of SSMs—namely, their susceptibility to clean-label data poisoning. Prior work largely suggested that the implicit bias of these models leads to good generalization in low-dimensional teacher-student setups. This paper identifies a nuanced and previously unknown vulnerability.
	2.	Theoretical Rigor: The authors present a non-trivial dynamical characterization that employs advanced mathematical tools. The work extends the current theoretical understanding of the implicit bias in SSMs, linking it to a “greedy low-rank learning” phenomenon.

**Additional Comments On Reviewer Discussion:**

Weaknesses and Reviewer Concerns:
	1.	Over-Specificity and Practical Significance: The primary theoretical result (Theorem 1) focuses on a highly specific, simplified scenario. Certain reviewers were concerned that the constructed example is too contrived and does not convincingly establish the vulnerability in a setting resembling common practice. While the existence proof is mathematically involved, its direct applicability to complex, real-world SSM training scenarios is unclear.
	2.	Restricted Assumptions and Data Setup: Some reviewers pointed out that the training sequences are highly constrained (e.g., sequences differing by scaling factors, specific conditions on last entries, diagonal structure of the state matrix, fixed auxiliary matrices). They questioned whether such conditions were essential or artificially simplified. It was suggested that the lack of “persistence of excitation” makes the identification problem trivial and unrealistic.
	3.	Writing and Presentation Style: One reviewer criticized the embedding of assumptions and core technical definitions within the main text rather than using clear assumption environments or stating them upfront. This created difficulty in evaluating the clarity and scope of the theoretical claims.
	4.	Reviewer Disagreement and Limited Convincing Power of Experiments: While one reviewer (Qi9A) was initially satisfied after the authors’ revisions, another (mXtt) remained unconvinced. The main sticking point was that the scenario presented appears too far removed from standard practice. The experiments, initially mostly synthetic, were later complemented by a real-world example on sequential MNIST, but this addition did not fully alleviate concerns about the constructed theoretical setting’s realism. The final assessment was that the revisions did not address the core conceptual reservations about the contrived nature of the example.

Overall Assessment:
The paper tackles an interesting theoretical question and identifies a plausible vulnerability in SSMs. However, the majority of the concerns revolve around whether the main theoretical result—though mathematically elegant and rigorous—is relevant and robust enough to influence the community’s understanding of practical SSM training. The proof-of-concept is rigorous but too narrowly tailored, and despite the addition of a real-world experiment, it does not convincingly bridge the gap to typical SSM use cases (e.g., large language models). Furthermore, some reviewers remain unsatisfied with the communication of assumptions and the extent to which the result can be considered meaningful in general scenarios.

Recommendation:
While the idea is interesting and the theory non-trivial, the paper in its current form does not sufficiently address concerns about the practical implications and generality of its core theoretical demonstration. Consequently, I recommend rejection at this time. The authors are encouraged to broaden their theoretical setting, demonstrate persistence of excitation or other realistic conditions, clarify assumptions and definitions, and provide more extensive empirical support connecting the theoretical insights to practical, real-world scenarios.

---

### Decision · Program_Chairs · 2025-01-22

Reject